# Noise Stability Optimization for Finding Flat Minima: A Hessian-based Regularization Approach

**Hongyang R. Zhang**                                    *ho.zhang@northeastern.edu*
*Northeastern University, Boston*

**Dongyue Li**                                           *li.dongyu@northeastern.edu*
*Northeastern University, Boston*

**Haotian Ju**                                           *ju.h@northeastern.edu*
*Northeastern University, Boston*

**Reviewed on OpenReview:** *https://openreview.net/forum?id=yfrNkb2Ldd*

## Abstract

The training of over-parameterized neural networks has received much study in recent literature. An important consideration is the regularization of over-parameterized networks due to their highly nonconvex and nonlinear geometry. In this paper, we study noise injection algorithms, which can regularize the Hessian of the loss, leading to regions with flat loss surfaces. Specifically, by injecting isotropic Gaussian noise into the weight matrices of a neural network, we can obtain an approximately unbiased estimate of the trace of the Hessian. However, naively implementing the noise injection via adding noise to the weight matrices before backpropagation presents limited empirical improvements. To address this limitation, we design a two-point estimate of the Hessian penalty, which injects noise into the weight matrices along both positive and negative directions of the random noise. In particular, this two-point estimate eliminates the variance of the first-order Taylor's expansion term on the Hessian. We show a PAC-Bayes generalization bound that depends on the trace of the Hessian (and the radius of the weight space), which can be measured from data.

We conduct a detailed experimental study to validate our approach and show that it can effectively regularize the Hessian and improve generalization. First, our algorithm can outperform prior approaches on sharpness-reduced training, delivering up to a 2.4% test accuracy increase for fine-tuning ResNets on six image classification datasets. Moreover, the trace of the Hessian reduces by 15.8%, and the largest eigenvalue is reduced by 9.7% with our approach. We also find that the regularization of the Hessian can be combined with alternative regularization methods, such as weight decay and data augmentation, leading to stronger regularization. Second, our approach remains highly effective for improving generalization in pretraining multimodal CLIP models and chain-of-thought fine-tuning.

## 1 Introduction

The loss landscape and its geometry properties are a recurring theme in the study of neural networks (Keskar et al., 2017; Hochreiter & Schmidhuber, 1997). Recently, the design of training methods such as sharpness-aware minimization and stochastic weight averaging has led to empirical advances in a wide variety of settings (Izmailov et al., 2018; Foret et al., 2021; Wortsman et al., 2022). The theoretical study of these training methods has also been explored (Andriushchenko & Flammarion, 2022). For instance, recent work shows that sharpness-aware minimization (Foret et al., 2021) has an implicit bias to flat surface regions by penalizing the largest eigenvalue of the loss Hessian matrix (Wen et al., 2023; Bartlett et al., 2023). In this paper, we study methods that can provide explicit regularization of the trace of the Hessian, and we will

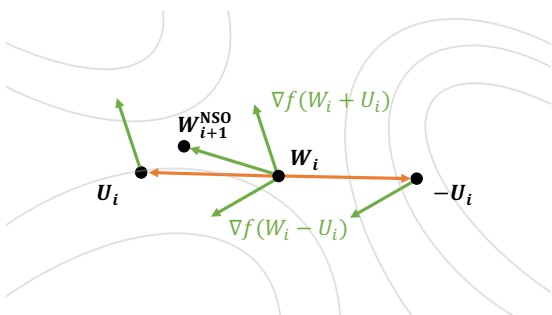

Figure 1: An illustration of one update step in our algorithm. At each iteration $i$, we sample a random variable $U_i$ from a zero-mean distribution $\mathcal{P}$ (e.g., an isotropic Gaussian with variance $\sigma^2$), where $\sigma$ is a hyper-parameter that controls the strength of the noise injection (hence the regularization). We query the gradient of $f$, at $f(W_i + U_i)$, and $f(W_i - U_i)$, and take their average. This results in a two-point noise injection scheme, whose computation cost is the same as sharpness-aware minimization (Foret et al., 2021), and twice the cost of running SGD. Notice that in practice, we can also implement an extension of this algorithm, which samples multiple $U$s. For details, see Algorithm 1.

show provable generalization guarantees of our methods. More formally, given an input function $f : \mathbb{R}^d \to \mathbb{R}$ that represents the empirical risk of a neural network and a $d$-dimensional distribution $\mathcal{P}$ with mean zero, we consider minimizing the noise-perturbed function

$$F(W) := \underset{U \sim \mathcal{P}}{\mathbb{E}} \left[ f(W + U) \right]. \tag{1}$$

Minimizing this perturbed function can improve the resilience of the neural network to noise injection, leading to flatter loss surfaces and improved regularization (Nagarajan & Kolter, 2020). By analyzing the perturbed loss of a fine-tuned model, one can identify a measure of the *sharpness* of loss surfaces based on the trace of the Hessian (Ju et al., 2022; 2023). We remark that the minimization problem of the form (1) traces back to earlier works on randomized smoothing (Duchi et al., 2012), which have provided a detailed study of convergence rates for nonsmooth stochastic optimization. *Our work differs from this line of literature in that we focus on evaluating the regularization effect of penalizing the Hessian trace upon neural network training.*

Although noise injection algorithms can be theoretically motivated as improving generalization (and stability), its practical implication is not evident (Hinton & Van Camp, 1993; An, 1996; Graves, 2011). To motivate our study, we begin by running several empirical studies to compare the performance of (standard) SGD and weight-perturbed SGD (WP-SGD), which first injects random noise into the weight matrices of a neural network before computing its gradient in SGD. As mentioned above, this would provide a randomized smoothing effect to the loss surface (Duchi et al., 2012). We will fine-tune (pretrained) ResNets on three image classification tasks for this empirical study. To ensure the robustness of the analysis, we also vary the distribution of $\mathcal{P}$ and the variance of $U$. Our overall finding is that WP-SGD (or randomized smoothing) does not offer clear benefits over SGD, which is also consistent with recent studies of weight noise injection (Orvieto et al., 2023; Dauphin et al., 2024) (see Section 2.2, Table 2 for the complete results). However, we hypothesize that these results may be due to the randomness of the noise injection (upon the Hessian penalty term) rather than the ineffectiveness of regularizing the Hessian trace.

Our approach to mitigate the randomness of the noise injection on the Hessian penalty involves two parts. First, we retrieve the gradient at $W - U$ to cancel out the first-order expansion term of $W + U$ (recall that $U$ is a random sample from $\mathcal{P}$). Meanwhile, the second-order expansion term remains the same after this cancellation. We term this modification a two-point noise injection scheme, which is reminiscent of two-point gradient estimates in zeroth-order optimization (Duchi et al., 2015). The difference in our setting is that this two-point averaging cancels out the first-order gradient term, thereby eliminating its variance on the Hessian

Table 1: Comparison between our approach (NSO) and SAM (Foret et al., 2021), based on inductive bias, generalization guarantee, and convergence rate. In particular, the inductive bias of SAM is based on the results of Wen et al. (2023). The list of notations used in the table is explained as follows. $\nabla^2 \ell$ refers to the Hessian matrix of the loss function $\ell$. $\lambda_1[\cdot]$ and $\text{Tr}[\cdot]$ refer to the largest eigenvalue and the trace of an input matrix. $\alpha$ refers to the trace norm, taken over the maximum of the entire hypothesis space and data distribution (including unseen test data). $r$ is the radius of the fine-tuning region measured in $\ell_2$ distance. $n$ is the number of samples in the training dataset. $T$ is the total number of iterations run by our algorithm.

| Approach | Inductive Bias | Generalization Guarantee | Convergence Rate |
|---|---|---|---|
| Sharpness-Aware Minimization (SAM) | $\lambda_1[\nabla^2 \ell]$ | - | - |
| Noise Stability Optimization (NSO) | $\text{Tr}[\nabla^2 \ell]$ | $\sqrt{\frac{\alpha r^2}{n}}$ (Theorem 2.1) | $\Theta\left(\frac{1}{\sqrt{T}}\right)$ (Section 4) |

penalty. Second, we sample multiple perturbations $U_1, U_2, \ldots, U_k$ at each epoch and take their averaged two-point (noise-injected) gradients. See Figure 1 for an illustration of one step.

A primary advantage of our approach compared to prior sharpness minimization algorithms is that our approach can provide an approximately unbiased estimate of the Hessian trace. We empirically validate this claim across three real-world settings (see Figure 2, Section 2.2 for an illustration). By utilizing this property, we show a PAC-Bayes bound that depends on the trace of the Hessian and the radius of the weight hypothesis space. We briefly describe this result, leaving a formal statement to Theorem 2.1. Let $\alpha$ be an upper bound on the trace of the Hessian measured within the hypothesis space and the data distribution (in practice, one may take this as the union of training and testing data). Let $r$ be the radius of the hypothesis space, measured in $\ell_2$ distance. Suppose there are $n$ empirical samples from an unknown distribution. We show a generalization bound that scales as $O\left(\sqrt{\frac{\alpha r^2}{n}}\right)$. Our proof utilizes a linear PAC-Bayes bound (Catoni, 2007; McAllester, 2013; Alquier, 2021), and we optimize the variance of the prior and posterior distributions to derive the result. A detailed proof sketch is presented in Section 2.3.

Next, we validate our approach with a detailed empirical study. First, we compare our approach with four prior approaches for the setting of fine-tuning pretrained ResNets, including sharpness-aware minimization (Foret et al., 2021), tested on six image classification datasets. We show that our algorithm can reduce the trace and the largest eigenvalue of the loss Hessian matrix by 15.8% and 9.7%, respectively. Our approach also improves test accuracy by 2.4%. Second, we show that by combining our approach with regularization methods such as data augmentation and distance-based regularization (Gouk et al., 2022), we can further regularize the Hessien, leading to 13.6% lower trace values and 16.3% lower test loss values (averaged over six tasks). Third, we extend our approach to pretraining and chain-of-thought fine-tuning. The details can be found in Section 3.2. Overall, our algorithm can consistently provide better regularization of Hessian and improved test accuracy across these different settings and datasets. Some of these empirical results are not completely explained by our theory, and we discuss the limitations in Section 7.

Finally, we analyze the convergence of our algorithm using techniques from the stochastic optimization literature (Ghadimi & Lan, 2013; Lan, 2020; Carmon et al., 2020; Drori & Shamir, 2020; Zhang, 2023), leading to matching upper and lower bounds. We also present a case study of Hessian regularization in over-parametrized matrix sensing and show that it is equivalent to nuclear norm regularization for this setting. Our work raises several new questions that may be worth revisiting: can accelerated gradient descent methods be applied to design flat-minima optimizers? Can recent advances in zeroth-order optimization be leveraged to better regularize the training of transformer neural networks?

In summary, the contributions of this paper are three-fold. First, we present an algorithm that can explicitly regularize the Hessian trace and show a PAC-Bayes generalization bound that could be measured from data. Second, we conduct experiments on multiple settings to validate our approach by comparing downstream performance and Hessian statistics with prior sharpness minimization algorithms and alternative regularization methods. Third, we analyze the convergence of our algorithm using stochastic optimization techniques. In Table 1, we highlight the key aspects of our approach compared to prior approaches.

**Organization:** The rest of this paper is organized as follows. In Section 2, we will present our approach. We will start by presenting the motivating experiments. Then, we describe our algorithm and a PAC-Bayes bound that depends on the Hessian. In Section 3, we present our experiments for validating the proposed approach. Section 4 presents an analysis of the convergence rate. Section 5 provides a case study of the regularization effect of the Hessian trace in the over-parameterized matrix sensing problem. In Section 6, we discuss the related works. Finally, in Section 7, we state the conclusion and the limitations of this work. We provide complete proof of our theoretical results in Appendix A-B. We provide additional experimental details in Appendix C.

## 2 Our Approach

In this section, we present our approach. First, to set up the stage, we will study the straightforward implementation of noise injection by directly adding noise to the weight matrices of the neural network before computing the gradients in backpropagation. We term this procedure weight-perturbed SGD (or WP-SGD in short), also known as randomized smoothing (Duchi et al., 2012). We will compare the empirical performance of these two approaches to evaluate the effect of noise injection. Then, we describe our algorithm and provide empirical measurements of the trace of the Hessian, along with the actual perturbation gaps observed in practice. Finally, we will show a PAC-Bayes generalization bound that depends on the trace of the Hessian, which can be measured from data to compare different methods.

### 2.1 Motivating Experiments

We compare the results from running WP-SGD to standard SGD. We choose the setting of fine-tuning pretrained foundation models, as overfitting is a common problem for this setting (Wortsman et al., 2022), and strong regularization is needed (Li & Zhang, 2021; Ju et al., 2022). We will fine-tune a pretrained ResNet-34 on several image classification datasets, including aircraft recognition (Aircraft) (Maji et al., 2013), indoor scene recognition (Caltech-256) (Griffin et al., 2007), and medical image classification (retina images for diabetic retinopathy classification) (Pachade et al., 2021). To implement WP-SGD, we sample a random vector from $\mathcal{P}$ and add it to the model weights at each iteration before computing the gradient. We set $\mathcal{P}$ as the isotropic Gaussian and adjust its standard deviation between $0.008, 0.01$, and $0.012$ via cross-validation.

We report our results in Table 2, which indicate that the performance gap is less than $0.5\%, \approx 0.75$ standard deviations based on five independent runs. Furthermore, varying $\mathcal{P}$ does not change the results. In particular, we test four types of $\mathcal{P}$, including Gaussian, Laplace, uniform, and Binomial. We adjust standard deviations between $0.008, 0.01$, and $0.012$ via cross-validation. We find that using Laplace and uniform distributions achieves a performance comparable to that of Gaussian. However, using Binomial results in worse results. These experiments suggest that the straightforward implementation of noise injection does not offer clear benefits over SGD.

### 2.2 Description of Our Algorithm

In our approach, we make two modifications to WP-SGD. First, we add the perturbation from both the positive and negative directions during the noise injection, as shown in line 5. Second, we average over multiple noise injections to reduce the variance from noise injection, as described in line 7. As for the first modification, recall that $\mathcal{P}$ is a symmetric distribution. We use Taylor's expansion on both $f(W + U)$ and $f(W - U)$ as follows:

$$f(W + U) = f(W) + \langle U, \nabla f(W) \rangle + \frac{1}{2} U^\top \nabla^2 f(W) U + O(\|\Sigma\|_2^{\frac{3}{2}}), \tag{2}$$

$$f(W - U) = f(W) - \langle U, \nabla f(W) \rangle + \frac{1}{2} U^\top \nabla^2 f(W) U + O(\|\Sigma\|_2^{\frac{3}{2}}). \tag{3}$$

Table 2: Comparing the outcome of running WP-SGD to standard SGD across four different $\mathcal{P}$, measured over three image classification datasets. Recall that WP-SGD refers to normal weight perturbation (without the paired perturbation). To be concise, we have included the results of running our approach (i.e., NSO). All the results and their standard deviations are based on five independent runs.

| | $\mathcal{P}$ | Aircraft | | Indoor | | Retina Disease | |
| | | Train Acc. | Test Acc. | Train Acc. | Test Acc. | Train Acc. | Test Acc. |
|---|---|---|---|---|---|---|---|
| SGD | None | $100.0\% \pm 0.0$ | $59.8\% \pm 0.7$ | $100.0\% \pm 0.0$ | $76.0\% \pm 0.4$ | $100.0\% \pm 0.0$ | $61.7\% \pm 0.8$ |
| WP-SGD | Gaussian | $98.4\% \pm 0.2$ | $60.4\% \pm 0.1$ | $99.0\% \pm 0.3$ | $76.3\% \pm 0.0$ | $100.0\% \pm 0.0$ | $62.3\% \pm 0.5$ |
| WP-SGD | Laplace | $98.3\% \pm 0.1$ | $60.3\% \pm 0.3$ | $98.9\% \pm 0.1$ | $76.4\% \pm 0.3$ | $100.0\% \pm 0.0$ | $62.0\% \pm 0.1$ |
| WP-SGD | Uniform | $98.6\% \pm 0.3$ | $60.3\% \pm 0.5$ | $98.6\% \pm 0.3$ | $76.6\% \pm 0.1$ | $100.0\% \pm 0.0$ | $62.3\% \pm 0.0$ |
| WP-SGD | Binomial | $19.6\% \pm 0.1$ | $11.3\% \pm 0.1$ | $18.2\% \pm 0.9$ | $10.7\% \pm 0.1$ | $58.1\% \pm 0.1$ | $57.1\% \pm 0.0$ |
| NSO | Gaussian | $95.8\% \pm 0.4$ | $62.3\% \pm 0.3$ | $95.7\% \pm 0.2$ | $77.4\% \pm 0.3$ | $100.0\% \pm 0.0$ | $66.6\% \pm 0.7$ |
| NSO | Laplace | $96.5\% \pm 0.3$ | $61.9\% \pm 0.3$ | $96.1\% \pm 0.3$ | $77.1\% \pm 0.1$ | $100.0\% \pm 0.0$ | $65.9\% \pm 0.1$ |
| NSO | Uniform | $96.4\% \pm 0.4$ | $61.9\% \pm 0.5$ | $96.4\% \pm 0.2$ | $76.8\% \pm 0.2$ | $100.0\% \pm 0.0$ | $65.7\% \pm 0.1$ |
| NSO | Binomial | $20.1\% \pm 0.1$ | $14.3\% \pm 0.3$ | $22.8\% \pm 0.1$ | $17.9\% \pm 0.2$ | $59.2\% \pm 0.1$ | $57.8\% \pm 0.1$ |

We have that $\mathbb{E}\left[U\right] = 0$, and $\mathbb{E}\left[UU^\top\right] = \Sigma$. Thus, by taking the average of equations (2) and (3), we get

$$\mathop{\mathbb{E}}_{U \sim \mathcal{P}}\left[\frac{1}{2}(f(W + U) + f(W - U))\right] = F(W) = f(W) + \frac{1}{2}\langle\Sigma, \nabla^2 f(W)\rangle + O\left(\|\Sigma\|_2^{\frac{3}{2}}\right). \qquad (4)$$

We can see that the two-point estimate eliminates the first-order gradient term, potentially reducing its variance in estimating the Hessian term. The second modification reduces the variance of the stochastic gradient, using the fact that each perturbation is independent of the others. The entire procedure is summarized in Algorithm 1. As a remark, two-point gradient estimators are commonly used in zeroth-order optimization (Duchi et al., 2015). However, their use in designing flat minima optimizers has not been explored much.

---

**Algorithm 1 Noise stability optimization (NSO) for regularizing the Hessian of neural networks**

---

**Input**: Initialization $W_0 \in \mathbb{R}^d$, a function $f : \mathbb{R}^d \to \mathbb{R}$
**Require**: An estimator $g : \mathbb{R}^d \to \mathbb{R}^d$ that for any $W$, returns $g(W)$ s.t. $\mathbb{E}\left[g(W)\right] = \nabla f(W)$
**Parameters:** # perturbations $k$, # epochs $T$, step sizes $\eta_0, \ldots, \eta_{T-1}$

1: **for** $i = 0, 1, \ldots, T - 1$ **do**
2:     /*    *Compute the two-point averaged stochastic gradient for each independent noise injection*    */
3:     **for** $j = 0, 1, \ldots, k - 1$ **do**
4:         $U_i^{(j)} \leftarrow$ sampled independently from $\mathcal{P}$
5:         $G_i^{(j)} \leftarrow g\big(W_i + U_i^{(j)}\big) + g\big(W_i - U_i^{(j)}\big)$
6:     **end for**
7:     $W_{i+1} \leftarrow W_i - \eta_i \left(\frac{1}{2k}\sum_{j=1}^k G_i^{(j)}\right)$
8: **end for**

---

**Measurements of the Hessian trace and the perturbation gap:** Next, we provide several examples to measure the approximation quality of equation (4). Following the experimental setup of Section 2.1, we will fine-tune a foundation model on a downstream task. After training, we will set $W$ as the model weight at the last epoch for all the measurements.

To measure equation (4), we then add $U$ to $W$, where $U$ is sampled from an isotropic Gaussian. We will measure $f(W + U) - f(W)$, averaged over 100 independent samples of $U$, and we measure this and $\nabla^2 f(W)$ by taking the average over the training dataset.

The results are shown in Figure 2. We can see that $\nabla^2 f$ provides an accurate approximation to $F(W) - f(W)$ for various values of $\sigma$. In particular, the approximation error of equation (4) using the Hessian trace is less

than 3%. As a remark, the range of $\sigma^2$ differs across architectures because of the differing scales of their weights. More details about the neural network architectures can be found in Appendix C.

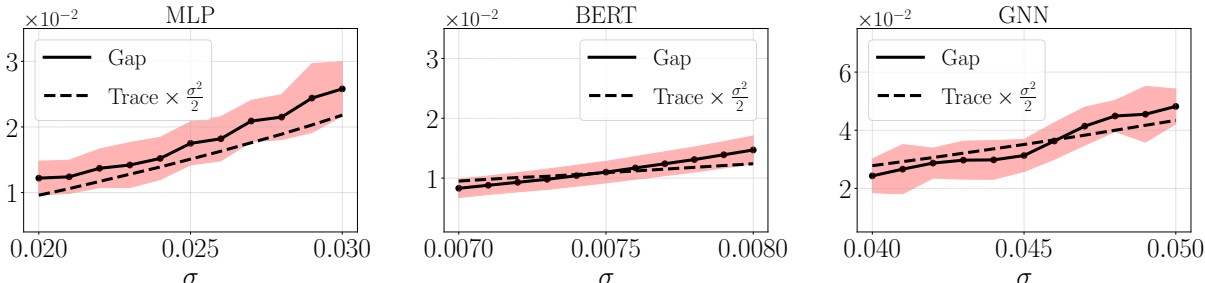

Figure 2: Illustration of the approximation quality of equation (4). We report all measurements based on the network weight at the last epoch of fine-tuning. We can see that the perturbation gap (i.e., $F(W) - f(W)$ in equation (4)) and $\frac{\sigma^2}{2} \text{Tr}[\nabla^2 f(W)]$ are at the same order. Recall that $\sigma$ refers to the standard deviation of the Gaussian noise injected into the weight matrices. More specifically, $\sigma$ will decide the strength of noise injection or the strength of regularization on the Hessian trace.

## 2.3 Generalization Guarantee and Proof Sketch

Next, we present a PAC-Bayes generalization bound that depends on the trace of the Hessian. Our bound can be related to the notion of trace norm, which has been used in earlier works for quantifying sample complexity in the context of matrix recovery (Srebro & Shraibman, 2005).

Concretely, suppose we have a pretrained model in the fine-tuning setting. This can be viewed as our prior belief of the target hypothesis in PAC-Bayes analysis. Once we have learned a model (though fine-tuning), we can view this as the posterior in PAC-Bayes analysis. Let $\mathcal{D} \subseteq \mathcal{X} \times \mathcal{Y}$ be an unknown data distribution, supported on the feature space $\mathcal{X}$ and the label space $\mathcal{Y}$. Given $n$ random samples $(x_1, y_1), (x_2, y_2), \ldots, (x_n, y_n)$ drawn from $\mathcal{D}$, the empirical loss (measured by loss function $\ell$) applied to a model $f_W$ (with $W \in \mathbb{R}^p$) is:

$$\hat{L}(W) = \frac{1}{n} \sum_{i=1}^{n} \ell(f_W(x_i), y_i).$$

The population loss is $L(W) = \mathbb{E}_{(x,y) \sim \mathcal{D}} \left[ \ell(f_W(x), y) \right]$. It is sufficient to think that the empirical loss is less than the population loss, and the goal is to bound the gap between $\hat{L}(W)$ and $L(W)$ from above (Shalev-Shwartz & Ben-David, 2014).

Let $W$ be any learned hypothesis within the hypothesis space, denoted as $\mathcal{H}$. Our generalization bound will apply uniformly to $W$ within the hypothesis space. We state our result, including the required assumptions, as follows.

**Theorem 2.1.** *Assume that the loss function $\ell$ is bounded between $0$ and $C$ for a fixed constant $C > 0$ on the data distribution $\mathcal{D}$. Suppose $\ell(f_W(\cdot), \cdot)$ is twice-differentiable in $W$ and the Hessian matrix $\nabla^2[\ell(f_W(\cdot), \cdot)]$ is Lipschitz continuous within the hypothesis space. Suppose for any $W$ in $\mathcal{H}$, the trace norm of the Hessian is less than $\alpha$:*

$$\alpha := \max_{W \in \mathcal{H}} \max_{(x,y) \sim \mathcal{D}} \text{Tr} \left[ \nabla^2 \ell(f_W(x), y) \right], \tag{5}$$

*and the $\ell_2$-norm of $W$ is at most $r$ for any $W \in \mathcal{H}$. Then, for any $W$ in $\mathcal{H}$, with probability at least $1 - \delta$ for any $\delta > 0$, the following must hold, for any $\epsilon$ close to zero:*

$$L(W) \le (1 + \epsilon)\hat{L}(W) + (1 + \epsilon)\sqrt{\frac{C\alpha r^2}{n}} + O\left(n^{-\frac{3}{4}} \log(\delta^{-1})\right). \tag{6}$$

**Proof Sketch:** We provide a high-level illustration of the proof of Theorem 2.1. Let $\mathcal{Q}$ denote the *posterior* distribution. Specifically, we consider $\mathcal{Q}$ as being centered at the learned hypothesis $W$ (which could be anywhere within the hypothesis space), given by a Gaussian distribution $\mathcal{N}(W, \sigma^2 \operatorname{Id}_p)$, where $\operatorname{Id}_p$ denotes the $p$ by $p$ identity matrix. Given a sample $U \sim \mathcal{N}(0, \sigma^2 \operatorname{Id}_p)$, let the perturbed loss be given by

$$\ell_{\mathcal{Q}}(f_W(x), y) = \mathbb{E}_U \left[\ell(f_{W+U}(x), y)\right]. \tag{7}$$

Then, let $\hat{L}_{\mathcal{Q}}(W)$ be the averaged value of $\ell_{\mathcal{Q}}(f_W(\cdot), \cdot)$, taken over $n$ empirical samples from the training dataset. Likewise, let $L_{\mathcal{Q}}(W)$ be the population average of $\ell_{\mathcal{Q}}(f_W(\cdot), \cdot)$, in expectation over an unseen data sample from the underlying data distribution.

Having introduced the notations, we start with the linear PAC-Bayes bound (Catoni, 2007; McAllester, 2013; Alquier, 2021) (see Theorem A.1 for reference), stated as follows, which holds with probability $1 - \delta$ for any $\delta \in (0, 1)$ and $\beta \in (0, 1)$:

$$L_{\mathcal{Q}}(W) \leq \frac{1}{\beta}\hat{L}_{\mathcal{Q}}(W) + \frac{C(KL(\mathcal{Q}||\mathcal{P}) + \log(\delta^{-1}))}{2\beta(1-\beta)n}, \tag{8}$$

where $\mathcal{P}$ refers to the *prior* distribution, $C$ refers to the upper bound on the loss value $\ell$. For analyzing fine-tuning, we view $\mathcal{P}$ as centered at the pretrained model, with covariance matrix $\sigma^2 \operatorname{Id}_p$. By Taylor's expansion of $\ell_{\mathcal{Q}}$ (see Lemma A.4 for the precise statement), we show that:

$$L_{\mathcal{Q}}(W) = L(W) + \frac{\sigma^2}{2} \mathbb{E}_{(x,y)\sim\mathcal{D}} \left[\operatorname{Tr}\left[\nabla^2 \ell(f_W(x), y)\right]\right] + O(\sigma^3) \tag{9}$$

$$\hat{L}_{\mathcal{Q}}(W) = \hat{L}(W) + \frac{\sigma^2}{2n} \sum_{i=1}^{n} \operatorname{Tr}\left[\nabla^2 \ell(f_W(x_i), y_i)\right] + O(\sigma^3). \tag{10}$$

Since the Hessian operator is Lipschitz continuous by the assumption of Theorem 2.1, we can bound the gap between the above two quantities with $\epsilon$-covering arguments (see Lemma A.5 for the precise statement). By plugging in these results back to the PAC-Bayes bound of equation (8), after some calculation, we can get:

$$L(W) \leq \frac{1}{\beta}\hat{L}(W) + \frac{\sigma^2(1-\beta)\alpha}{2\beta} + \frac{Cr^2/2\sigma^2}{2\beta(1-\beta)n} + O\left(\sigma^3 + \frac{\sigma^2\sqrt{p}}{\sqrt{n}} + \frac{\log(\delta^{-1})}{n}\right). \tag{11}$$

In particular, the above uses the fact that the $\ell_2$-norm of $W$ is less than $r$ for any $W \in \mathcal{H}$ (the KL divergence is discussed in Proposition A.2). By choosing $\sigma^2$ and $\beta$ to minimize equation (11), we will obtain equation (6). This summarizes the high-level proof idea. The complete proof can be found in Appendix A.1.

**Remark 2.2.** *We highlight two key aspects of our results. The first is that our PAC-Bayes bound is non-vacuous, meaning that it matches the scale of empirically observed gaps when measured in practice; this is based on the trace measurements in Figures 2 and 3. The second is that this non-vacuous bound has practical implications, meaning that we can utilize this bound to design optimization algorithms that improve generalization.*

*These are non-trivial to achieve. To give some context, prior work has provided a PAC-Bayes margin bound for multi-layer neural networks, which depends on the product of the spectral norm of the network layers (Neyshabur et al., 2018). While this paper provides important insights regarding the generalization of deep networks, the bound is vacuous when measured in practice. Arora et al. (2018) provide another data-dependent PAC-Bayes bound based on compression techniques. Their work started with an experiment in which they injected noise into the network layers and showed that deep nets can absorb the noise after retraining. However, their bound remains orders of magnitude higher than the actual generalization errors observed in practice.*

*In contrast, our bound matches the scale of empirically observed gaps. To achieve this, we start from the line of work on data-dependent PAC-Bayes bounds. We build on the line of work on distance from the initialization (Nagarajan & Kolter, 2020), which is ideal for understanding fine-tuning (Li & Zhang, 2021).*

*Our key breakthrough is to connect noise stability in PAC-Bayes bound with the loss Hessian matrix (cf. equations* (9) *and* (10)*). Then, we can measure the Hessian of loss landscapes from data.*

*We additionally note that few existing works have considered using PAC-Bayes bounds to design algorithms. The reason is that for new algorithm designs, we need to connect the PAC-Bayes bound with data in a non-vacuous way. The work of* Dziugaite & Roy (2017) *has provided a computational framework to achieve non-vacuous generalization bounds. Instead, our result provides an analytical expression that can be leveraged in algorithm design. To operationalize the design, we utilize the explicit dependence of our result on the Hessian to design the regularization scheme.*

## 3 Experiments

We now turn to empirical validations of our algorithm. First, we apply our approach to fine-tune pretrained ResNets on various image classification datasets. We find that NSO can more significantly regularize the Hessian of the loss surface, resulting in reductions in the trace and the largest eigenvalue by **15.8**% and **9.7**%, respectively. After controlling computation costs, it can outperform four sharpness-reducing methods by up to **2.4**%. In addition, we justify our algorithm design through detailed ablation analysis. We also show that our approach is compatible with alternative regularization techniques, including distance-based regularization and data augmentation, and combining these methods with our approach leads to more significant regularization and test performance. Second, we show similar results for pretraining and chain-of-thought fine-tuning. The experiment code for reproducing our empirical findings can be found online at: `https://github.com/VirtuosoResearch/Noise-stability-optimization`.

### 3.1 Comparison with Sharpness Minimization Methods

We now compare Algorithm 1 with five sharpness-reducing training methods, including sharpness-aware minimization (SAM) (Foret et al., 2021), unnormalized SAM (USAM) (Agarwala & Dauphin, 2023), adaptive variants of SAM (ASAM), and random SAM (RSAM) (Liu et al., 2022). During the comparison, we control for the same amount of computation (for Algorithm 1, we will set the number of sampled injections $k$ as 1). Thus, all the methods under consideration will use twice the computation of SGD. For NSO, we sample perturbation from an isotropic Gaussian distribution and adjust $\sigma$ between $0.008, 0.01$, and $0.012$. For SAM, we adjust the $\ell_2$ norm of the perturbation between $0.01, 0.02$, and $0.05$. For each method, we run it with both momentum and weight decay. We ensure that all the training methods are carefully adjusted. See Appendix C for the details.

#### 3.1.1 Empirical Findings

In Table 3, we report the comparison between NSO, SGD, SAM, unnormalized SAM (USAM), and adaptive SAM (ASAM). We find that our approach reduces the trace of Hessian by **15.8**% on average. The largest eigenvalue of the Hessian is also reduced by **9.7**%. This finding is intriguing since SAM has been motivated by a min-max problem. As for test accuracy, our approach can provide up to **2.4**% lift, with an average improvement of **1.2**%. Additional comparisons are deferred to Table 6 in Appendix C.

Figure 3 illustrates the measurements between SGD, WP-SGD, and NSO. Curiously, we find that the trace of the Hessian also decreases for SGD, possibly due to implicit norm control of SGD. While both WP-SGD and NSO reduce the trace of the Hessian, our approach penalizes the Hessian more. Besides, the generalization gap and the test loss are consistently lower during NSO training.

As a remark, the regularization effect of noise injection should be orthogonal to training methods such as momentum, weight decay, learning rate scheduling, etc. To this end, we performed comparisons without using either momentum or weight decay. Our approach can again reduce the trace of the Hessian by **17.7**% compared to the five sharpness-reducing methods on average, with up to **1.8**% higher test accuracy.

Table 3: Comparison between our approach (NSO) with SGD, sharpness-aware minimization (SAM), unnormalized SAM (USAM), and adaptive SAM (ASAM). We fine-tune the ResNet-34 network on six image classification datasets and report the test accuracy and the trace of Hessian using the model in the last epoch of training. The results are averaged over five random seeds.

| | | CIFAR-10 | CIFAR-100 | Aircrafts | Caltech-256 | Indoor | Retina |
|---|---|---|---|---|---|---|---|
| **Basic Statistics** | # Training | 45,000 | 45,000 | 3,334 | 7,680 | 4,824 | 1,396 |
| | # Validation | 5,000 | 5,000 | 3,333 | 5,120 | 536 | 248 |
| | # Test | 10,000 | 10,000 | 3,333 | 5,120 | 1,340 | 250 |
| | # Classes | 10 | 100 | 100 | 256 | 67 | 5 |
| **Trace** ($\downarrow$) | SGD | $4128 \pm 83$ | $13188 \pm 221$ | $5471 \pm 65$ | $3674 \pm 95$ | $3629 \pm 61$ | $28607 \pm 226$ |
| | SAM | $2429 \pm 87$ | $9227 \pm 286$ | $4499 \pm 70$ | $3285 \pm 95$ | $3159 \pm 75$ | $15444 \pm 173$ |
| | USAM | $2352 \pm 61$ | $7382 \pm 222$ | $4298 \pm 94$ | $3174 \pm 52$ | $3072 \pm 51$ | $12068 \pm 246$ |
| | ASAM | $2445 \pm 63$ | $9960 \pm 313$ | $4475 \pm 69$ | $3339 \pm 78$ | $3014 \pm 53$ | $14155 \pm 136$ |
| | **NSO** | $\mathbf{1728} \pm 79$ | $\mathbf{5244} \pm 89$ | $\mathbf{3678} \pm 83$ | $\mathbf{2958} \pm 77$ | $\mathbf{2737} \pm 90$ | $\mathbf{10970} \pm 146$ |
| **Test Acc.** ($\uparrow$) | SGD | $96.1\% \pm 0.1$ | $82.8\% \pm 0.1$ | $60.5\% \pm 0.7$ | $80.0\% \pm 0.1$ | $76.7\% \pm 0.4$ | $62.2\% \pm 0.8$ |
| | SAM | $97.0\% \pm 0.2$ | $84.0\% \pm 0.4$ | $62.3\% \pm 0.3$ | $77.0\% \pm 0.4$ | $77.2\% \pm 0.3$ | $65.0\% \pm 0.3$ |
| | USAM | $96.9\% \pm 0.2$ | $83.7\% \pm 0.2$ | $61.9\% \pm 0.3$ | $76.9\% \pm 0.2$ | $76.7\% \pm 0.3$ | $64.7\% \pm 0.1$ |
| | ASAM | $97.1\% \pm 0.1$ | $84.2\% \pm 0.3$ | $62.4\% \pm 0.5$ | $77.3\% \pm 0.2$ | $77.2\% \pm 0.2$ | $65.2\% \pm 0.3$ |
| | **NSO** | $\mathbf{97.6\%} \pm 0.4$ | $\mathbf{84.9\%} \pm 0.3$ | $\mathbf{63.2\%} \pm 0.3$ | $\mathbf{78.1\%} \pm 0.5$ | $\mathbf{78.2\%} \pm 0.3$ | $\mathbf{67.0\%} \pm 0.4$ |

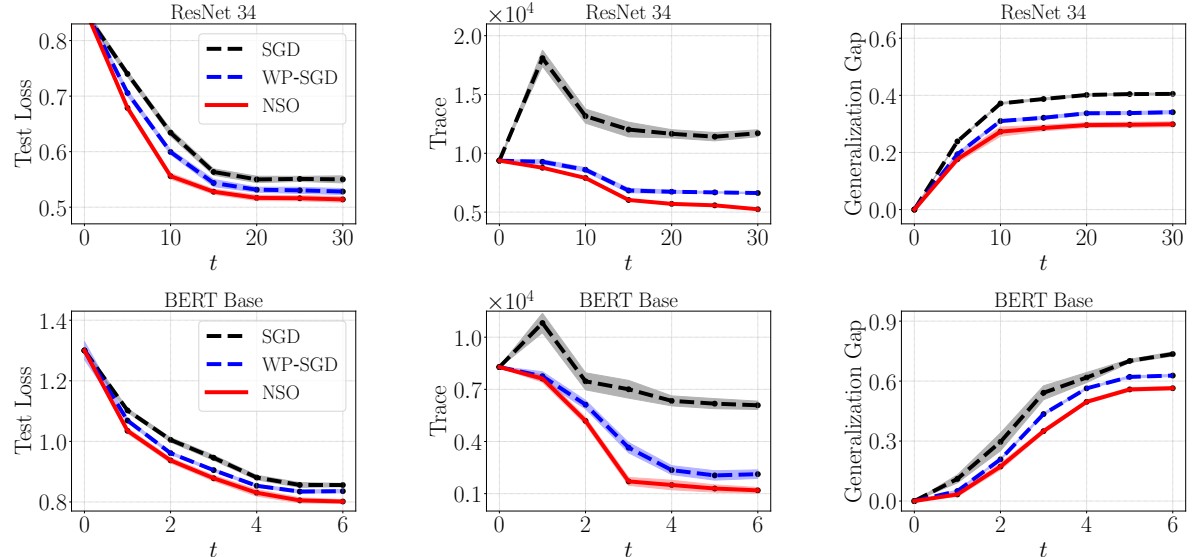

Figure 3: Comparison between SGD, WP-SGD, and NSO for fine-tuning ResNet-34 and BERT-Base, respectively, on an image and a text classification dataset. We evaluate the test loss, the trace of the Hessian, and the generalization gap for the trained model at each epoch. For WP-SGD and NSO, we sample noise from isotropic Gaussian with standard deviation $\sigma = 0.01$ in both settings.

### 3.1.2 Ablation Analysis

Next, we conduct ablation studies of two modifications in our approach: the use of negative perturbations and the sampling of multiple perturbations.

**Comparing using negative cancellation or not after controlling computation costs:** Recall that our algorithm uses negative perturbations to zero out the first-order term in Taylor's expansion of $F(W)$. We validate this by comparing the performance between using or not using the negative perturbation. We control for the same amount of computation costs to ensure a fair comparison. In particular, we sample two independent perturbations and take their averaged stochastic gradient. We find that using the nega-

tive perturbation achieves a **3.6**% improvement in test accuracy (on average) over not using the negative perturbation, i.e., randomized smoothing.

As discussed in Section 2.2, our intuition on why NSO can be expected to generalize better than randomized smoothing is that it can better regularize the Hessian. In particular, even though, in theory, the expectation of $f(W + U)$ and $\frac{1}{2}(f(W + U) + f(W - U))$ over $U$ are both equal to $F(W)$. However, the two-point scheme cancels out the gradient expansion term compared to randomized smoothing at every epoch. More precisely, we believe that the improved regularization from our approach stems from its better estimate of the Hessian penalty term. As illustrated in Figure 3, NSO consistently reduces the trace of the Hessian and achieves lower generalization errors compared to randomized smoothing throughout model training. At the end of the training, NSO yields **10.6**% smaller trace of the Hessian on average than randomized smoothing.

**Increasing the number of noise injections** $k$**:** Recall that increasing the number of perturbations $k$ can reduce the variance of the estimated gradient. Thus, we consider increasing $k$ in NSO and compare that with a specific implementation of WP-SGD that uses the same amount of computation. Using $k = 2$ perturbations improves the test accuracy by **1.2**% on average compared to $k = 1$.

**Varying the learning rate and the number of epochs.** We provide a detailed comparison between NSO and WP-SGD when varying the learning rate and the number of epochs. The learning rate is varied between $0.0001, 0.0002, 0.0005, 0.001, 0.002$, and $0.005$. The number of epochs is varied between $10, 20, 30, 40, 50$, and 60. We report the test loss from running an image classification task in Figure 4.

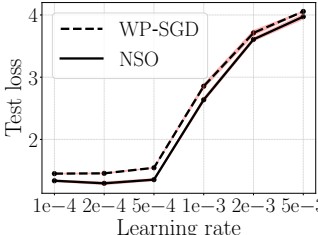
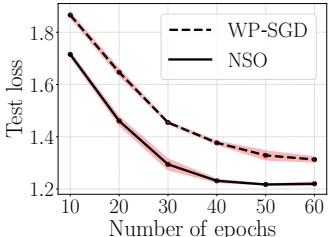

Figure 4: Results of varying the learning rate and the number of epochs for running our approach and WP-SGD. We report the test loss from the last epoch and average the results over five random seeds.

**Remark 3.1** (Noise variance scheduling as $k$ increases)**.** *A natural question is whether one can gradually increase or decrease the regularization strength by $\sigma$ during training, similar to learning rate scheduling. To facilitate this discussion, we test two schedules for adjusting $\sigma$. The first schedule is to increase $\sigma$ to a specified value at a linear rate. The second schedule exponentially increases $\sigma$ to reach a specified value. Our preliminary experiments show that neither schedule offers significant performance improvements over using a constant noise variance. One might also consider other scheduling schemes; we leave this to future work.*

### 3.1.3 Detailed Comparison with Sharpness-Aware Minimization (SAM)

**Varying the radius of SAM:** We provide a detailed comparison to SAM by varying the perturbation radius of SAM (denoted as $\rho$). To illustrate this comparison, we vary $\rho$ between $0.001, 0.002, 0.005, 0.01, 0.02$, and $0.05$. We report both the validation accuracy and the trace of the Hessian for SAM and unnormalized SAM on an image classification dataset. We present the results in Table 4. We observe that using a smaller $\rho$ (i.e., less than 0.01) results in worse results. Thus, we choose $\rho$ between $0.01, 0.02$, and $0.05$ in our experiments.

**Varying the batch size of SAM:** Next, we measure the sensitivity of our approach concerning the batch size. In particular, we vary the batch size between 8, 16, 32, and 64 for fine-tuning ResNet-34 on two image classification datasets. The results are shown in the leftmost two panels of Figure 5. We use the same number of epochs for each batch size configuration to ensure a fair comparison. On the indoor dataset, our approach is less sensitive to different batch sizes than SAM. Across all the batch sizes and datasets, our approach consistently provides a more robust regularization of the Hessian compared to SAM. The best results are achieved when the batch size is 32. Thus, we use this particular setting in our experiments.

Table 4: Results of varying the perturbation radius of SAM (denoted as $\rho$) and unnormalized SAM. We report both the test accuracy and the trace of the Hessian based on the model trained at the last epoch. We report the averaged results and their standard deviations across five random seeds.

| | $\rho$ | 0.001 | 0.002 | 0.005 | 0.01 | 0.02 | 0.05 |
|---|---|---|---|---|---|---|---|
| **Trace** | SAM | $4920 \pm 158$ | $4347 \pm 166$ | $4016 \pm 80$ | $3918 \pm 94$ | $3159 \pm 75$ | $\mathbf{3028} \pm 78$ |
| ($\downarrow$) | Unnormalized SAM | $4352 \pm 169$ | $3990 \pm 70$ | $3723 \pm 87$ | $3427 \pm 57$ | $3072 \pm 51$ | $\mathbf{3048} \pm 22$ |
| **Test Accuracy** | SAM | $73.6 \pm 0.2$ | $74.4 \pm 0.4$ | $74.8 \pm 0.6$ | $75.2 \pm 0.3$ | $\mathbf{76.6} \pm 0.5$ | $73.8 \pm 0.7$ |
| ($\uparrow$) | Unnormalized SAM | $74.1 \pm 0.1$ | $74.1 \pm 0.7$ | $74.7 \pm 0.5$ | $74.6 \pm 0.3$ | $\mathbf{76.3} \pm 0.3$ | $73.1 \pm 0.6$ |

(a) Loss on Indoor dataset  (b) Loss on Aircraft dataset  (c) Test loss, w/ dist. reg.  (d) Test loss, w/ data aug.

(e) Hessian, Indoor dataset  (f) Hessian, Aircraft dataset  (g) Hessian, w/ dist. reg.  (h) Hessian, w/ data aug.

Figure 5: Results of varying the batch size of our approach and SAM ran on two image classification datasets (indoor scene recognition and Aircraft detection). We report the test loss and the trace of Hessian using the model from the last epoch of training. The results are averaged over five random seeds. The regularization provided by noise injection can be combined with distance-based regularization and data augmentation to reduce the test loss and the Hessian trace.

### 3.1.4 Combining Algorithm 1 with Alternative Regularization Methods

In this section, we show that the regularization of the Hessian can serve as a complement to existing, alternative regularization methods. To validate this, we combine our training approach with data augmentation and distance-based regularization (Gouk et al., 2022). In particular, the latter approach has been used to regularize fine-tuning algorithms. We use a popular scheme for data augmentation that applies random horizontal flipping and random cropping sequentially to each training image. As for distance-based regularization, we penalize the $\ell_2$ distance between the fine-tuned model and the pretrained initialization.

The results are shown in Figure 5 within the two rightmost panels. Combining our approach with each regularization method further reduces the trace of the loss Hessian matrix by **13.6**% (on average). This further leads to **16.3**% lower test loss of the fine-tuned network, suggesting that our approach can be used on top of these preexisting regularization methods.

### 3.2 Applying Algorithm 1 to Pretraining and Fine-tuning

We apply our approach to pretraining randomly initialized models by replacing SGD to train contrastive language-image (CLIP) models on a dataset of image-caption pairs. In particular, we use the Conceptual

Caption dataset, which contains 3.3 million image caption pairs. Each caption briefly describes the corresponding image, with ten tokens on average. We use a 12-layer Vision Transformer as the image encoder and a 12-layer GPT-2 transformer as the text encoder. We train the encoders jointly to maximize the cosine similarity between the embedding of image caption pairs following the protocol of Radford et al. (2021).

Table 5 presents the results. For each algorithm, we evaluate the trace of the loss Hessian and recall scores (of the top-10 scored images in retrieving images from texts) on the development set. The results show that our approach can reduce the trace of the Hessian by **17**% compared to both SAM and SGD. In addition, our approach achieves **1.4**% higher recall scores in image retrieval.

Lastly, we apply our algorithm to fine-tuning pretrained language models on chain-of-thought reasoning datasets. The task is to generate the reasoning process, i.e., a chain of thoughts and the answer for a given commonsense reasoning question. We fine-tune pretrained GPT-2 models on two question-answering datasets: Commonsense QA and Strategy QA. Table 5 shows that our approach can yield **25**% lower trace values than SAM and SGD. In addition, we can obtain **5.3**% higher test accuracy.

Table 5: Results for pretraining CLIP the Conceptual Caption and chain-of-thought fine-tuning on Commonsense/Strategy QA. We report the recall score of image retrieval/test accuracy and trace/$\lambda_1$ using the model at the last epoch. We report the averaged results and standard deviations over five random seeds.

| Conceptual Caption | **Trace** ($\downarrow$) | $\boldsymbol{\lambda_1}$ ($\downarrow$) | **Recall@10** ($\uparrow$) |
|---|---|---|---|
| SGD | $220 \pm 24$ | $41 \pm 2.8$ | $36.1\% \pm 0.3$ |
| SAM | $144 \pm 20$ | $30 \pm 1.1$ | $36.9\% \pm 0.4$ |
| **NSO** | $\mathbf{119} \pm 34$ | $\mathbf{22} \pm 1.2$ | $\mathbf{37.5}\% \pm 0.3$ |
| CommonsenseQA | **Trace** ($\downarrow$) | $\boldsymbol{\lambda_1}$ ($\downarrow$) | **Test Accuracy** ($\uparrow$) |
| SGD | $372 \pm 34$ | $19 \pm 0.8$ | $27.7\% \pm 1.8$ |
| SAM | $288 \pm 15$ | $15 \pm 0.3$ | $32.7\% \pm 1.4$ |
| **NSO** | $\mathbf{208} \pm 31$ | $\mathbf{13} \pm 0.6$ | $\mathbf{39.2}\% \pm 1.4$ |
| StrategyQA | **Trace** ($\downarrow$) | $\boldsymbol{\lambda_1}$ ($\downarrow$) | **Test Accuracy** ($\uparrow$) |
| SGD | $294 \pm 13$ | $44 \pm 1.5$ | $68.9\% \pm 1.0$ |
| SAM | $249 \pm 33$ | $42 \pm 2.6$ | $71.1\% \pm 1.2$ |
| **NSO** | $\mathbf{193} \pm 31$ | $\mathbf{33} \pm 1.8$ | $\mathbf{75.2}\% \pm 1.2$ |

## 4  Convergence Rates

We now study the convergence of Algorithm 1. Recall that our algorithm minimizes $f(W)$ plus a regularization term on the Hessian trace. As is typical with regularization, the penalty is usually small relative to the loss value. Thus, we aim to find a stationary point of $F(W)$ instead of $f(W)$ because otherwise, we would not have the desired regularization. We state the convergence to an approximate stationary point such that $\|\nabla F(W)\|$ is small, building on the following gradient oracle assumption (see, e.g., Ghadimi & Lan (2013); Duchi et al. (2015)).

**Assumption 4.1.** *Given a random seed $z$, let $g_z : \mathbb{R}^d \to \mathbb{R}^d$ be a continuous function that gives an unbiased estimate of the gradient: $\mathbb{E}_z\left[g_z(W)\right] = \nabla f(W)$, for any $W \in \mathbb{R}^d$. Additionally, the variance is bounded in the sense that $\mathbb{E}_z\left[\|g_z(W) - \nabla f(W)\|^2\right] \le \sigma^2$.*

To help understand the above assumption, suppose there is a dataset of size $n$. Then, in SGD, the stochastic gradient would be an unbiased estimate of the gradient of the entire dataset. As for the variance of the gradient estimator, we note that as long as the $\ell_2$ norm of the gradient remains bounded, which will always hold in practice, then the last equation of the above assumption will hold. We now state an upper bound on the convergence rate of Algorithm 1.

**Proposition 4.2.** *Suppose Assumption 4.1 holds. Let $\mathcal{P}$ be a distribution that is symmetric at zero and let $H(\mathcal{P}) = \mathbb{E}[\|U\|^2]$. Let $C$ and $D$ be fixed, positive constants. Let $W_0 \in \mathbb{R}^d$ denote an arbitrary initialization. Suppose $F(W_0) - \min_{W \in \mathbb{R}^d} F(W) \leq D^2$, and $\nabla f$ is $C$-Lipschitz continuous. There exists a fixed learning rate $\eta < C^{-1}$ such that if we run Algorithm 1 with $\eta_i = \eta$ for all $i$ for $T$ steps, the algorithm returns $W_t$ (where $t$ is a random integer between $1, 2, \ldots, T$), such that in expectation over the randomness of $W_t$:*

$$\mathbb{E}\left[\|\nabla F(W_t)\|^2\right] \leq \sqrt{\frac{2CD^2(\sigma^2 + C^2 H(\mathcal{P}))}{kT}} + \frac{2CD^2}{T}. \tag{12}$$

As a remark, existing sharpness-reducing methods such as SAM seem to suffer from oscillation around the local basin (Bartlett et al., 2023). Thus, the convergence behavior of SAM seems challenging to analyze for nonconvex functions. By contrast, our algorithm is amenable to stochastic optimization techniques. Our proof slightly extends the proof of Theorem 2.1, Ghadimi & Lan (2013), to tackle noise injection and other variations. For details, see Appendix B.1.

**Lower bounds:** Next, we construct an example to match the rate of equation (12), essentially showing that this is tight under the same set of assumptions. We use an example from the work of Drori & Shamir (2020). The difference is that we have to deal with the perturbations added to the objective. For $t = 0, 1, \ldots, d - 1$, let $e_t \in \mathbb{R}^d$ be the basis vector in dimension $d$, whose $t$-th coordinate is 1, while the remaining coordinates are all zero. Let $f : \mathbb{R}^d \to \mathbb{R}$ be defined as

$$f(W) = \frac{1}{2G}\langle W, e_0\rangle^2 + \sum_{i=0}^{T-1} h_i(\langle W, e_{i+1}\rangle), \tag{13}$$

where $h_i$ is a piece-wise quadratic function parameterized by $\alpha_i$, defined as follow:

$$h_i(x) = \begin{cases} \frac{C\alpha_i^2}{4} & |x| \leq \alpha_i, \\ -\frac{C\left(|x|-\alpha_i\right)^2}{2} + \frac{C\alpha_i^2}{4} & \alpha_i \leq |x| \leq \frac{3}{2}\alpha_i, \\ \frac{C\left(|x|-2\alpha_i\right)^2}{2} & \frac{3}{2}\alpha_i \leq |x| \leq 2\alpha_i, \\ 0 & 2\alpha_i \leq |x|. \end{cases}$$

One can verify that for each piece above, $\nabla h_i$ is $C$-Lipschitz. As a result, provided that $G \leq C^{-1}$, $\nabla f$ is $C$-Lipschitz, based on the definition of $f$ in equation (13).

The stochastic function $F$ requires setting the perturbation distribution $\mathcal{P}$. We set $\mathcal{P}$ by truncating an isotropic Gaussian $N(0, \sigma^2 \operatorname{Id}_d)$ so that the $i$-th coordinate is at most $2^{-1}\alpha_{i-1}$, for $i = 1, \ldots, T$. Additionally, we set the initialization $W_0$ to satisfy $\langle W_0, e_i\rangle = 0$ for any $i \geq 1$ while $\langle W_0, e_0\rangle \neq 0$. Finally, we choose the gradient oracle to satisfy that the $i$-th step's gradient noise $\xi_i = \langle \xi_i, e_{i+1}\rangle e_{i+1}$, which means that $\xi_i$ is along the direction of the basis vector $e_{i+1}$. In particular, this implies only coordinate $i + 1$ is updated in step $i$, as long as $\langle \xi_i, e_{i+1}\rangle \leq 2^{-1}\alpha_i$. With this construction, we state the lower bound below.

**Theorem 4.3.** *Let the learning rates $\eta_0, \ldots, \eta_{T-1}$ be at most $C^{-1}$. Let $D > 0$ be a fixed value. When either $\sum_{i=0}^{T-1} \eta_i \lesssim \sqrt{kT}$, or $\eta_i = \eta < C^{-1}$ for any epoch $i$, then for the above construction, the following must hold*

$$\min_{1 \leq t \leq T} \mathbb{E}\left[\|\nabla F(W_t)\|^2\right] \geq D\sqrt{\frac{C\sigma^2}{32k \cdot T}}. \tag{14}$$

We remark that the above construction requires $T \leq d$. Notice that this is purely for technical reasons. We briefly illustrate the key steps of the proof. At step $i$, the gradient noise $\xi_i$ plus the perturbation noise is less than $2^{-1}\alpha_i + 2^{-1}\alpha_i = \alpha_i$ at coordinate $i + 1$ (by triangle inequality). Thus, $h_i'(\langle W_t, e_{i+1}\rangle) = 0$, which holds for all prior update steps. This implies

$$\nabla f(W_i) = G^{-1}\langle W_i, e_0\rangle.$$

Recall that $F(W_0) \leq D^2$. This condition imposes how large the $\alpha_i$'s can be. In particular, we will set $\alpha_i = 2\eta_i\sigma/\sqrt{k}$ in the proof. Then, based on the definition of $f(W_0)$,

$$h_i(\langle W_0, e_{i+1}\rangle) = \frac{C\alpha_i^2}{4}, \text{ since } \langle W_0 + U, e_{i+1}\rangle \leq \alpha_i.$$

In Lemma B.3, we then argue that the learning rates in this case must satisfy $\sum_{i=0}^{T-1} \eta_i \leq O(\sqrt{T})$. When the learning rate is fixed and at least $\Omega(T^{-1/2})$, we construct a piece-wise quadratic function (similar to equation (13)), now with a fixed $\alpha$. This is described in Lemma B.4. In this case, the gradient noise grows by $1 - C^{-1}\eta$ up to $T$ steps. We then carefully set $\alpha$ to lower bound the norm of the gradient. Combining these two cases, we conclude the proof of Theorem 4.3. For details, see Appendix B.2. As is typical in lower-bound constructions, our result holds for a specific instance (with a particular learning rate range).

The proof can also be extended to adaptive learning rate schedules. Notice that the above construction holds for arbitrary learning rates defined as a function of previous iterates. Then, we set the width of each function $h_t$, $\alpha_t$, proportional to $\eta_t > 0$, for any $\eta_t$ that may depend on previous iterates, as long as they satisfy the constraint that $\sum_{i=0}^{T-1} \eta_i \leq O(\sqrt{T})$.

**Extension:** We can show a similar lower bound for the momentum update rule. Recall this is defined as

$$M_{i+1} = \mu M_i - \eta_i G_i, \text{ and } W_{i+1} = W_i + M_{i+1}, \tag{15}$$

for $i = 0, 1, \ldots, T - 1$, where $G_i$ is the specific gradient at step $i$. To handle this case, we will need a more fine-grained control on the gradient, so we consider a quadratic function as $f(W) = \frac{C}{2} \|W\|^2$. We leave the result and its proof to Appendix B.3.

**Remark 4.4.** *The novelty of our results lies in analyzing sharpness minimization using techniques from stochastic optimization. This appears to be new, and we hope our work can inspire further studies, such as designing accelerated sharpness minimization methods.*

## 5 Regularization Effect of Hessian Trace in Over-Parameterized Matrix Sensing

Before proceeding, let us give an example of the regularization effect of penalizing the Hessian trace. We consider the matrix sensing problem, whose generalization properties are particularly well-understood in the nonconvex factorization setting (Li et al., 2018). Let there be an unknown, rank-$r$ positive semi-definite matrix $X^\star = U^\star U^{\star\top} \in \mathbb{R}^{d \times d}$. The input consists of a list of $d$ by $d$ Gaussian measurement matrix $A_1, A_2, \ldots, A_n$. The labels are given by $y_i = \langle A_i, X^\star\rangle$, for every $i = 1, 2, \ldots n$. The empirical loss is

$$\hat{L}(W) = \frac{1}{2n} \sum_{i=1}^{n} \left(\langle A_i, WW^\top\rangle - y_i\right)^2, \text{ where } W \in \mathbb{R}^{d \times d}. \tag{16}$$

When the loss reaches near zero (which implies the gradient also reaches near zero), it is known that multiple local minimum solutions exist (Li et al., 2018), and the Hessian becomes

$$\frac{1}{n} \sum_{i=1}^{n} \|A_i W\|_F^2 \approx d \|W\|_F^2 = d \|WW^\top\|_\star.$$

By prior results (Recht et al., 2010), among all $X = WW^\top$ such that $\hat{L}(W) = 0$, $X^\star$ has the lowest nuclear norm. Thus, the regularization placed on $\hat{L}(W)$ is similar to nuclear norm regularization after interpolating the training dataset. We formalize this discussion and state the result below.

**Proposition 5.1.** *In the setting above, for any $W$ that satisfies $\hat{L}(W) = 0$, the following must hold with high probability:*

$$\text{Tr}\left[\nabla^2[\hat{L}(U^\star)]\right] \leq \text{Tr}\left[\nabla^2[\hat{L}(W)]\right] + O(n^{-\frac{1}{2}}). \tag{17}$$

A similar statement holds if the trace operator is replaced by $\lambda_1$ in equation (17). To see this, we look at the quadratic form of the Hessian to find the maximum eigenvalue. Let $u$ be a $d^2$ dimension vector with length equal to one, $\|u\| = 1$. One can derive that:

$$\lambda_1[\nabla^2 \hat{L}(W)] = \max_{u \in \mathbb{R}^{d^2}: \|u\|=1} u^\top \nabla^2 \hat{L}(W) u = \max_{u \in \mathbb{R}^{d^2}: \|u\|=1} \frac{1}{n} \sum_{i=1}^{n} \langle A_i W, u \rangle^2 \geq \frac{1}{d^2 n} \sum_{i=1}^{n} \|A_i W\|_F^2 \,.$$

The last step is by setting $u = d^{-1}\mathbf{1}_{d^2}$, whose length is equal to one. The proof of Proposition 5.1 can be found in Appendix A.2.

**Simulation:** We conduct a numerical simulation to compare algorithmic behaviors. We generate a low-rank matrix $U^\star \in \mathbb{R}^{d \times r}$ from the isotropic Gaussian. We set $d = 100$ and $r = 5$. Then, we test three algorithms: gradient descent (GD), weight-perturbed gradient descent (WP-GD), and Algorithm 1 (NSO). In particular, we will implement the full gradient update rather than using the stochastic updates. We use an initialization $U_0 \in \mathbb{R}^{d \times d}$ where each matrix entry is sampled independently from standard Gaussian $\mathcal{N}(0, 1)$.

Recall that WP-GD and NSO require setting $\sigma$. We choose $\sigma$ between $0.001, 0.002, 0.004, 0.008, 0.0016$. NSO additionally requires setting the number of sampled perturbations $k$. We set $k = 1$ for faster computing. As for the learning rate, we choose a fixed $\eta$ for each run and vary its value between $0.001, 0.0025, 0.005$, and $0.01$. We find that setting $\eta$ as either $0.005$ or $0.01$ would be too large, leading the loss values to explode. Hence, we report the results for setting $\eta$ as $0.0025$ or $0.001$.

Our findings are illustrated in Figure 6.

- We see that all three algorithms can reduce the training MSE to near zero, as shown in Figure 6a. From the trends, we can see that the training loss has fully converged for all cases.

- GD suffers from overfitting to training data, while both WP-GD and NSO can generalize to the validation samples. Moreover, NSO reduces this validation loss further in Figure 6b.

- Finally, we can see in Figure 6c that our algorithm can indeed produce a more accurate estimate of the ground truth matrix $X^\star$, as measured by the Frobenius norm distance between $W_i W_i^\top$ and $X^\star$.

- The results for varying learning rates can be found in the bottom panel from Figure 6d to 6f. The comparative results remain consistent.

## 6 Discussions and Related Work

Using noise injection during neural network training has appeared in very early studies of machine learning research (Hinton & Van Camp, 1993; An, 1996). Graves (2011) test a variety of variational inference approaches with different prior and posterior distributions with recurrent neural networks. Cohen et al. (2019) examine the use of randomized smoothing (with different smoothing distributions) against different $\ell_p$ adversaries for certified robustness. Camuto et al. (2020) propose a layer-wise regularization scheme motivated by adaptation patterns of weights through deeper layers. Yang et al. (2020) show how to turn any classifier that classifies well under Gaussian noise into a new classifier robust to adversarial perturbation under the $\ell_2$ norm. One of the implications of their work is that smoothing with Gaussian noise naturally confers adversarial robustness in the $\ell_2$ norm. Bisla et al. (2022) conduct an extensive empirical study to explore the connection between sharpness and generalization for training neural networks. Orvieto et al. (2023) analyze Taylor's expansion of the stochastic objective after noise injection, examining the induced regularization in various neural network training settings, and find that layer-wise perturbation can improve generalization.

There is also a line of work on Hessian and sharpness in the edge of stability regime during gradient descent dynamics (Cohen et al., 2021). In particular, the edge of stability refers to scenarios where the learning rate goes out of bounds beyond the Lipschitz continuity of a function, which is inversely proportional to the largest eigenvalue of the Hessian matrix. Long & Bartlett (2024) identify the edge of stability regime

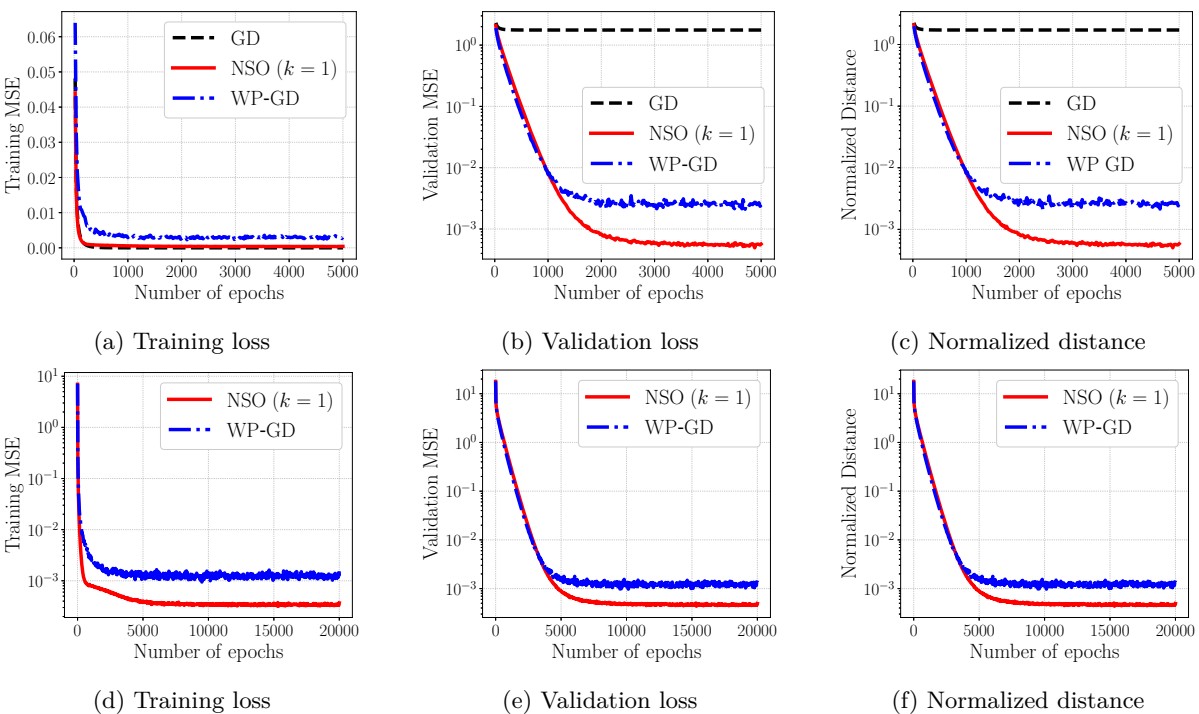

Figure 6: Comparing the training loss, validation loss, and the normalized Frobenius norm distance, i.e., $\frac{\left\|W_i W_i^\top - X^\star\right\|_F^2}{\left\|X^\star\right\|_F^2}$, between GD, our approach (NSO), and weight-perturbed (WP) GD (which computes the full gradient as opposed to the stochastic gradient). For the top panel, the learning rate is fixed at 0.0025 for all the runs. For the bottom panel, the learning rate is set at 0.0001. $\sigma$ is set as 0.008 for WP-GD and NSO. Also, we trained sufficiently long until the loss curves fully converged.

for the SAM algorithm, highlighting the differences of these regimes between SAM and gradient descent. Agarwala & Dauphin (2023) present a detailed study of the gradient dynamics of SAM, documenting various respects of this algorithm. They first analyze the full-batch gradient descent with unnormalized SAM in a quadratic regression model. This analysis suggests that at initialization, full-batch SAM presents limited suppression of the largest eigenvalue of the Hessian matrix. They also show that as the batch size decreases, the regularization of SAM becomes stronger. This work underscores the intricate dynamics of SAM due to its connection to the min-max problem, which is computationally intractable (Daskalakis et al., 2021). Dauphin et al. (2024) provide an in-depth comparison between SAM and weight noise by examining the structure of the Hessian during training. Our results in Section 2.1, which show that weight noise remains ineffective (for fine-tuning), are consistent with the findings of this work. Wu et al. (2020) study the structure of the Hessian and conduct experiments on how the Hessian structure changes based on architecture and the training method.

Randomized smoothing has been studied in stochastic optimization under various contexts, for instance, estimating gradients in zeroth-order optimization (Duchi et al., 2015), and for nonsmooth convex optimization problems (Duchi et al., 2012). In particular, Duchi et al. (2012) analyze the convergence rates of stochastic optimization algorithms and examine a convolution-based smoothing technique for nonsmooth stochastic optimization problems by drawing stochastic gradient samples from the smoothed problem with an appropriate choice of smoothing density. They show that with the ability to issue several queries to the stochastic oracle, the original problem can be solved with faster convergence rates than a simple stochastic oracle. Besides, recent research has investigated the query complexity of finding stationary points of nonconvex functions (Carmon et al., 2020; Arjevani et al., 2023). These results provide a fine-grained characterization of the complexity of iterative methods under different orders of gradient oracles.

The findings from our work suggest several avenues that seem ripe for future work. Can recent advancements in optimization be used to design better noise injection algorithms with faster convergence? Can we better understand the effect of noise injection on the Hessian during training (e.g., in tensor regression where saddle points are known to exist (Li et al., 2020))? In particular, our work highlights the need for more accurate measurements to understand the learning mechanisms of complex models.

## 7 Conclusion and Limitations

This paper examines the regularization and generalization effects of noise-injection methods for training neural networks. The study begins by noting that a straightforward implementation of injecting noise into weight matrices (of a neural network) before computing the gradient in SGD does not perform well in practice. Thus, an alternative, two-point noise injection scheme is proposed and is shown to be effective through extensive experiments. In particular, this new algorithm can be used to regularize the Hessian and improve generalization. The results are tested on fine-tuning, pretraining, and instruction tuning. As a complement, a PAC-Bayes generalization bound is provided to support the rationale of this approach. Finally, this paper presents a detailed convergence analysis of the proposed algorithm.

**Limitations:** In Theorem 2.1, we have shown that the generalization error of a training algorithm can be bounded by the trace of the Hessian of the loss matrix, scaled by the distance of the hypothesis space. Notice that this result applies to both Algorithm 1 (NSO) and the naive noise injection algorithm (WP-SGD). As shown in Figure 3, this result can provide a descriptive measure to explain different algorithms. Since the Hessian measurements can be used on both algorithms, they can only distinguish one from another after taking the measurements from the data. Thus, our generalization theory should be interpreted with this data-dependent lens in mind. We hope future work could work on addressing such limitations, along with designing more principled optimization algorithms for training neural networks.

## Acknowledgements

H. Z. would like to thank Huy Nguyen, Zhiyuan Li, and Guanghui Lan for the discussions and for pointing out several references during various stages of this work. The authors would also like to thank the anonymous reviewers and the action editor for their constructive feedback. We acknowledge financial support from NSF award IIS-2412008.

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

# A    Omitted Proofs from Section 2

We state a few standard notations. Given two matrices $X, Y$ having the same dimension, let $\langle X, Y \rangle = \text{Tr}[X^\top Y]$ denote the matrix inner product of $X$ and $Y$. Let $\|X\|_2$ denote the spectral norm (largest singular value) of $X$, and let $\|X\|_F$ denote the Frobenius norm of $X$. We use the big-O notation $f(x) = O(g(x))$ to indicate that there exists a fixed constant $C$ independent of $x$ such that $f(x) \leq C \cdot g(x)$ for large enough $x$.

## A.1    Proof of the PAC-Bayes Bound

We will use the following PAC-Bayes bound. For reference, see, e.g., Theorem 2, McAllester (2013).

**Theorem A.1.** *Suppose the loss function $\ell(f_W(x), y)$ lies in a bounded range $[0, C]$ given any $x \in \mathcal{X}$ with label $y$. For any $\beta \in (0, 1)$ and $\delta \in (0, 1)$, with probability at least $1 - \delta$, the following holds:*

$$L_{\mathcal{Q}}(W) \leq \frac{1}{\beta} \hat{L}_{\mathcal{Q}}(W) + \frac{C\big(KL(\mathcal{Q}||\mathcal{P}) + \log \frac{1}{\delta}\big)}{2\beta(1 - \beta)n}. \tag{18}$$

This result provides flexibility in setting $\beta$. Our results will set $\beta$ to balance the perturbation error of $\mathcal{Q}$ and the KL divergence between $\mathcal{P}$ and $\mathcal{Q}$. We will need the KL divergence between the prior $\mathcal{P}$ and the posterior $\mathcal{Q}$ in the PAC-Bayesian analysis. This is stated in the following result.

**Proposition A.2.** *Suppose $\mathcal{P} = N(X, \Sigma)$ and $\mathcal{Q} = N(Y, \Sigma)$ are both Gaussian distributions with mean vectors given by $X \in \mathbb{R}^p, Y \in \mathbb{R}^p$, and population covariance matrix $\Sigma \in \mathbb{R}^{p \times p}$. The KL divergence between $\mathcal{P}$ and $\mathcal{Q}$ is equal to*

$$KL(\mathcal{Q}||\mathcal{P}) = \frac{1}{2}(X - Y)^\top \Sigma^{-1}(X - Y).$$

*Specifically, if $\Sigma = \sigma^2 \, \text{Id}_p$, then the above simplifies to*

$$KL(\mathcal{Q}||\mathcal{P}) = \frac{\|X - Y\|_2^2}{2\sigma^2}.$$

We will use Taylor's expansion on the perturbed loss. This is stated precisely as follows.

**Claim A.3.** *Let $f_W$ be twice-differentiable, parameterized by weight vector $W \in \mathbb{R}^p$. Let $U \in \mathbb{R}^p$ be another vector with dimension $p$. For any $W$ and $U$, the following identity holds*

$$\ell(f_{W+U}(x), y) = \ell(f_W(x), y) + U^\top \nabla \ell(f_W(x), y) + U^\top [\nabla^2 \ell(f_W(x), y)]U + R_2(\ell(f_W(x), y)),$$

*where $R_2(\ell(f_W(x), y)))$ is a second-order error term in Taylor's expansion.*

*Proof.* The proof follows by the fact that $\ell \circ f_W$ is twice-differentiable. Let $\eta \in \mathbb{R}^p$ be a vector with the same dimension as $W$ and $U$ from the mean value theorem. There must exist an $\eta$ between $W$ and $U + W$ such that the following equality holds:

$$R_2(\ell(f_W(x), y)) = U^\top \Big(\nabla^2[\ell(f_\eta(x), y)] - \nabla^2[\ell(f_W(x), y)]\Big)U.$$

This completes the proof of the claim. $\qquad \square$

We provide Taylor's expansion of $\ell_{\mathcal{Q}} - \ell$ based on the above.

**Lemma A.4.** *In the setting of Theorem 2.1, suppose each parameter is perturbed by an independent noise drawn from $N(0, \sigma^2)$. Let $\ell_{\mathcal{Q}}(f_W(x), y)$ be the perturbed loss with noise perturbation injection vector on $W$. There exist some fixed value $C_1$ that do not grow with $n$ and $1/\delta$ such that*

$$\left| \ell_{\mathcal{Q}}(f_W(x), y) - \ell(f_W(x), y) - \frac{1}{2}\sigma^2 \text{Tr}\left[\nabla^2[\ell(f_W(x), y)]\right] \right| \leq C_1 \sigma^3.$$

*Proof.* We take the expectation over $U$ for both sides of the equation in Claim A.3. The result becomes

$$\mathbb{E}_U [\ell(f_{W+U}(x), y)] = \mathbb{E}_U \left[ \ell(f_W(x), y) + U^\top \nabla \ell(f_W(x), y) + U^\top \nabla^2 [\ell(f_W(x), y)] U + R_2(\ell(f_W(x), y)) \right].$$

Then, we use the perturbation distribution $\mathcal{Q}$ on $\mathbb{E}_U[\ell(f_{W+U}(x), y)]$, and get

$$\ell_{\mathcal{Q}}(f_W(x), y) = \mathbb{E}_U [\ell(f_W(x), y)] + \mathbb{E}_U \left[ U^\top \nabla \ell(f_W(x), y) \right] + \mathbb{E}_U \left[ U^\top \nabla^2 [\ell(f_W(x), y)] U \right] + \mathbb{E}_U [R_2(\ell(f_W(x), y))].$$

Since $\mathbb{E}[U] = 0$, the first-order term will be zero in expectation. The second-order term becomes equal to

$$\mathbb{E}_U \left[ U^\top [\nabla^2 \ell(f_W(x), y)] U \right] = \sigma^2 \operatorname{Tr} \left[ \nabla^2 [\ell(f_W(x), y)] \right]. \tag{19}$$

The expectation of the error term $R_2(\ell(f_W(x), y))$ be

$$\begin{aligned}
\mathbb{E}_U [R_2(\ell(f_W(x), y))] &= \mathbb{E}_U \left[ U^\top \left( \nabla^2 [\ell(f_\eta(x), y)] - \nabla^2 [\ell(f_W(x), y)] \right) U \right] \\
&\leq \mathbb{E}_U \left[ \|U\|_2^2 \cdot \left\| \nabla^2 [\ell(f_\eta(x), y)] - \nabla^2 [\ell(f_W(x), y)] \right\|_F \right] \\
&\lesssim \mathbb{E}_U \left[ \|U\|_2^2 \cdot C_1 \|U\|_2 \right] \lesssim C_1 \sigma^3.
\end{aligned}$$

Thus, the proof is complete. $\qquad \square$

The last piece we will need is the uniform convergence of the Hessian operator. The result uses the fact that the Hessian matrix is Lipschitz continuous.

**Lemma A.5.** *In the setting of Theorem 2.1, there exist some fixed values $C_2, C_3$ that do not grow with $n$ and $1/\delta$, such that with probability at least $1 - \delta$ for any $\delta > 0$, over the randomness of the $n$ training examples, we have*

$$\left\| \frac{1}{n} \sum_{i=1}^n \nabla^2 [\ell(f_W(x_i), y_i)] - \mathbb{E}_{(x,y) \sim \mathcal{D}} \left[ \nabla^2 [\ell(f_W(x), y)] \right] \right\|_F \leq \frac{C_2 \sqrt{\log(C_3 n/\delta)}}{\sqrt{n}}. \tag{20}$$

The proof will be deferred to Section A.1.2. With these results ready, we will provide proof of the Hessian-based generalization bound.

### A.1.1 Proof of Theorem 2.1

*Proof of Theorem 2.1.* First, we separate the gap of $L(W)$ and $\frac{1}{\beta} \hat{L}(W)$ into three parts:

$$L(W) - \frac{1}{\beta} \hat{L}(W) = L(W) - L_{\mathcal{Q}}(W) + L_{\mathcal{Q}}(W) - \frac{1}{\beta} \hat{L}_{\mathcal{Q}}(W) + \frac{1}{\beta} \hat{L}_{\mathcal{Q}}(W) - \frac{1}{\beta} \hat{L}(W).$$

By Lemma A.4, we can bound the difference between $L(W)$ and $L_{\mathcal{Q}}(W)$ by the Hessian trace plus an error:

$$\begin{aligned}
L(W) - \frac{1}{\beta} \hat{L}(W) \leq &- \mathbb{E}_{(x,y) \sim \mathcal{D}} \left[ \frac{\sigma^2}{2} \operatorname{Tr} \left[ \nabla^2 [\ell(f_W(x), y)] \right] \right] + C_1 \sigma^3 + \left( L_{\mathcal{Q}}(W) - \frac{1}{\beta} \hat{L}_{\mathcal{Q}}(W) \right) \\
&+ \frac{1}{\beta} \left( \frac{1}{n} \sum_{i=1}^n \frac{\sigma^2}{2} \operatorname{Tr} \left[ \nabla^2 [\ell(f_W(x_i), y_i)] \right] + C_1 \sigma^3 \right).
\end{aligned}$$

After re-arranging the terms, we can get the following:

$$\begin{aligned}
L(W) - \frac{1}{\beta} \hat{L}(W) \leq &\underbrace{- \mathbb{E}_{(x,y) \sim \mathcal{D}} \left[ \frac{\sigma^2}{2} \operatorname{Tr} \left[ \nabla^2 [\ell(f_W(x), y)] \right] \right] + \frac{1}{n\beta} \sum_{i=1}^n \frac{\sigma^2}{2} \operatorname{Tr} \left[ \nabla^2 [\ell(f_W(x_i), y_i)] \right]}_{E_1} \\
&+ \frac{1+\beta}{\beta} C_1 \sigma^3 + \underbrace{L_{\mathcal{Q}}(W) - \frac{1}{\beta} \hat{L}_{\mathcal{Q}}(W)}_{E_2}.
\end{aligned} \tag{21}$$

We will examine $E_1$ by separating it into two parts:

$$E_1 = \frac{1}{\beta}\left(\frac{1}{n}\sum_{i=1}^{n}\frac{\sigma^2}{2}\operatorname{Tr}\left[\nabla^2[\ell(f_{\hat{W}}(x_i),y_i)]\right] - \mathop{\mathbb{E}}_{(x,y)\sim\mathcal{D}}\left[\frac{\sigma^2}{2}\operatorname{Tr}\left[\nabla^2[\ell(f_W(x),y)]\right]\right]\right) \tag{22}$$

$$+ \frac{1-\beta}{\beta}\frac{\sigma^2}{2}\mathop{\mathbb{E}}_{(x,y)\sim\mathcal{D}}\left[\operatorname{Tr}\left[\nabla^2\ell(f_W(x),y)\right]\right]. \tag{23}$$

We can use the uniform convergence result of Lemma A.5 to bound equation (22), leading to:

$$\frac{\sigma^2}{2\beta}\left(\frac{1}{n}\sum_{i=1}^{n}\operatorname{Tr}\left[\nabla^2\ell(f_W(x_i),y_i)\right] - \mathop{\mathbb{E}}_{(x,y)\sim\mathcal{D}}\left[\operatorname{Tr}\left[\nabla^2\ell(f_W(x),y))\right]\right]\right)$$

$$\leq\frac{\sigma^2}{2\beta}\cdot\sqrt{p}\cdot\left\|\frac{1}{n}\sum_{i=1}^{n}\nabla^2[\ell(f_W(x_i),y_i)] - \mathop{\mathbb{E}}_{(x,y)\sim\mathcal{D}}\left[\nabla^2[\ell(f_W(x),y)]\right]\right\|_F \qquad \text{(by Cauchy-Schwarz)}$$

$$\leq\frac{\sigma^2\sqrt{p}\cdot C_2\sqrt{\log(C_3 n/\delta)}}{2\beta\sqrt{n}}. \tag{24}$$

In particular, the second step also uses the fact that the Hessian matrix is a symmetric $p$ by $p$ matrix. As for equation (23), we recall that

$$\alpha := \max_{(x,y)\sim\mathcal{D}}\operatorname{Tr}\left[\nabla^2\ell(f_W(x),y)\right].$$

Combined with equation (24), we have shown that

$$E_1 \leq \frac{\sigma^2\sqrt{p}\cdot C_2\sqrt{\log(C_3 n/\delta)}}{2\beta\sqrt{n}} + \frac{1-\beta}{\beta}\frac{\sigma^2}{2}\cdot\alpha. \tag{25}$$

As for $E_2$, we will use the PAC-Bayes bound of Theorem A.1. In particular, we set the prior distribution $\mathcal{P}$ as the distribution of $U$ and the posterior distribution $\mathcal{Q}$ as the distribution of $W + U$. Thus,

$$E_2 \leq \frac{C\left(KL(\mathcal{Q}\|\mathcal{P}) + \log\frac{1}{\delta}\right)}{2\beta(1-\beta)n} \leq \frac{C\left(\frac{\|W\|_2^2}{2\sigma^2} + \log\frac{1}{\delta}\right)}{2\beta(1-\beta)n} \leq \frac{C\left(\frac{r^2}{2\sigma^2} + \log\delta^{-1}\right)}{2\beta(1-\beta)n}. \tag{26}$$

The last step is because $\|W\|_2 \leq r$ by the assumption of the hypothesis space. Combining equations (21), (25), (26), we claim that with probability at least $1 - 2\delta$, the following must be true:

$$L(W) - \frac{1}{\beta}\hat{L}(W) \leq \frac{\sigma^2\sqrt{p}\cdot C_2\sqrt{\log(C_3 n/\delta)}}{2\beta\sqrt{n}} + \frac{1-\beta}{\beta}\frac{\sigma^2}{2}\alpha + \frac{1+\beta}{\beta}C_1\sigma^3 + \frac{C\left(\frac{r^2}{2\sigma^2} + \log\frac{1}{\delta}\right)}{2\beta(1-\beta)n}. \tag{27}$$

Thus, we will now choose $\sigma$ and $\beta \in (0,1)$ to minimize the term above. In particular, we will set $\sigma$ as

$$\sigma^2 = \frac{r}{1-\beta}\sqrt{\frac{C}{\alpha n}}. \tag{28}$$

By plugging in $\sigma$ to equation (27) and re-arranging terms, the gap between $L(W)$ and $\frac{\hat{L}(W)}{\beta}$ becomes:

$$L(W) - \frac{1}{\beta}\hat{L}(W) \leq \frac{1}{\beta}\sqrt{\frac{C\alpha r^2}{n}} + \frac{C_2\sqrt{2p\log(C_3 n/\delta)}}{2\beta\sqrt{n}}\sigma^2 + \frac{1+\beta}{\beta}C_1\sigma^3 + \frac{C}{2\beta(1-\beta)n}\log\frac{1}{\delta}.$$

Let $\beta$ be a fixed value close to 1 and independent of $N$ and $\delta^{-1}$, and let $\epsilon = (1-\beta)/\beta$. We get

$$L(W) \leq (1+\epsilon)\hat{L}(W) + (1+\epsilon)\sqrt{\frac{C\alpha r^2}{n}} + \xi, \text{ where}$$

$$\xi = \frac{C_2\sqrt{2p\log(C_3 n/\delta)}}{2\beta\sqrt{n}}\sigma^2 + \left(1 + \frac{1}{\beta}\right)C_1\sigma^3 + \frac{C}{2\beta(1-\beta)n}\log\frac{1}{\delta}.$$

Notice that $\xi$ is of order $O(n^{-\frac{3}{4}} + n^{-\frac{3}{4}} + \log(\delta^{-1})n^{-1}) \leq O(\log(\delta^{-1})n^{-\frac{3}{4}})$. Therefore, we have finished the proof of equation (6). $\qquad\square$

**Remark A.6.** *When $f$ is strongly convex, the lowest eigenvalue of the Hessian is bounded from below. Once the algorithm reaches the global minimizer, our result from Theorem 6 can be used to provide a generalization bound based on the trace of the Hessian. Notice that the noise injection will add some bias to this minimizer, leading to a sub-optimal empirical loss. To remedy this issue, one can place the regularization of Hessian as a constraint, similar to how $\ell_2$-regularization can be implemented as a constraint.*

### A.1.2   Proof of Lemma A.5

In this section, we provide the proof of Lemma A.5, which shows the uniform convergence of the loss Hessian.

*Proof of Lemma A.5.* Let $C$, $\epsilon > 0$, and let $S = \{W \in \mathbb{R}^p : \|W\|_2 \le C\}$. There exists an $\epsilon$-cover of $S$ with respect to the $\ell_2$-norm at most $\max\left(\left(\frac{3C}{\epsilon}\right)^p, 1\right)$ elements; see, e.g., Example 5.8 (Wainwright, 2019). Let $T \subseteq S$ denote the set of this cover. Recall that the Hessian $\nabla^2[\ell(f_W(x), y)]$ is $C_1$-Lipschitz for all $(W + U) \in S, W \in S$. Then we have

$$\left\|\nabla^2[\ell(f_{W+U}(x), y)] - \nabla^2[\ell(f_W(x), y)]\right\|_F \le C_1 \|U\|_2.$$

For parameters $\delta, \epsilon > 0$, let $\mathcal{N}$ be the $\epsilon$-cover of $S$ with respect to the $\ell_2$-norm. Define the event

$$E = \left\{\forall W \in T, \left\|\frac{1}{n}\sum_{i=1}^n \nabla^2[\ell(f_W(x_i), y_i)] - \mathop{\mathbb{E}}_{(x,y)\sim\mathcal{D}}\left[\nabla^2[\ell(f_W(x), y)]\right]\right\|_F \le \delta\right\}.$$

By the matrix Bernstein inequality, we have

$$\Pr[E] \ge 1 - 4 \cdot |\mathcal{N}| \cdot p \cdot \exp\left(-\frac{n\delta^2}{2\alpha^2}\right).$$

Next, for any $W \in S$, we can pick some $W + U \in T$ such that $\|U\|_2 \le \epsilon$. We have

$$\left\|\mathop{\mathbb{E}}_{(x,y)\sim\mathcal{D}}\left[\nabla^2[\ell(f_{W+U}(x), y)]\right] - \mathop{\mathbb{E}}_{(x,y)\sim\mathcal{D}}\left[\nabla^2[\ell(f_W(x), y)]\right]\right\|_F \le C_1 \|U\|_2 \le C_1\epsilon$$

$$\left\|\frac{1}{n}\sum_{j=1}^n \nabla^2[\ell(f_{W+U}(x_j), y_j)] - \frac{1}{n}\sum_{j=1}^n \nabla^2[\ell(f_W(x_j), y_j)]\right\|_F \le C_1 \|U\|_2 \le C_1\epsilon.$$

Therefore, for any $W \in S$, we obtain:

$$\left\|\frac{1}{n}\sum_{j=1}^n \nabla^2[\ell(f_W(x_j), y_j)] - \mathop{\mathbb{E}}_{(x,y)\sim\mathcal{D}}\left[\nabla^2[\ell(f_W(x), y)]\right]\right\|_F \le 2C_1\epsilon + \delta.$$

We will also set the value of $\delta$ and $\epsilon$. First, set $\epsilon = \delta/(2C_1)$ so that conditional on $E$,

$$\left\|\frac{1}{n}\sum_{j=1}^n \nabla^2[\ell(f_W(x_j), y_j)] - \mathop{\mathbb{E}}_{(x,y)\sim\mathcal{D}}\left[\nabla^2[\ell(f_W(x), y)]\right]\right\|_F \le 2\delta.$$

The event $E$ happens with a probability of at least:

$$1 - 4|T|p \cdot \exp\left(-\frac{n\delta^2}{2\alpha^2}\right) = 1 - 4p \cdot \exp\left(\log|T| - \frac{n\delta^2}{2\alpha^2}\right).$$

We have $\log|T| \le p\log(3B/\epsilon) = p\log(6CC_1/\delta)$. If we set

$$\delta = \sqrt{\frac{4p\alpha^2 \log(3\tau CC_1 n/\alpha)}{n}}$$

so that $\log(3\tau CC_1 n/\alpha) \geq 1$ (because $n \geq \frac{e\alpha}{3C_1}$ and $\tau \geq 1$), then we get

$$
\begin{aligned}
p\log(6CC_1/\delta) - n\delta^2/(2\alpha^2) =& p\log\left(\frac{6CC_1\sqrt{n}}{\sqrt{4p\alpha^2\log(3\tau CC_1 n/\alpha)}}\right) - 2p\log\left(3\tau CC_1 n/\alpha\right) \\
=& p\log\left(\frac{3CC_1\sqrt{n}}{\alpha\sqrt{p\log(3\tau CC_1 n/\alpha)}}\right) - 2p\log\left(3\tau CC_1 n/\alpha\right) \\
\leq& p\log\left(3\tau CC_1 n/\alpha\right) - 2p\log\left(3\tau CC_1 n/\alpha\right) \qquad (\tau \geq 1, \log(3\tau CC_1 n/\alpha) \geq 1) \\
=& -p\log\left(3\tau CC_1 n/\alpha\right) \leq -p\log(e\tau). \qquad\qquad\qquad (3CC_1 n/\alpha \geq e)
\end{aligned}
$$

Therefore, with a probability greater than

$$
1 - 4|\mathcal{N}|p \cdot \exp(-n\delta^2/(2\alpha^2)) \geq 1 - 4p(e\tau)^{-p},
$$

the following estimate holds:

$$
\left\| \frac{1}{n}\sum_{j=1}^n \nabla^2[\ell(f_W(x_j), y_j)] - \mathop{\mathbb{E}}_{(x,y)\sim\mathcal{D}}\left[\nabla^2[\ell(f_W(x), y)]\right] \right\|_F \leq \sqrt{\frac{16p\alpha^2\log(3\tau CC_1 n/\alpha)}{n}}.
$$

Denote $\delta' = 4p(e\tau)^{-p}$, $C_2 = 4\alpha\sqrt{p}$, and $C_3 = 12pCC_1/(e\alpha)$. With probability greater than $1 - \delta'$, the final result is:

$$
\left\| \frac{1}{n}\sum_{i=1}^n \nabla^2[\ell(f_W(x_i), y_i)] - \mathop{\mathbb{E}}_{(x,y)\sim\mathcal{D}}\left[\nabla^2[\ell(f_W(x), y)]\right] \right\|_F \leq C_2\sqrt{\frac{\log(C_3 n/\delta')}{n}}.
$$

This completes the proof of Lemma A.5. $\qquad\square$

## A.2 Proof of Proposition 5.1

*Proof of Proposition 5.1.* We can calculate the gradient as

$$
\nabla\hat{L}(W) = \frac{1}{n}\sum_{i=1}^n (\langle A_i, WW^\top\rangle - y_i)A_iW. \tag{29}
$$

For a particular entry $W_{j,k}$ of $W$, for any $1 \leq j, k \leq d$, the derivative of the gradient with respect to $W_{j,k}$ is

$$
\frac{1}{n}\sum_{i=1}^n \left( [A_iW]_{j,k}A_iW + \left(\langle A_i, WW^\top\rangle - y_i\right)\frac{\partial(A_iW)}{\partial W_{j,k}}\right). \tag{30}
$$

When $\hat{L}(W)$ is zero, the second term of equation (30) above must be zero, because $\langle A_i, WW^\top\rangle$ is equal to $y_i$, for any $i = 1, \ldots, n$.

We use the assumption that $A_i$ is a random Gaussian matrix, in which every entry is drawn from a normal distribution with mean zero and variance one. Notice that the expectation of $\|A_iW\|_F^2$ satisfies:

$$
\mathbb{E}\left[\|A_iW\|_F^2\right] = \mathbb{E}\left[\mathrm{Tr}\left[W^\top A_i^\top A_i W\right]\right] = \mathrm{Tr}\left[W^\top (d\cdot\mathrm{Id}_{d\times d})W^\top\right] = d\cdot\mathrm{Tr}\left[W^\top W\right] = d\|W\|_F^2.
$$

Thus, by concentration inequality for $\chi^2$ random variables (e.g., Wainwright (2019, equation (2.19))), the following holds for any $0 < \epsilon < 1$,

$$
\Pr\left[\left|\frac{1}{n}\sum_{i=1}^n \|A_iW\|_F^2 - d\|W\|_F^2\right| \geq \epsilon d\|W\|_F^2\right] \leq 2\exp\left(-\frac{n\epsilon^2}{8}\right). \tag{31}
$$

This implies that $\epsilon$ must be smaller than $O(n^{-1/2})$ with high probability. As a result, the average of $\|A_iW\|_F^2$ must be $d\|W\|_F^2$ plus some deviation error that scales with $n^{-1/2}$ times the expectation.

By Theorem 3.2, Recht et al. (2010), the minimum Frobenius norm ($\|W\|_F^2$) solution that satisfies $\hat{L}(W) = 0$ (for Gaussian random matrices) is precisely $U^\star$. Thus, we conclude that equation (17) holds. $\qquad\square$

## B    Omitted Proofs from Section 4

### B.1    Proof of Proposition 4.2

Recall that each iteration involves two sources of randomness stemming from $g_z$ and $\{U_i^{(j)}\}_{j=1}^k$, respectively. Let us define

$$
\delta_i = \frac{1}{2k} \sum_{j=1}^k \left( \nabla f\big(W_i + U_i^{(j)}\big) + \nabla f\big(W_i - U_i^{(j)}\big) \right) - \nabla F(W_i),
$$

$$
\xi_i = \frac{1}{2k} \sum_{j=1}^k \left( G_i^{(j)} - \nabla f\big(W_i + U_i^{(j)}\big) - \nabla f\big(W_i - U_i^{(j)}\big) \right),
$$

for $i = 0, \ldots, T-1$. One can see that both $\delta_i$ and $\xi_i$ have mean zero. The former is by the symmetry of $\mathcal{P}$. The latter is because $g_z$ is unbiased under Assumption 4.1. The following result gives their variance.

**Lemma B.1.** *In the setting of Proposition 4.2, for any $i = 1, \ldots, T$, we have*

$$
\mathbb{E}\left[\|\xi_i\|^2\right] \leq \frac{\sigma^2}{k} \quad and \quad \mathbb{E}\left[\|\delta_i\|^2\right] \leq \frac{C^2 H(\mathcal{P})}{k}. \tag{32}
$$

The last step uses smoothness to show that $\|\nabla F(W_t)\|$ keeps reducing.

*Proof.* Let us bound the variance of $\delta_i$ and $\xi_i$ for $i = 0, 1, \ldots, T-1$. First, we see that

$$
\mathbb{E}_{U_i^1, \ldots, U_i^k}\left[\|\delta_i\|^2\right] = \mathbb{E}_{U_i^1, \ldots, U_i^k}\left[\left\| \frac{1}{2k} \sum_{j=1}^k \left( \nabla f(W_i + U_i^j) + \nabla f(W_i - U_i^j) - 2\nabla F(W_i) \right) \right\|^2\right]
$$

$$
= \frac{1}{k^2} \sum_{j=1}^k \mathbb{E}_{U_i^j}\left[\left\| \frac{1}{2} \left( \nabla f(W_i + U_i^j) + \nabla f(W_i - U_i^j) - 2\nabla F(W_i) \right) \right\|^2\right] \tag{33}
$$

$$
= \frac{1}{k} \mathbb{E}_{U_i^1}\left[\left\| \frac{1}{2} \left( \nabla f(W_i + U_i^1) + \nabla f(W_i - U_i^1) \right) - \nabla F(W_i) \right\|^2\right] \tag{34}
$$

where in the second line we use that $U_i^{j_1}$ and $U_i^{j_2}$ are independent when $j_1 \neq j_2$, in the last line we use fact that $U_i^1, \ldots, U_i^k$ are identically distributed. In the second step, we use the fact that for two independent random variables $U, V$, and any continuous functions $h(U), g(V)$, $h(U)$ and $g(V)$ are still independent (recall that $f$ is continuous since it is twice-differentiable). We include a short proof of this fact for completeness. If $U$ and $V$ are independent, we have $\Pr[U \in A, V \in B] = \Pr[U \in A] \cdot \Pr[V \in B]$, for any $A, B \in \mathrm{Borel}(\mathbb{R})$. Thus, if $h$ and $g$ are continuous functions, we obtain

$$
\Pr[h(U) \in A, g(V) \in B] = \Pr[U \in h^{-1}(A), V \in g^{-1}(B)]
$$
$$
= \Pr[U \in h^{-1}(A)] \cdot \Pr[V \in g^{-1}(B)] = \Pr[h(U) \in A] \cdot \Pr[g(V) \in B].
$$

Thus, we have shown that

$$
\mathbb{E}\left[\|\delta_i\|^2\right] = \frac{1}{k} \mathbb{E}_{U \sim \mathcal{P}}\left[\left\| \frac{1}{2} \left( \nabla f(W_i + U) + f(W_i - U) \right) - \nabla F(W_i) \right\|^2\right]. \tag{35}
$$

Next, we deal with the variance of the two-point stochastic gradient. We will show that

$$
\mathbb{E}_U\left[\left\| \frac{1}{2} \left( \nabla f(W + U) + \nabla f(W - U) \right) - \nabla F(W) \right\|^2\right] \leq C^2 H(\mathcal{P}). \tag{36}
$$

We mainly use the Lipschitz continuity of the gradient of $F$. The left-hand side of equation (36) is equal to

$$
\mathbb{E}_{U}\left[\left\|\frac{1}{2}\left(\nabla f(W+U)-\nabla F(W)\right)+\frac{1}{2}\left(\nabla f(W-U)-\nabla F(W)\right)\right\|^2\right]
$$

$$
\leq \mathbb{E}_{U}\left[\frac{1}{2}\left\|\nabla f(W+U)-\nabla F(W)\right\|^2+\frac{1}{2}\left\|\nabla f(W-U)-\nabla F(W)\right\|^2\right] \qquad \text{(by Cauchy-Schwartz)}
$$

$$
=\frac{1}{2}\mathbb{E}_{U}\left[\left\|\nabla f(W+U)-\nabla F(W)\right\|^2\right] \qquad \text{(by symmetry of } \mathcal{P} \text{ since it has mean zero)}
$$

$$
=\frac{1}{2}\mathbb{E}_{U}\left[\left\|\mathbb{E}_{U'\sim\mathcal{P}}[\nabla f(W+U)-\nabla f(W+U')]\right\|^2\right] \leq \frac{1}{2}\mathbb{E}_{U}\left[\mathbb{E}_{U'\sim\mathcal{P}}\left[\left\|\nabla f(W+U)-\nabla f(W+U')\right\|^2\right]\right]
$$

$$
\leq \frac{1}{2}\mathbb{E}_{U,U'}\left[C^2\left\|U-U'\right\|^2\right]=\frac{1}{2}C^2\mathbb{E}_{U,U'}\left[\left\|U\right\|^2+\left\|U'\right\|^2\right]=C^2 H(\mathcal{P}) \qquad \text{(by equation (38))}
$$

As for the variance of $\xi_i$, we note that $U_i^{(1)},\ldots,U_i^{(j)}$ are all independent from each other. Therefore,

$$
\mathbb{E}_{\left\{U_i^{(j)},z_i^{(j)}\right\}_{j=1}^{k}}\left[\left\|\xi_i\right\|^2\right]=\frac{1}{4k}\mathbb{E}_{U,z}\left[\left\|g_z(W+U)-\nabla f(W+U)+g_z(W-U)-f(W-U)\right\|^2\right]
$$

$$
\leq \frac{1}{2k}\mathbb{E}_{U,z}\left[\left\|g_z(W+U)-\nabla f(W+U)\right\|^2+\left\|g_z(W-U)-\nabla f(W-U)\right\|^2\right]\leq \frac{\sigma^2}{k}.
$$

The first step uses the fact that both $g_z(\cdot)$ and $f(\cdot)$ are continuous functions The second step above uses Cauchy-Schwartz inequality. The last step uses the variance bound of $g_z(\cdot)$, Thus, the proof is finished. $\square$

In the next step, we use a result from Theorem 2.1, Ghadimi & Lan (2013). Our proof follows from their work, but we deal with some extra technical details related to the noise injection.

**Lemma B.2** (Slightly adapted from Theorem 2.1, Ghadimi & Lan (2013))**.** *In the setting of Proposition 4.2, for any $\eta_0,\cdots,\eta_{T-1}$ less than $C^{-1}$ and a random variable according to a distribution $\Pr[t=j]=\frac{\eta_j}{\sum_{i=0}^{T-1}\eta_i}$, for any $j=0,\ldots,T-1$, the following holds:*

$$
\mathbb{E}\left[\left\|\nabla F(W_t)\right\|^2\right]\leq \frac{2C}{\sum_{i=0}^{T-1}\eta_i}D^2+\frac{C\sum_{i=0}^{T-1}\eta_i^2\left(\mathbb{E}\left[\left\|\delta_i\right\|^2\right]+\mathbb{E}\left[\left\|\xi_i\right\|^2\right]\right)}{\sum_{i=0}^{T-1}\eta_i}. \tag{37}
$$

*Proof.* First, let us show that $\nabla F$ is $C$-Lipschitz continuous. To see this, we apply the Lipschitz condition of the gradient inside the expectation of $F(W)$. For any $W_1,W_2\in\mathbb{R}^d$, by definition,

$$
\left\|\nabla F(W_1)-\nabla F(W_2)\right\|=\left\|\nabla\mathbb{E}_{U\sim\mathcal{P}}[f(W_1+U)]-\nabla\mathbb{E}_{U\sim\mathcal{P}}[f(W_2+U)]\right\|
$$

$$
=\left\|\mathbb{E}_{U\sim\mathcal{P}}[\nabla f(W_1+U)-\nabla f(W_2+U)]\right\|
$$

$$
\leq \mathbb{E}_{U\sim\mathcal{P}}[\left\|\nabla f(W_1+U)-\nabla f(W_2+U)\right\|]\leq C\left\|W_1-W_2\right\|.
$$

Since $\nabla F(W)$ is $C$-Lipschitz continuous, we have the following domination inequality:

$$
\left|F(W_2)-F(W_1)-\langle\nabla F(W_1),W_2-W_1\rangle\right|\leq\frac{C}{2}\left\|W_2-W_1\right\|^2. \tag{38}
$$

Based on the above inequality, we have

$$
\begin{aligned}
F(W_{i+1}) \leq & F(W_i) + \langle \nabla F(W_i), W_{i+1} - W_i \rangle + \frac{C}{2}\eta_i^2 \left\| \frac{1}{2}\Big( \nabla f(W_i + U_i) + \nabla f(W_i - U_i) \Big) + \xi_i \right\|^2 \\
= & F(W_i) - \eta_i \langle \nabla F(W_i), \delta_i + \xi_i + \nabla F(W_i) \rangle + \frac{C\eta_i^2}{2}\|\delta_i + \xi_i + \nabla F(W_i)\|^2 \\
= & F(W_i) - \Big(\eta_i - \frac{C\eta_i^2}{2}\Big)\|\nabla F(W_i)\|^2 - \Big(\eta_i - C\eta_i^2\Big)\langle \nabla F(W_i), \delta_i + \xi_i \rangle + \frac{C\eta_i^2}{2}\|\delta_i + \xi_i\|^2 .
\end{aligned}
$$

Summing up the above inequalities for $i = 0, 1, \ldots, T-1$, we obtain

$$
\begin{aligned}
\sum_{i=0}^{T-1} F(W_{i+1}) \leq & \sum_{i=0}^{T-1} F(W_i) - \sum_{i=0}^{T-1}\Big(\eta_i - \frac{C\eta_i^2}{2}\Big)\|\nabla F(W_i)\|^2 \\
& - \sum_{i=0}^{T-1}\Big(\eta_i - C\eta_i^2\Big)\langle \nabla F(W_i), \delta_i + \xi_i \rangle + \sum_{i=0}^{T-1}\frac{C\eta_i^2}{2}\|\delta_i + \xi_i\|^2 ,
\end{aligned}
$$

which implies that

$$
\begin{aligned}
\sum_{i=0}^{T-1}\Big(\eta_i - \frac{C\eta_i^2}{2}\Big)\|\nabla F(W_i)\|^2 \leq & F(W_0) - F(W_T) - \sum_{i=0}^{T-1}\Big(\eta_i - C\eta_i^2\Big)\langle \nabla F(W_i), \delta_i + \xi_i \rangle + \frac{C}{2}\sum_{i=0}^{T-1}\eta_i^2\|\delta_i + \xi_i\|^2 \\
\leq & D^2 - \sum_{i=0}^{T-1}\Big(\eta_i - C\eta_i^2\Big)\langle \nabla F(W_i), \delta_i + \xi_i \rangle + \frac{C}{2}\sum_{i=0}^{T-1}\eta_i^2\|\delta_i + \xi_i\|^2 . \quad (39)
\end{aligned}
$$

where in the last step, we use the fact that

$$
F(W_0) - F(W_T) \leq F(W_0) - \min_{W \in \mathbb{R}^d} F(W) \leq D^2.
$$

For any $t = 0, 1, \ldots, T-1$, notice that as long as $0 < \eta_t \leq \frac{1}{C}$, then $\eta_t \leq 2\eta_t - C\eta_t^2$. Hence, we have

$$
\frac{1}{2}\sum_{t=0}^{T-1}\eta_t \|\nabla F(W_t)\|^2 \leq \sum_{t=0}^{T-1}\Big(\eta_t - \frac{C\eta_t^2}{2}\Big)\|\nabla F(W_t)\|^2 ,
$$

which implies that

$$
\frac{1}{2}\sum_{i=0}^{T-1}\eta_i \|\nabla F(W_i)\|^2 \leq D^2 - \sum_{i=0}^{T-1}\Big(\eta_i - C\eta_i^2\Big)\langle \nabla F(W_i), \delta_i + \xi_i \rangle + \frac{C}{2}\sum_{i=0}^{T-1}\eta_i^2\|\delta_i + \xi_i\|^2 . \quad (40)
$$

Additionally, since $U_t$ is drawn from a distribution with mean zero. Hence, by symmetry, we get that

$$
\mathbb{E}_{U_t}[\delta_t] = \frac{1}{2}\mathbb{E}_{U_t}[\nabla f(W_t - U_t) - \nabla f(W_t + U_t)] = 0. \quad (41)
$$

Thus, if we take the expectation over $U_0, U_1, \ldots, U_{T-1}, \xi_0, \xi_1, \ldots, \xi_{T-1}$, then $\mathbb{E}[\langle \nabla F(W_i), \delta_i + \xi_i \rangle] = 0$. Recall that $t$ is a random variable whose probability mass is specified in Lemma B.2. We can write equation (40) equivalently as (below, we take expectation over all the random variables along the update since $W_t$ is a function of the previous gradient updates, for each $t = 0, 1, \ldots, T-1$, recalling that $\Pr[t = i] = \frac{\eta_i}{\sum_{j=0}^{T-1}\eta_j}$)

$$
\begin{aligned}
\mathop{\mathbb{E}}_{t;\, U_0,\ldots,U_{T-1},\xi_0,\xi_1,\ldots,\xi_{T-1}}\left[\|\nabla F(W_t)\|^2\right] = & \frac{\sum_{i=0}^{T-1}\eta_i \,\mathbb{E}\left[\|\nabla F(W_i)\|^2\right]}{\sum_{i=0}^{T-1}\eta_i} \\
\leq & \frac{2D^2 + C\sum_{i=0}^{T-1}\eta_i^2 \,\mathbb{E}\left[\|\delta_i + \xi_i\|^2\right]}{\sum_{i=0}^{T-1}\eta_i} \\
= & \frac{2D^2 + C\sum_{i=0}^{T-1}\eta_i^2 \big(\mathbb{E}\left[\|\delta_i\|^2\right] + \mathbb{E}\left[\|\xi_i\|^2\right]\big)}{\sum_{i=0}^{T-1}\eta_i} .
\end{aligned}
$$

where we use the fact that $\delta_i$ and $\xi_i$ are independent for any $i$. Hence, equation (37) is proved. $\qquad\square$

Based on the above result, we now finish the proof of Proposition 4.2.

*Proof of Proposition 4.2.* Let the step sizes be a fixed $\eta$ for all epochs. Thus, equation (37) becomes

$$\mathbb{E}\left[\|\nabla F(W_t)\|^2\right] \leq \frac{2}{T\eta}D^2 + \frac{C\eta}{T}\sum_{i=0}^{T-1}\left(\mathbb{E}\left[\|\delta_i\|^2\right] + \mathbb{E}\left[\|\xi_i\|^2\right]\right). \tag{42}$$

By Lemma B.1,

$$\sum_{i=0}^{T-1}\left(\mathbb{E}\left[\|\delta_i\|^2\right] + \mathbb{E}\left[\|\xi_i\|^2\right]\right) \leq T \cdot \frac{\sigma^2 + C^2 H(\mathcal{P})}{k}. \tag{43}$$

For simplicity, let us denote $\Delta = \frac{\sigma^2 + C^2 H(\mathcal{P})}{k}$. The proof is divided into two cases.

**Case 1: $\Delta$ is large.** More precisely, suppose that $\Delta \geq 2CD^2/T$. Then, minimizing over $\eta$ above leads us to the following upper bound on the right-hand side of equation (42):

$$\sqrt{\frac{2CD^2\Delta}{T}}, \tag{44}$$

which is obtained by setting $\eta = \sqrt{\frac{2D^2}{C\Delta T}}$. One can verify that this step size is less than $\frac{1}{C}$ since $\Delta$ is at least $2CD^2$. Thus, we conclude that equation (42) must be less than

$$\sqrt{\frac{2CD^2\Delta}{T}} = \sqrt{\frac{2CD^2(\sigma^2 + C^2 H(\mathcal{P})))}{kT}}. \tag{45}$$

**Case 2: $\Delta$ is small.** In this case, suppose $\Delta < 2CD^2/T$. Then, the right-hand side of equation (42) must be less than

$$\frac{2D^2}{T\eta} + \frac{2C^2D^2\eta}{T} \leq \frac{2CD^2}{T}. \tag{46}$$

Thus, combining equations (45) and (46), we have completed the proof of equation (12).

$\square$

## B.2 Proof of Theorem 4.3

Recall our construction from Section 4 as follows. Let $e_t$ be the basis vector for the $t$-th dimension, for $t = 0, 1, \ldots, T-1$. Define $f(W)$ as

$$f(W) = \frac{1}{2G}\langle W, e_0\rangle^2 + \sum_{i=0}^{T-1} h_i(\langle W, e_{i+1}\rangle),$$

where $h_i$ a quadratic function parameterized by $\alpha_i$, defined as follow:

$$h_i(x) = \begin{cases} \frac{C\alpha_i^2}{4} & |x| \leq \alpha_i \\ -\frac{C(|x|-\alpha_i)^2}{2} + \frac{C\alpha_i^2}{4} & \alpha_i \leq |x| \leq \frac{3}{2}\alpha_i \\ \frac{C(|x|-2\alpha_i)^2}{2} & \frac{3}{2}\alpha_i \leq |x| \leq 2\alpha_i \\ 0 & 2\alpha_i \leq |x|. \end{cases}$$

For technical reasons, we define a truncated perturbation distribution $\mathcal{P}$. Given a sample $U$ from a $d$-dimensional isotropic Gaussian $N(0, \mathrm{Id}_d)$, we truncate the $i$-th coordinate of $U$ so that $\tilde{U}_i = \min(U_i, a_i)$, for some fixed $a_i > 0$ that we will specify below, for all $i = 0, 1, \ldots, d-1$. Let $\mathcal{P}$ denote the distribution of $\tilde{U}$. The proof of Theorem 4.3 is divided into two cases. First, we examine the case when the averaged learning rate is $O(T^{-1/2})$.

**Lemma B.3.** *In the setting of Theorem 4.3, suppose the learning rates satisfy that $\sum_{i=0}^{T-1} \eta_i \leq \sqrt{\frac{D^2 kT}{2\sigma^2 C}}$, consider the function $f(W)$ constructed in equation (13), we have*

$$\min_{1 \leq t \leq T} \mathbb{E}\left[\|\nabla F(W_t)\|^2\right] \geq D\sqrt{\frac{C\sigma^2}{32kT}}.$$

*Proof.* We start by defining a gradient oracle by choosing the noise vectors $\{\xi_t\}_{t=0}^{T-1}$ to be independent random variables such that

$$\xi_t = \langle \xi_t, e_{t+1} \rangle e_{t+1} \text{ and } |\langle \xi_t, e_{t+1} \rangle| \leq \frac{\sigma}{\sqrt{k}}, \tag{47}$$

where $e_{t+1}$ is a basis vector whose $(t+1)$-th entry is one and otherwise is zero. In other words, only the $(t+1)$-th coordinate of $\xi_t$ is nonzero. Otherwise, the rest of the vector remains zero. We use $\bar{\xi}_t$ to denote the averaged noise variable as

$$\bar{\xi}_t = \frac{1}{k}\sum_{i=1}^{k} \xi_t^{(i)},$$

where $\xi_t^{(i)}$ is defined following the condition specified in equation (47). Thus, we can also conclude that

$$|\langle \bar{\xi}_t, e_{t+1} \rangle| \leq \frac{\sigma}{\sqrt{k}}.$$

We consider the objective function $f(W) : \mathbb{R}^d \to \mathbb{R}$ defined above (see also equation (13), Section 4), with

$$\alpha_i = \frac{2\eta_i \sigma}{\sqrt{k}}, \text{ for } i = 0, 1, \ldots, T. \tag{48}$$

We will analyze the dynamics of Algorithm 1 with the objective function $f(W)$ and the starting point $W_0 = D\sqrt{G} \cdot e_0$, where $G = \max\left\{C^{-1}, 2\sum_{i=0}^{T-1} \eta_i\right\}$. For the first iteration, we have

$$W_1 = W_0 - \eta_0 \Big(\frac{1}{2}\sum_{i=1}^{k}\big(\nabla f(W_0 + U_0^{(i)}) + \nabla f(W_0 - U_0^{(i)})\big) + \bar{\xi}_0\Big)$$
$$= (1 - \eta_0 G^{-1})W_0 - \eta_0\bar{\xi}_0,$$

where $U$ is a random draw from the truncated distribution $\mathcal{P}$ with $\langle U, e_i \rangle = \min\{\mathcal{P}_i, a_i\}$ for $a_i = \frac{\eta_{i-1}\sigma}{\sqrt{k}}$. Next, from the construction of $h_1$, we get

$$\frac{1}{2}\big(\nabla f(W_1 + U) + \nabla f(W_1 - U)\big)$$
$$= G^{-1}\langle W_1, e_0 \rangle e_0 + \frac{1}{2}\Big(h_0'\big(\eta_0\langle\bar{\xi}_0, e_1\rangle + \langle U, e_1\rangle\big)e_1 + h_0'\big(\eta_0\langle\bar{\xi}_0, e_1\rangle - \langle U, e_1\rangle\big)e_1\Big).$$

Here, using the fact that $\alpha_0 = \frac{2\eta_0\sigma}{\sqrt{k}}$ from equation (48) above, and the truncation of $U$, which implies $|\langle U, e_1 \rangle| \leq \frac{\eta_0\sigma}{\sqrt{k}}$, and $\langle\bar{\xi}_0, e_1\rangle \leq \frac{\sigma}{\sqrt{k}}$, we obtain

$$\left|\eta_0\langle\bar{\xi}_0, e_1\rangle + \langle U, e_1\rangle\right| \leq \frac{2\eta_0\sigma}{\sqrt{k}} = \alpha_0, \text{ and similarly } \left|\eta_0\langle\bar{\xi}_0, e_1\rangle - \langle U, e_1\rangle\right| \leq \frac{2\eta_0\sigma}{\sqrt{k}} = \alpha_0,$$

which implies that

$$h_0'(\eta_0\langle\bar{\xi}_0, e_1\rangle + \langle U, e_1\rangle) = h_0'(\eta_0\langle\bar{\xi}_0, e_1\rangle - \langle U, e_1\rangle) = 0.$$

This is the first update. Then, in the next iteration,

$$W_2 = W_1 - \eta_1\Big(G^{-1}\langle W_1, e_0\rangle + \bar{\xi}_1\Big)$$
$$= -(1 - \eta_1 G^{-1})(1 - \eta_0 G^{-1})W_0 - \eta_0\bar{\xi}_0 - \eta_1\bar{\xi}_1.$$

Similarly, we use the fact that $\alpha_i = \frac{2\eta_i\sigma}{\sqrt{k}}$ and the fact that $|\langle U, e_{i+1}\rangle| \leq \frac{\eta_i\sigma}{\sqrt{k}}$, which renders the gradient as zero similar to the above reasoning. This holds for any $i = 1, 2, \ldots, T-1$.

At the $t$-th iteration, suppose we have that

$$W_t = W_0 \prod_{i=0}^{t-1}\left(1 - \eta_i G^{-1}\right) - \sum_{i=0}^{t-1}\eta_i \bar{\xi}_i.$$

Then by induction, at the $(t+1)$-th iteration, we must have

$$
\begin{aligned}
W_{t+1} &= W_t - \eta_t\left(G^{-1}\langle W_t, e_0\rangle + \bar{\xi}_t\right) \\
&= W_0 \prod_{i=0}^{t}\left(1 - \eta_i G^{-1}\right) - \sum_{i=0}^{t}\eta_i \bar{\xi}_i.
\end{aligned}
\tag{49}
$$

Next, from the definition of $h_t$ above, we have that

$$
\begin{aligned}
F(W_0) - \min_{W \in \mathbb{R}^d} F(W) = F(W_0) \qquad\qquad & \text{(the minimum can be attained at zero)} \\
= \frac{1}{2G}(D\sqrt{G})^2 + \sum_{i=0}^{T-1}\frac{C}{4}\left(\frac{2\eta_i\sigma}{\sqrt{k}}\right)^2 \qquad\qquad & \text{(since } \langle W_0 + U, e_{i+1}\rangle \leq \alpha_i)
\end{aligned}
$$

The above must be at most $D^2$, which implies that we should set the learning rates to satisfy (after some calculation)

$$\frac{1}{T}\left(\sum_{i=0}^{T-1}\eta_i\right)^2 \leq \sum_{i=0}^{T-1}\eta_i^2 \leq \frac{kD^2}{2C\sigma^2}.\tag{50}$$

We note that for all $z \in [0,1]$, $1 - \frac{z}{2} \geq \exp(\log \frac{z}{2})$. Thus, applying this to the right-hand side of equation (49), we obtain that for any $t$,

$$\prod_{i=0}^{t}\left(1 - \eta_i G^{-1}\right) \geq \frac{1}{2},\tag{51}$$

where we recall that $G = \max\{C^{-1}, 2\sum_{i=0}^{T-1}\eta_i\}$. Our calculation so far shows that for all the $h_i$ except $h_0$, the algorithm has not moved at all from its initialization at $W_0$ under the above gradient noise. We thus conclude that

$$
\begin{aligned}
\min_{1 \leq i \leq T}\|\nabla F(W_i)\|^2 &= \min_{1 \leq i \leq T}\left(G^{-1}\langle W_0, e_0\rangle\right)^2 & \text{(by the construction of } F(\cdot)) \\
&\geq \frac{1}{4}G^{-2}(D\sqrt{G})^2 & \text{(by equations (49) and (51))} \\
&= \frac{D^2}{4}\min\left\{C, \frac{1}{2\sum_{i=0}^{T-1}\eta_i}\right\} & \text{(recall the definition of } G \text{ above)} \\
&\geq \frac{D^2}{4}\min\left\{C, \frac{\sqrt{2C\sigma^2}}{2D\sqrt{kT}}\right\} \geq D\sqrt{\frac{C\sigma^2}{32kT}}. & \text{(by equation (50))}
\end{aligned}
$$

In the first step, we use the fact that $\langle \bar{\xi}_i, e_0\rangle = 0$, for all $0 = 1, 2, \ldots, T-1$. Thus, we have proved that equation (14) holds for $W_i$ for any $i = 1, 2, \ldots, T$. The proof of Lemma B.3 is finished. $\qquad\square$

Next, let us consider another case of the lower bound.

**Lemma B.4.** *In the setting of Theorem 4.3, suppose the learning rates satisfy that $\sum_{i=0}^{T-1}\eta_i \geq \sqrt{\frac{D^2kT}{2\sigma^2C}}$ and $\eta_i = \eta$ for some fixed $\eta \leq C^{-1}$. Then, consider the function from equation (13), we have that $\min_{1 \leq t \leq T}\mathbb{E}\left[\|\nabla F(W_t)\|^2\right] \geq D\sqrt{\frac{C\sigma^2}{32kT}}$.*

*Proof.* We define the functions $g$, parametrized by a fixed, positive constants $\alpha = \frac{1-\rho^T}{1-\rho} \cdot 2c\eta\sigma$, as follows:

$$g(x) = \begin{cases} -\frac{C}{2}x^2 + \frac{C}{4}\alpha^2 & |x| \leq \frac{\alpha}{2}, \\ \frac{C}{2}(|x| - \alpha)^2 & \frac{\alpha}{2} \leq |x| \leq \alpha, \\ 0 & \alpha \leq |x|. \end{cases}$$

One can verify that $\nabla g$ is $C$-Lipschitz continuous, but $g$ is not twice-differentiable. We also consider a chain-like function:

$$f(W) = g(\langle W, e_0 \rangle) + \sum_{t=0}^{d-1} \frac{C}{2} \langle W, e_{t+1} \rangle^2. \tag{52}$$

From the definition of $f$, its gradient is $C$-Lipschitz continuous. Similar to equation (47), we define an adversarial gradient oracle by choosing the noise vectors $\{\xi_t\}_{t=0}^{T-1}$ to be independent random variables such that

$$\xi_t = \langle \xi_t, e_{t+1} \rangle, \mathbb{E}\left[\langle \xi_t, e_{t+1} \rangle^2\right] = \sigma^2, \text{ and } |\langle \xi_t, e_{t+1} \rangle| \leq c\sigma,$$

where $c$ is a fixed constant. We use $\bar{\xi}_t$ to denote the averaged noise variable as

$$\bar{\xi}_t = \sum_{i=1}^{k} \xi_t^{(i)}.$$

Suppose $\{\xi_t^{(i)}\}_{i=1}^{k}$ are i.i.d. random variables for any $t$, we have

$$|\langle \bar{\xi}_t, e_{t+1} \rangle| \leq c\sigma \text{ and } \mathbb{E}\left[\left\|\bar{\xi}_t\right\|^2\right] \leq \frac{\sigma^2}{k}. \tag{53}$$

Next, we analyze the dynamics of Algorithm 1 with the objective function $f(W)$ and the starting point $W_0 = \sum_{i=1}^{d} \sqrt{\frac{D^2}{Cd}} \cdot e_i$. In this case, by setting $\eta_i = \eta$ for all $i = 0, 1, \ldots, T-1$. Recall that $\eta < C^{-1}$. Denote by $\rho = C\eta$, which is strictly less than one.

Since $h_t$ is an even function, its derivative $h_t'$ is odd. For the first iteration, we have

$$W_1 = W_0 - \eta\left(\frac{1}{2}\big(\nabla f(W_0 + U) + \nabla f(W_0 - U)\big) + \bar{\xi}_0\right) = (1 - C\eta)W_0 - \eta\bar{\xi}_0.$$

where $U$ is a truncate distribution of $\mathcal{P} \sim N(0, \text{Id}_d)$ with $\langle U, e_0 \rangle = \min\{\mathcal{P}_0, a_0\}$ and $a_0 = c\eta\sigma$.

Using the fact that $\alpha = \frac{1-\rho^T}{1-\rho} \cdot 2c\eta\sigma$, $|\langle U, e_0 \rangle| \leq c\eta\sigma$, and $\langle \bar{\xi}_0, e_0 \rangle \leq c\sigma$, we have

$$g'(\eta\langle \bar{\xi}_0, e_0 \rangle + \langle U, e_0 \rangle) + g'(\eta\langle \bar{\xi}_0, e_0 \rangle - \langle U, e_0 \rangle) = -2C\eta\langle \bar{\xi}_0, e_0 \rangle.$$

Then, in the next iteration,

$$W_2 = W_1 - \eta\left(C\sum_{i=1}^{d}\langle W_1, e_i \rangle - C\eta\bar{\xi}_0 + \bar{\xi}_1\right) = (1 - C\eta)^2 W_0 - (1 - C\eta)\eta\bar{\xi}_0 - \eta\bar{\xi}_1.$$

Similarly, we use the fact that $\alpha = \frac{1-\rho^T}{1-\rho} \cdot 2c\eta\sigma$ and the fact that $|\langle U, e_0 \rangle| \leq c\eta\sigma$, which renders the gradient as $g'(x) = -Cx$, for any $i = 1, 2, \ldots, T-1$.

At the $t$-th iteration, suppose that

$$W_t = (1 - C\eta)^t W_0 - \sum_{i=0}^{t-1}(1 - C\eta)^{t-1-i}\eta\bar{\xi}_i.$$

Then by induction, at the $(t+1)$-th iteration, we have

$$W_{t+1} = W_t - \eta\Big(C\sum_{i=1}^{d}\langle W_t, e_i\rangle - C\sum_{i=0}^{t-1}(1-C\eta)^{t-1-i}\eta\bar{\xi}_i + \bar{\xi}_t\Big)$$

$$= (1-C\eta)^{t+1}W_0 - \sum_{i=0}^{t}(1-C\eta)^{t-1-i}\eta\bar{\xi}_i. \tag{54}$$

Next, from the definition of $F$ above, we have that

$$F(W_0) - \min_{W\in\mathbb{R}^d}F(W) = F(W_0) = \frac{dC}{2}\Big(\sqrt{\frac{D^2}{Cd}}\Big)^2 + \frac{C}{4}\Big(\frac{2(1-\rho^T)c\eta\sigma}{(1-\rho)}\Big)^2, \qquad \text{(since } \langle W_0 + U, e_0\rangle \le \alpha)$$

which must be at most $D^2$. Thus, we must have (after some calculation)

$$c^2 \le \frac{D^2(1-\rho)^2}{2\sigma^2\rho^2(1-\rho^T)^2}.$$

We conclude that

$$\min_{1\le i\le T}\mathbb{E}\Big[\|\nabla F(W_i)\|^2\Big] = \min_{1\le i\le T}\mathbb{E}\Big[\sum_{j=1}^{d}C^2\langle W_i, e_j\rangle^2 + C^2\langle W_i, e_0\rangle^2\Big]$$

$$= \min_{1\le i\le T}\Big(dC^2(1-\rho)^{2t}\Big(\sqrt{\frac{D^2}{Cd}}\Big)^2 + \frac{\sigma^2}{k}\cdot\rho^2\sum_{i=0}^{t}(1-\rho)^{2(t-1-i)}\Big)$$

$$\ge \min_{1\le i\le T}\Big(CD^2(1-\rho)^{2t} + \frac{\sigma^2}{k}\frac{\rho}{2-\rho}\big(1-(1-\rho)^{2t}\big)\Big) \ge \min\Big\{CD^2, \frac{\sigma^2}{k}\frac{\rho}{2-\rho}\Big\}$$

$$\ge \frac{\sigma^2}{k}C\sqrt{\frac{kD^2}{2T\sigma^2C}}\frac{1}{2-C\sqrt{\frac{kD^2}{2T\sigma^2C}}} \ge D\sqrt{\frac{C\sigma^2}{16k\cdot T}}. \qquad \text{(after some calculation)}$$

Thus, we have proved this lemma. $\qquad\square$

Taking both Lemma B.3 and B.4 together, we thus conclude the proof of Theorem 4.3.

## B.3 Proof of momentum lower bound

In this section, we prove the following result.

**Theorem B.5.** *There exists a quadratic function $f$ such that for the iterates $W_1,\ldots,W_T$ generated by equation* (15)*, we must have:* $\min_{1\le t\le T}\mathbb{E}\Big[\|\nabla F(W_t)\|^2\Big] \ge O\big(D\sqrt{\frac{C\sigma^2}{k\cdot T}}\big)$*.*

We will focus on a perturbation distribution $\mathcal{P}$ equal to the isotropic Gaussian distribution for this result. In this case, we know that $F(W) = f(W) + d$. For the quadratic function $f(W) = \frac{C}{2}\|W\|^2$, its gradient is $C$-Lipschitz continuous. We set the initialization $W_0\in\mathbb{R}^d$ such that

$$F(W_0) - \min_{W\in\mathbb{R}^d}F(W) = D^2.$$

This condition can be met when we set $W_0$ as a vector whose Euclidean norm is equal to

$$D\sqrt{2\max\Big\{C^{-1}, 2\sum_{i=0}^{T-1}\eta_i\Big\}}.$$

**The case when $\mu = 0$.** We begin by considering the case when $\mu = 0$. In this case, the update reduces to SGD, and the iterate $W_{t+1}$ evolves as follows:

$$W_{t+1} = \left(1 - C\eta_t\right)W_t - \eta_t\bar{\xi}_t, \tag{55}$$

where we denote $\bar{\xi}_t$ as the averaged noise $k^{-1}\sum_{j=1}^k \xi_t^{(j)}$, and the noise perturbation $U_t^{(j)}$ cancelled out between the plus and minus perturbations. The case when $\mu > 0$ builds on this simpler case, as we will describe below.

The key observation is that the gradient noise sequence $\bar{\xi}_1, \bar{\xi}_2, \ldots, \bar{\xi}_T$ forms a martingale sequence:

- For any $i = 1, 2, \ldots, T$, conditioned on the previous random variables $\xi_{i'}^{(j)}$ for any $i' < i$ and any $j = 1, 2, \ldots, k$, the expectation of $\bar{\xi}_i$ is equal to zero.

- In addition, the variance of $\bar{\xi}_i$ is equal to $k^{-1}\sigma^2$, since conditional on the previous random variables, the $\xi_i^{(j)}$s are all independent from each other.

The martingale property allows us to characterize the SGD path of $\|W_t\|^2$, as shown in the following result.

**Lemma B.6.** *In the setting of Theorem B.5, for any step sizes $\eta_0, \ldots, \eta_{T-1}$ less than $C^{-1}$, and any $t = 1, \ldots, T$, the expected gradient of $W_t$, $\mathbb{E}\left[\|\nabla F(W_t)\|^2\right]$, is equal to*

$$2CD^2 \prod_{j=0}^{t-1}\left(1 - C\eta_j\right)^2 + \frac{C\sigma^2}{k}\sum_{i=0}^{t-1}\eta_i^2 \prod_{j=i+1}^{t-1}\left(1 - C\eta_j\right)^2.$$

*Proof.* By iterating over equation (55), we can get

$$W_t = W_0 \prod_{j=0}^{t-1}\left(1 - C\eta_j\right) - \sum_{i=0}^{t-1}\eta_i\bar{\xi}_i \prod_{j=i+1}^{t-1}\left(1 - C\eta_j\right).$$

Meanwhile,

$$\nabla F(W_t) = CW_t \Rightarrow \|\nabla F(W_t)\|^2 = C^2\|W_t\|^2.$$

Thus, by squaring the norm of $W_t$ and taking the expectation, we can get

$$\mathbb{E}\left[\|\nabla F(W_t)\|^2\right] = C^2\|W_0\|^2 \prod_{j=0}^{t-1}\left(1 - C\eta_j\right)^2 + C^2\sum_{i=0}^{t-1}\mathbb{E}\left[\left\|\eta_i\bar{\xi}_i \prod_{j=i+1}^{t-1}\left(1 - C\eta_j\right)\right\|^2\right]. \tag{56}$$

Above, we use martingale property a), which says the expectation of $\bar{\xi}_i$ is equal to zero for all $i$. In addition, based on property b), equation (56) is equal to

$$C^2\sum_{i=0}^{t-1}\eta_i^2\left(\prod_{j=i+1}^{t-1}\left(1 - C\eta_j\right)^2 \mathbb{E}\left[\|\bar{\xi}_i\|^2\right]\right) = \frac{C^2\sigma^2}{k}\sum_{i=0}^{t-1}\eta_i^2 \prod_{j=i+1}^{t-1}\left(1 - C\eta_j\right)^2.$$

To see this, based on the martingale property of $\bar{\xi}$ again, the cross terms between $\bar{\xi}_i$ and $\bar{\xi}_j$ for different $i, j$ are equal to zero in expectation:

$$\mathbb{E}\left[\langle\bar{\xi}_i, \bar{\xi}_j\rangle | \bar{\xi}_j\right] = 0, \text{ for all } 1 \le j < i \le T.$$

Additionally, the second moment of $\bar{\xi}_i$ satisfies:

$$\mathbb{E}\left[\|\bar{\xi}_i\|^2\right] = \frac{\sigma^2}{k}, \text{ for any } i = 1, \ldots, T.$$

Lastly, let $W_0$ be a vector such that

$$\|W_0\| = D\sqrt{2C^{-1}} \Rightarrow F(W_0) - \min_{W \in \mathbb{R}^d} F(W) \leq D^2.$$

Setting $\|W_0\| = D\sqrt{2C^{-1}}$ in equation (56) leads to

$$\mathbb{E}\left[\|\nabla F(W_t)\|^2\right] = 2CD^2 \prod_{j=0}^{t-1} \left(1 - C\eta_j\right)^2 \quad + \frac{C^2\sigma^2}{k} \sum_{i=0}^{t-1} \eta_i^2 \prod_{j=i+1}^{t-1} \left(1 - C\eta_j\right)^2.$$

Thus, we conclude the proof of this result. $\qquad\qquad\square$

We now present the proof for the case when $\sum_{i=0}^{T-1} \eta_i \leq O(\sqrt{T})$. For this result, we will use the following quadratic function:

$$f(W) = \frac{1}{2\kappa} \|W\|^2, \text{ where } \kappa = \max\{C^{-1}, 2\sum_{i=0}^{T-1} \eta_i\}, \tag{57}$$

**Lemma B.7.** *Consider $f$ given in equation (57) above. For any step sizes $\eta_0, \ldots, \eta_{T-1}$ less than $C^{-1}$, the following holds for the stochastic objective $F$:*

$$\min_{1 \leq t \leq T} \mathbb{E}\left[\|\nabla F(W_t)\|^2\right] \geq \frac{D^2}{2\max\{C^{-1}, 2\sum_{i=0}^{T-1} \eta_i\}}.$$

*Proof.* The norm of the gradient of $F(W)$ is equal to

$$\|\nabla F(W)\| = \frac{1}{\kappa} \|W\|. \tag{58}$$

Following the update rule in NSO, similar to equation (55), $W_t$ evolves as follows:

$$W_{t+1} = \left(1 - \frac{\eta_t}{\kappa}\right) W_t - \eta_t \bar{\xi}_t, \tag{59}$$

where $\bar{\xi}_t$ has variance equal to $\sigma^2/k$, according to the proof of Lemma B.6. By iterating equation (59) from the initialization, we can get a closed-form equation for $W_t^{(1)}$, for any $t = 1, 2, \ldots, T$:

$$W_t = W_0 \prod_{j=0}^{t-1} \left(1 - \frac{\eta_j}{\kappa}\right) - \sum_{k=0}^{t-1} \eta_k \xi_k \prod_{j=k+1}^{t-1} \left(1 - \frac{\eta_j}{\kappa}\right). \tag{60}$$

Following equation (58), we can show that $\|\nabla F(W)\|^2 = \kappa^{-2} \|W_t\|^2$. Thus, in expectation,

$$\mathbb{E}\left[\|\nabla F(W_t)\|^2\right] = \kappa^{-2} \mathbb{E}\left[\|W_t\|^2\right]$$

$$= \kappa^{-2} \|W_0\|^2 \prod_{j=0}^{t-1} \left(1 - \kappa^{-1}\eta_j\right)^2 + \kappa^{-2} \sum_{i=0}^{t-1} \mathbb{E}\left[\left(\eta_i \bar{\xi}_i \prod_{j=i+1}^{t-1} \left(1 - \kappa^{-1}\eta_j\right)\right)^2\right]$$

$$= \kappa^{-2} \|W_0\|^2 \prod_{j=0}^{t-1} \left(1 - \kappa^{-1}\eta_j\right)^2 + \kappa^{-2} \sum_{i=0}^{t-1} \eta_i^2 \prod_{j=i+1}^{t-1} \left(1 - \kappa^{-1}\eta_j\right)^2 \mathbb{E}\left[\|\bar{\xi}_i\|^2\right]$$

$$= 2D^2\kappa^{-1} \prod_{j=0}^{t-1} \left(1 - \kappa^{-1}\eta_j\right)^2 + \frac{\sigma^2\kappa^{-2}}{k} \sum_{i=0}^{t-1} \eta_i^2 \prod_{j=i+1}^{t-1} \left(1 - \kappa^{-1}\eta_j\right)^2, \tag{61}$$

where we use the definition of initialization $W_0$ and the variance of $\bar{\xi}_i$ in the last step. In order to tackle equation (61), we note that for all $z \in [0,1]$,

$$1 - \frac{z}{2} \geq \exp\left(\log\frac{1}{2} \cdot z\right). \tag{62}$$

Hence, applying equation (62) to the right-hand side of equation (61), we obtain that for any $i = 0, 1, \ldots, t-1$,

$$\prod_{j=i}^{t-1}\left(1 - \frac{\eta_j}{\max\{C^{-1}, 2\sum_{j=i}^{T-1}\eta_i\}}\right) \geq \exp\left(\log\frac{1}{2} \cdot \sum_{j=i}^{t-1}\frac{\eta_j}{\max\{(2C)^{-1}, \sum_{i=0}^{T-1}\eta_i\}}\right) \geq \frac{1}{2}.$$

Thus, equation (61) must be at least

$$\mathbb{E}\left[\|\nabla F(W_t)\|^2\right] \geq \frac{2D^2\kappa^{-1}}{4} + \frac{\sigma^2\kappa^{-2}}{k}\sum_{i=0}^{t-1}\frac{\eta_i^2}{4}. \tag{63}$$

The above result holds for any $t = 1, 2, \ldots, T$. Therefore, we conclude that

$$\min_{1 \leq t \leq T}\mathbb{E}\left[\|\nabla F(W_t)\|^2\right] \geq \frac{D^2}{2\max\{C^{-1}, 2\sum_{i=0}^{T-1}\eta_i\}}.$$

Thus, the proof of Lemma B.7 is finished. $\qquad\square$

Next, we consider the other case, which is when the learning rates are fixed.

**Lemma B.8.** *There exists convex quadratic functions $f$ such that for any gradient oracle satisfying Assumption 4.1 and any distribution $\mathcal{P}$ with mean zero, if $\eta_i = \eta < C^{-1}$ for any $i = 1, \ldots, T$, or if $\sum_{i=0}^{T-1}\eta_i \lesssim \sqrt{T}$, then the following must hold:*

$$\min_{1 \leq t \leq T}\mathbb{E}\left[\|\nabla F(W_t)\|^2\right] \geq D\sqrt{\frac{C\sigma^2}{32k \cdot T}}. \tag{64}$$

*Proof.* By Lemma B.7, there exists a function such that the left-hand side of equation (64) is at least

$$\frac{D^2}{2\max\{C^{-1}, 2\sum_{i=0}^{T-1}\eta_i\}} \geq \frac{CD^2}{2\max\{1, 2x^{-1}\sqrt{T}\}} = \frac{D^2 x}{4\sqrt{T}}, \tag{65}$$

which holds if $\sum_{i=0}^{T-1}\eta_i \leq \sqrt{T}x^{-1}$ for any fixed $x > 0$.

On the other hand, if $\sum_{i=0}^{T-1}\eta_i \geq x^{-1}\sqrt{T}$ and $\eta_i = \eta$ for a fixed $\eta$, then $\eta > x^{-1}/\sqrt{T}$. By setting $\eta_i = \eta$ for all $i$ in Lemma B.6, the left-hand side of equation (64) is equal to

$$\min_{1 \leq t \leq T}\left(2CD^2(1 - C\eta)^{2t} + \frac{C^2\sigma^2}{k}\sum_{k=0}^{t-1}\eta^2(1 - C\eta)^{2(t-k-1)}\right).$$

Recall that $\eta < C^{-1}$. Thus, $\rho = C\eta$ must be less than one. With some calculations, we can simplify the above to

$$\min_{1 \leq t \leq T}\left(2CD^2(1-\rho)^{2t} + \frac{\sigma^2\rho^2}{k}\frac{1 - (1-\rho)^{2t}}{1 - (1-\rho)^2}\right)$$
$$= \min_{1 \leq t \leq T}\left(\frac{\sigma^2\rho}{k(2-\rho)} + (1-\rho)^{2t}\left(2CD^2 - \frac{\sigma^2\rho}{k(2-\rho)}\right)\right). \tag{66}$$

If $2CD^2 < \frac{\sigma^2\rho}{k(2-\rho)}$, the above is the smallest when $t = 1$. In this case, equation (66) is equal to

$$2CD^2(1-\rho)^2 + \frac{\sigma^2\rho^2}{k} \geq \frac{1}{\frac{1}{2CD^2} + \frac{k}{\sigma^2}} = O(1).$$

If $2CD^2 \geq \frac{\sigma^2 \rho}{k(2-\rho)}$, the above is the smallest when $t = T$. In this case, equation (66) is at least

$$\frac{\sigma^2 \rho}{k(2-\rho)} \geq \frac{\sigma^2 \rho}{2k} \geq \frac{\sigma^2 C x^{-1}}{2k} \cdot \frac{1}{\sqrt{T}}. \tag{67}$$

To conclude the proof, we set $x$ so that the right-hand side of equations (65) and (67) match each other. This leads to

$$x = \sqrt{\frac{2\sigma^2 C}{kD^2}}.$$

Thus, by combining the conclusions from both equations (65) and (67) with this value of $x$, we finally conclude that if $\sum_{i=0}^{T-1} \eta_i \leq \sqrt{T}x^{-1}$, or for all $i = 0, \ldots, T-1$, $\eta_i = \eta < C^{-1}$, then in both cases, there exists a function $f$ such that equation (64) holds. This completes the proof of Lemma B.8. □

**The case when $\mu > 0$.** In this case, since the update of $W_t$ also depends on the momentum update, it becomes significantly more involved. One can verify that the update from step $t$ to step $t+1$ is based on

$$X_u = \begin{bmatrix} 1 - C\eta_t & \mu \\ C\eta_t & \mu \end{bmatrix}. \tag{68}$$

Our analysis examines the eigenvalues of the matrix $X_u X_u^\top$ and the first entry in the corresponding eigenvectors. Particularly, we show that the two entries are bounded away from zero. Then, we apply the Hölder's inequality to reduce the case of $\mu > 0$ to the case of $\mu = 0$, Lemma B.8 in particular.

*Proof.* First, consider a quadratic function

$$f(W) = \frac{1}{2C} \|W\|^2.$$

Clearly, $f(W)$ is $C$-Lipschitz continuous. Further, $F(W) = f(W) + d$, for $\mathcal{P}$ being the isotropic Gaussian. Let $W_0$ be a vector whose Euclidean norm equals $D\sqrt{2C}$. Thus,

$$F(W_0) - \min_{W \in \mathbb{R}^d} F(W) = D^2.$$

As for the dynamic of momentum SGD, recall that

$$M_{t+1} = \mu M_t - \eta_t G_t \text{ and } W_{t+1} = W_t + M_{t+1}.$$

We consider the case where $\eta_t = \eta$ for all steps $t$. In this case, we can write the above update into a matrix notation as follows:

$$\begin{bmatrix} W_{t+1} \\ M_{t+1} \end{bmatrix} = \begin{bmatrix} 1 - C\eta & \mu \\ -C\eta & \mu \end{bmatrix} \begin{bmatrix} W_t \\ M_t \end{bmatrix} + C\eta \begin{bmatrix} \bar{\xi}_t \\ \bar{\xi}_t \end{bmatrix}.$$

Let $X_\mu = [1 - C\eta, \mu; -C\eta, \mu]$ denote the 2 by 2 matrix (that depends on $\mu$) above. Similar to Lemma B.6, we can apply the above iterative update to obtain the formula for $W_{t+1}$ as:

$$\begin{bmatrix} W_{t+1} \\ M_{t+1} \end{bmatrix} = X_u^t \begin{bmatrix} W_0 \\ M_0 \end{bmatrix} + \sum_{i=0}^{t} C\eta X_u^{t-i} \begin{bmatrix} \bar{\xi}_i \\ \bar{\xi}_i \end{bmatrix}. \tag{69}$$

By multiplying both sides by the vector $e_1 = [1, 0]^\top$, and then taking the Euclidean norm of the vector (notice that this now only evolves that $W_{t+1}$ vector on the left, and the $W_t$ vector on the right), we now obtain that, in expectation over the randomness of the $\bar{\xi}_i$'s, the following holds:

$$\mathbb{E}\left[\|W_{t+1}\|^2\right] = 2CD^2 (e_1^\top X_u^t e_1)^2 + \frac{C^2 \eta^2 \sigma^2}{k} \sum_{i=0}^{t} \|e_1^\top X_u^i e\|^2. \tag{70}$$

Above, similar to Lemma B.6, we have set the length of $W_0$ appropriately so that its size is equal to $D\sqrt{2C^{-1}}$, which has led to the $CD^2$ term above. Recall that $M_0$ is equal to zero in the beginning. To get the first term above, we follow this calculation:

$$
\begin{aligned}
\left\| e_1^\top X_\mu^t \begin{bmatrix} W_0 \\ M_0 \end{bmatrix} \right\|^2 &= \mathrm{Tr}\left[ e_1^\top X_\mu^t \begin{bmatrix} W_0 \\ M_0 \end{bmatrix} \begin{bmatrix} W_0 \\ M_0 \end{bmatrix}^\top {X_\mu^t}^\top e_1 \right] \\
&= \mathrm{Tr}\left[ e_1^\top X_\mu^t \begin{bmatrix} CD^2 & 0 \\ 0 & 0 \end{bmatrix} {X_\mu^t}^\top e_1 \right] \\
&= 2CD^2 (e_1^\top X_\mu^t e_1)^2.
\end{aligned}
$$

We use $e = [1,1]^\top$ to denote the vector of ones. Now, we focus on the 2 by 2 matrix $X_u$ (recall this is the coefficient matrix on the right side of equation (69)). Let its singular values be denoted as $\lambda_1$ and $\lambda_2$. In addition, to deal with equation (70), let $\alpha_1$ and $\alpha_2$ denote the first entry of $X_u$'s left singular vectors, corresponding to $a$ and $b$, respectively. Thus, we can write

$$
(e_1^\top X_\mu^i e)^2 = \alpha_1^2 \lambda_1^{2i} + \alpha_2^2 \lambda_2^{2i}. \tag{71}
$$

Now, one can verify that $\lambda_1^2$ and $\lambda_2^2$ are the roots of the following quadratic equation over $x$:

$$
x^2 - ((1-C\eta)^2 + (C\eta)^2 + 2\mu^2)x + \mu^2 = 0. \tag{72}
$$

This can be checked by first taking $X_u$ times $X_u^\top$, then using the definition of the eigenvalues by calculating the determinant of $X_u X_u^\top - x\,\mathrm{Id} = 0$. Thus, we have that $\lambda_1$ and $\lambda_2$ are equal to:

$$
\lambda_1, \lambda_2 = \frac{(1-C\eta)^2 + (C\eta)^2 + 2\mu^2 \pm \sqrt{((1-C\eta)^2 + (C\eta)^2 + 2\mu^2)^2 - 4\mu^2}}{2}. \tag{73}
$$

Now, $\alpha_1^2$ (and $\alpha_2^2$, respectively) satisfies that:

$$
\alpha_1^2 = \frac{-C\eta(1-C\eta) + \mu^2}{(1-C\eta)^2 + \mu^2 - \lambda_1 + -C\eta(1-C\eta) + \mu^2}. \tag{74}
$$

By enumerating the possible values of $C\eta$ between 0 and 1, one can verify that for a fixed value of $\mu$, $\alpha_1^2$ and $\alpha_2^2$ are both bounded below from zero. Therefore, we can claim that from equation (71),

$$
\alpha_1^2 \lambda_1^{2i} + \alpha_2^2 \lambda_2^{2i} \gtrsim \lambda_1^{2i} + \lambda_2^{2i}. \tag{75}
$$

By the Hölder's inequality,

$$
(\lambda_1^{2i} + \lambda_2^{2i})^{\frac{1}{2i}} (1+1)^{1-\frac{1}{2i}} \geq \lambda_1 + \lambda_2 = (1-C\eta)^2 + (C\eta)^2 + 2\mu^2 \tag{76}
$$
$$
\geq (1-C\eta)^2 + (C\eta)^2, \tag{77}
$$

which implies that

$$
\lambda_1^{2i} + \lambda_2^{2i} \geq \frac{((1-C\eta)^2 + (C\eta)^2)^i}{2^{(2i-1)}}. \tag{78}
$$

Now, we consider two cases. If $C\eta < 1/2$, then the above is greater than $(1-C\eta)^{2i}$, which holds for any $i = 0, 1, \ldots, T-1$. By way of reduction, we can follow the proof of Lemma B.8 to complete this proof. If $C\eta > 1/2$, then the above is greater than $(C\eta)^{2i}$. Again by following the proof steps in Lemma B.8, we can show that

$$
\min_{t=1}^{T} \mathbb{E}\left[ \|W_t\|^2 \right] \gtrsim D\sqrt{\frac{C\sigma^2}{k \cdot T}}.
$$

This completes the proof of Theorem B.5. $\qquad\qquad\square$

# C Experiment Details

We describe the setup for Figure 2, ran on (1) a two-layer Multi-Layer Perceptron (MLP) trained on the MNIST digit classification dataset, (2) a twelve-layer BERT-Base model trained on the MRPC sentence classification dataset from the GLUE benchmark and (3) a two-layer Graph Convolutional Network (GCN) trained on the COLLAB node classification dataset. We set both MLP and GCN with a hidden dimension of 128 for model architectures and initialize them randomly. We initialize the BERT model from pretrained BERT-Base-Uncased. We train each model on the provided training set for the training process until the training loss is close to zero. Specifically, we train the MLP, BERT, and GCN models for 30, 10, and 100 epochs. We use the model of the last epoch to measure the error in the approximation. We do this for 100 times and again measure the perturbed loss $\ell_{\mathcal{Q}}$ on the training set. We take the gap between $\ell_{\mathcal{Q}}$ and $\ell$. Our measurements show that the error between the actual gap and the Hessian approximation is within 3%.

Table 6 reports additional comparisons between our approach and several baselines, including label smoothing (LS), random SAM (RSAM), and Bayesian SAM (BSAM). We report the test accuracy and the trace of the Hessian for the model weights at the last epoch of training on six image classification datasets. We observe that NSO also further reduces the trace of the Hessian and improves the test accuracy over the baselines. The largest eigenvalue reduces by **9.7**%.

Table 6: Comparison between our approach (NSO), label smoothing (LS), random-SAM (RSAM), and Bayesian SAM (BSAM). Also included is the largest eigenvalue of the Hessian.

| | | CIFAR-10 | CIFAR-100 | Aircraft | Caltech-256 | Indoor | Retina |
|---|---|---|---|---|---|---|---|
| **Trace** ($\downarrow$) | LS | $2690 \pm 85$ | $10669 \pm 363$ | $5699 \pm 72$ | $3482 \pm 85$ | $3650 \pm 82$ | $17681 \pm 193$ |
| | RSAM | $2379 \pm 89$ | $9762 \pm 422$ | $4665 \pm 95$ | $3224 \pm 97$ | $3425 \pm 70$ | $16950 \pm 257$ |
| | BSAM | $2768 \pm 54$ | $9787 \pm 465$ | $4750 \pm 55$ | $3498 \pm 38$ | $3162 \pm 73$ | $16238 \pm 286$ |
| | **NSO** | $\mathbf{1728} \pm 79$ | $\mathbf{5244} \pm 89$ | $\mathbf{3678} \pm 83$ | $\mathbf{2958} \pm 77$ | $\mathbf{2737} \pm 90$ | $\mathbf{10970} \pm 146$ |
| **Test Acc** ($\uparrow$) | LS | $96.9\% \pm 0.1$ | $83.8\% \pm 0.1$ | $59.0\% \pm 0.2$ | $76.6\% \pm 0.2$ | $76.5\% \pm 0.3$ | $64.2\% \pm 0.7$ |
| | RSAM | $96.8\% \pm 0.1$ | $84.0\% \pm 0.1$ | $60.9\% \pm 0.4$ | $76.4\% \pm 0.1$ | $76.8\% \pm 0.5$ | $65.9\% \pm 0.3$ |
| | BSAM | $96.9\% \pm 0.1$ | $83.9\% \pm 0.2$ | $61.0\% \pm 0.3$ | $76.8\% \pm 0.3$ | $76.4\% \pm 0.3$ | $65.4\% \pm 0.2$ |
| | **NSO** | $\mathbf{97.6\%} \pm 0.4$ | $\mathbf{84.9\%} \pm 0.3$ | $\mathbf{63.2\%} \pm 0.3$ | $\mathbf{78.1\%} \pm 0.5$ | $\mathbf{78.2\%} \pm 0.3$ | $\mathbf{67.0\%} \pm 0.4$ |

| | | CIFAR-10 | CIFAR-100 | Aircraft | Caltech-256 | Indoor | Retina |
|---|---|---|---|---|---|---|---|
| **$\lambda_1$** ($\downarrow$) | SGD | $1442 \pm 63$ | $4639 \pm 95$ | $1152 \pm 40$ | $1064 \pm 44$ | $1087 \pm 56$ | $8276 \pm 91$ |
| | LS | $1311 \pm 81$ | $3051 \pm 95$ | $1144 \pm 88$ | $893 \pm 79$ | $764 \pm 75$ | $4296 \pm 74$ |
| | SAM | $1326 \pm 72$ | $2625 \pm 91$ | $890 \pm 90$ | $948 \pm 95$ | $887 \pm 53$ | $4033 \pm 52$ |
| | USAM | $1245 \pm 43$ | $2299 \pm 98$ | $592 \pm 32$ | $782 \pm 38$ | $755 \pm 58$ | $3893 \pm 55$ |
| | ASAM | $1383 \pm 73$ | $2638 \pm 86$ | $615 \pm 95$ | $795 \pm 72$ | $697 \pm 36$ | $3925 \pm 56$ |
| | RSAM | $1356 \pm 69$ | $2901 \pm 121$ | $895 \pm 74$ | $779 \pm 68$ | $988 \pm 65$ | $4537 \pm 58$ |
| | BSAM | $1375 \pm 86$ | $2788 \pm 177$ | $972 \pm 79$ | $843 \pm 97$ | $939 \pm 73$ | $4123 \pm 87$ |
| | **NSO** | $\mathbf{1070} \pm 74$ | $\mathbf{2059} \pm 45$ | $\mathbf{579} \pm 59$ | $\mathbf{643} \pm 57$ | $\mathbf{639} \pm 72$ | $\mathbf{3681} \pm 66$ |

Finally, we report the hyper-parameters for the experiments in Section 3. These include a learning rate of 0.0002, momentum of 0.99, weight decay of 0.0001, batch size of 32, and training epochs of 60. We reduce the learning rate by 0.1 every 20 epochs. We choose these hyper-parameters based on a grid search on the validation split. The range in which we conduct a grid search is as follows: Learning rate: 0.005, 0.002, 0.001, 0.0005, 0.0002, and 0.0001; Momentum: 0.9, 0.95, 0.99; Weight decay: 0.01, 0.001, 0.0001; Epochs: 20, 40, and 60; Batch size: 16, 32, and 64.

Each baseline method may have its own set of hyper-parameters, which are adjusted via a grid search. For label smoothing, we choose the weight of the loss calculated from the incorrect labels between 0.1, 0.2, and 0.3; For SAM and BSAM, we choose the $\ell_2$ norm of the perturbation between 0.01, 0.02, and 0.05; For ASAM, we choose the $\ell_2$ norm of the perturbation for the weights between 0.5, 1.0, and 2.0; For RSAM, we choose the $\ell_2$ norm of the perturbation between 0.01, 0.02, and 0.05 and the standard deviation for sampling perturbation between 0.008, 0.01, and 0.012.

