# OpenReview forum: "Noise Stability Optimization for Finding Flat Minima: A Hessian-based Regularization Approach"
_TMLR — Accepted by TMLR_

### Review · Reviewer_wuA1 · 2024-06-07

**Summary Of Contributions:**

The authors proposed a new algorithm called Noise Stability Optimization. The idea itself is very simple - to avoid introducing first-order term $U^\top \nabla f(W)$, use $\frac{1}{2}(g(W+U)+g(W-U))$ to cancel out the first-order perturbation.
They give upper and lower bounds on the gradient norms of the converged solution of their proposed algorithm. The bounds can also be extended to momentum updates.

They also conduct multiple experimental evaluations of their algorithm, compared with a number of “sharpness” reducing training methods. By their experiment, they empirically validate the regularization of the algorithm on the Hessian matrix.

**Audience:**

No

**Claims And Evidence:**

No

**Requested Changes:**

1) Even though they frequently mentioned the word PAC-Bayes, there is no PAC-Bayes generalization bound result in this main paper. It is very misleading (especially for the readers who are experts in PAC-Bayes but not in optimization). For completeness, it would be great if authors clearly state the effect of smaller hessian in the Appendix or somewhere. That said, I've read the related works and illustration paragraphs already. I want something more mathematical and rigorous statement to see how it works.

2) Also, their main result is mainly focusing on minimizing the gradient norm, but it is not crystal clear why minimizing the norm leads the better generalization. Demonstrating the loss convergence, even in a strongly convex environment, would be better to convince readers I think.

3) Is this the first paper using 'canceling perturbation'? I mean, $g(W-U)+g(W+U)$ type of approach to cancel out the first-order term is quite common in many statistical machine learning studies.

4) Minor: It would be great if authors could cite some studies that use Assumptions 2 and 3, to support the generality of those assumptions.

**Strengths And Weaknesses:**

# Strength:

1) Their algorithm is simple and intuitive.

2) They provided numerous experiments to prove the superiority of their algorithm. Maybe the practitioners will like this paper.


# Weakness:

1) Even though they emphasized the connection with PAC-Bayes bound throughout the paper, their result itself is not much related to PAC-Bayes. They don't provide any PAC-Bayes bound. What they did was simply sweep the hessian chores to PAC-Bayes result on page 6,  **Illustration** paragraph.

2) Their theoretical result itself is relatively weak. Their hessian result is empirical, and even the effect of hessian is not clearly stated (just illustration paragraph). Their theoretical result is only about the scale of $\|\nabla F(W)\|$, and they didn't provide enough theoretical comparison between other works. The objective of optimization is reducing the loss (maybe in this paper generalization error) but they didn't prove this smaller gradient norm leads to smaller generalization error mathematically. They all compared numerically, so as a theoretician, I am not clear about the superiority of their result.

3) (Not sure, I will remove this when it is true) I'm not sure whether this work is the first 'canceling trick of perturbation' in the field of optimization. I will also check other reviewers and literature to see whether their work is really novel.

---

> ### Author Response · Authors · 2024-06-07
> **Response to Reviewer wuA1**
>
> Response to Requested Changes:
> 1. Thank you for this kind suggestion! We are working on a PAC-Bayes bound based on the Hessian and we will add the details to the appendix of the manuscript.
> 2. Thanks again for this constructive feedback! We will add a result stating the generalization bound in a strongly convex function.
> 3. There’s a related work around “two-point gradient estimation” in Duchi et al. “Optimal rates for zero-order convex optimization: the power of two function evaluations.” That paper focuses on zeroth-order optimization. However, our paper studies a completely different setting. To the best of our knowledge, a thorough empirical analysis of this two-sample estimation for training neural networks has not been carried out before. Although the empirical results are consistently positive, getting such a result is not so obvious, since, naively adding noise (without the negative injection) does not always perform so well empirically. Please refer to our experimental analysis in Section 4.2.1 for an extended discussion.
> 4. Yes, these two assumptions are standard and we will add citations to optimization textbooks in the revision.
>
> We will post a revision to address the first and the second comment as soon as we can and discuss the third comment more extensively in the manuscript.
>
>
> Response to Weaknesses:
> 1. As described above, we are working on adding a PAC-Bayes bound to the paper. We hope this addition will alleviate the reviewer’s concern on this point.
> 2. We would like to emphasize that our main contribution is to provide a comprehensive empirical analysis of the “noise stability optimization” algorithm. This is what we have highlighted in our abstract. The theoretical results are meant to justify the consistency of our algorithm. This is unlike many other "sharpness reducing" deep learning training methods, whose theoretical properties are usually opaque. We believe this is a key strength of our approach, compared to prior methods.
> 3. As mentioned above, we believe that the thorough empirical analysis we have conducted is a novel contribution to the Machine Learning Research literature. Thus, we think that our research fits into the scope of this venue.
>
> Please kindly let us know if you have any additional concerns about our work. We'll do our best to address your comments in the revision. We thank you for your time in reviewing our paper.

---

> ### Author Response · Authors · 2024-07-26
> **Follow up response to Reviewer wuA1**
>
> Dear Reviewer wuA1,
>
> We have recently completed a revision of our paper. In particular, we have worked on addressing the comments that you highlighted, including weaknesses in our theoretical results, and a lack of PAC-Bayes generalization bounds in the paper.
>
> In particular, we added Theorem 2.1, in Section 2.3, Page 5. Just to give you a brief description of this PAC-Bayes bound, we used a linear PAC-Bayes bound from the work of Catoni (2007) and David McAllester (2013):
>
> - Catoni, Olivier. "PAC-Bayesian supervised classification: the thermodynamics of statistical learning." arXiv preprint arXiv:0712.0248 (2007).
> - McAllester, David. "A PAC-Bayesian tutorial with a dropout bound." arXiv preprint arXiv:1307.2118 (2013).
>
> In particular, building on Theorem 2 (McAllester, 2013), in our proof, we impose Gaussian prior and posterior distributions, and we optimize the variance of these two distributions to balance the PAC-Bayes bound.
>
> The Hessian operator naturally comes out from the proof, as we need to bound the difference between the perturbed loss and the original loss. By using the isotropic perturbation in the proof, we will get the trace of the Hessian.
>
> This is the high level idea of our proof. We have also included a detailed proof sketch in Section 2.4 of our paper.
>
> We thank you for your time in reading through our manuscript and providing critical feedback.
>
> If this response has addressed your concerns, we would appreciate it if you could consider adjusting the evaluation of our paper.
>
> Thank you,
> Authors

---

> > ### Author Response · Authors · 2024-08-03
> > **Sorry for the late response (due to visibility issues)**
> >
> > We would like to thank Reviewer wuA1 again for the detailed comments, which greatly improved the quality of this paper.
> >
> > We are truly sorry for our late response. Actually, we sent out the above two responses very early on (one dated Jun 7 and the second dated Jul 26).
> >
> > However, we only just realized that these two messages are not set to be visible to everyone. We have adjusted these settings so that the original responses are visible to everyone now.
> >
> > Please find our responses to your requested changes in the above message.

---

> > > ### Author Response · Authors · 2024-08-07
> > > **Follow-up response to Reviewer wuA1**
> > >
> > > We would like to know if you had a chance to read our revision, including the added PAC-Bayes bound (and various other improvements and experiments).
> > >
> > > We would like to highlight two key strengths of our results as compared to those in the literature. The first strength is that our PAC-Bayes bound is nonvacuous, meaning that it matches the scale of empirically observed gaps when measured in practice (this is based on the trace measurements in Figures 3 & 4; we are happy to add new figures to substantiate this if requested). The second strength is that this nonvacuous bound has practical implications, meaning that we can base on this bound to design novel optimization algorithms for improving generalization.
> > >
> > > These two strengths are non-trivial to achieve. To give some context, recall that in the work of Neyshabur et al. (ICLR’18), the authors provided a PAC-Bayes margin bound for multi-layer neural networks. Their bound depends on a product of the spectral norm of the layers of the network. While this paper provided important initial insights into the generalization of deep networks, the bound turns out to be vacuous when measured in practice (Arora et al. (ICML’18)). Motivated by this, Arora et al. (ICML’18) provide another data-dependent PAC-Bayes bound based on compression techniques. Their work started with an experiment in which they injected noise into the layers of the network, and showed that deep nets can somehow absorb the noise after retraining. However, their result still scales with orders of magnitude higher than the actual generalization errors observed in practice.
> > >
> > > In contrast, our bound matches the scale of empirically observed gaps. To achieve this, we start from the line of work on *data-dependent* PAC-Bayes bounds. In particular, we build on the Distance from Initialization paper by Nagarajan and Kolter (2019). In the fine-tuning setting, the distance from initialization can be used to characterize the generalization ability of fine-tuning (Li and Zhang (NeurIPS’21)). Our key breakthrough is to connect noise stability in PAC-Bayes bound with the loss Hessian matrix. See the proof sketch on page 7. As the Hessian of loss landscapes can be precisely measured from data (due to the flatness of the loss geometry), our bound matches the scale of empirically observed gaps.
> > >
> > > We additionally note that few existing works have considered using PAC-Bayes bounds to design algorithms for better generalization. An important reason is that in order for new algorithm designs, we need to connect the PAC-Bayes bound with data in a nonvacuous way. The seminal work of Dziugaite and Roy (2017) has provided a computational framework to achieve nonvacuous generalization bounds. However, their bound does not have an analytical expression that can be leveraged in algorithm design. To operationalize the design, we innovatively utilize the explicit dependence of our result on the Hessian to design the regularization scheme, which is effective in practice.
> > >
> > > Finally, we would like to thank you for taking the time to read our paper and provide critical comments, which has greatly improved the paper. We hope the revision (and this response) have met your expectations.
> > >
> > > **References**
> > >
> > > - Neyshabur, Behnam, Srinadh Bhojanapalli, and Nathan Srebro. "A pac-bayesian approach to spectrally-normalized margin bounds for neural networks." ICLR (2018).
> > >
> > > - Arora, Sanjeev, Rong Ge, Behnam Neyshabur, and Yi Zhang. "Stronger generalization bounds for deep nets via a compression approach." International conference on machine learning. PMLR, 2018.
> > >
> > > - Nagarajan, Vaishnavh, and J. Zico Kolter. "Generalization in deep networks: The role of distance from initialization." arXiv preprint arXiv:1901.01672 (2019).
> > >
> > > - Li, Dongyue, and Hongyang R. Zhang. "Improved regularization and robustness for fine-tuning in neural networks." Advances in Neural Information Processing Systems 34 (2021): 27249-27262.
> > >
> > > - Dziugaite, Gintare Karolina, and Daniel M. Roy. "Computing nonvacuous generalization bounds for deep (stochastic) neural networks with many more parameters than training data." UAI (2017).

---

> > > > ### Comment · Reviewer_wuA1 · 2024-08-10
> > > > **Response to the rebuttal**
> > > >
> > > > Thank you for your detailed response, and I am sorry for my late response, too. I also appreciate for accepting many of the points I mentioned. I was able to gain a better understanding of the research outcomes. Below is my understanding of the contribution of this paper.
> > > >
> > > > 1. As the authors mentioned, the main content of this paper lies in their algorithm, NSO, and its empirical analysis. While the theoretical foundation of their competitors, the SAM algorithms, is somewhat opaque in terms of PAC-Bayes theory, this paper at least proves in Theorem 2.1 that their population loss is reasonably bounded compared to the empirical loss using PAC-Bayes theory. This can be seen as an example that supports the superiority of the algorithm well by applying the results of the PAC-Bayes theory, rather than advancing the PAC-Bayes theory itself.
> > > >
> > > > 2. Particularly noteworthy is that Theorem 2.1 of this paper clearly describes the dependency on the dataset size $n$ as $n^{-1/2}$. Many previous papers, according to authors, were found to have been described without indicating any dependency on $n$.
> > > >
> > > > 3. Additionally, unlike other research results that compare $L(w)$ and $\hat{L}(w+\epsilon)$, this paper is notable for comparing $L(w)$ and $(1+\epsilon)\hat{L}(w)$. (Maybe it comes from the differentiability and the Lipschitz condition in Theorem 2.1, I guess?)
> > > >
> > > > If there is any misunderstanding on my part, I would appreciate it if you could point it out.
> > > >
> > > > Through this rebuttal, I have understood that there is considerable interest in this research area. Although the paper does not propose a new PAC-Bayes generalization bound theory, I understand that it can be interpreted as an excellent application. Based on this, I will evaluate the audience and claims accordingly.

---

> ### Author Response · Authors · 2024-08-13
> **RE: Response to the rebuttal**
>
> Thanks for the detailed response! You are correct that Theorem 2.1 critically depends on the Lipschitz-continuity and the twice-differentiability of the loss function on the variables $W$.
>
>
> - Regarding comments on comparing $L(W)$ to $(1 + \epsilon)\hat L(W)$, we have thought deeply about this before and tried to study whether the $\epsilon$ term is really necessary. In the case of fixed-design linear regression, we can reduce this $\epsilon$ term. The reason is that when we apply Taylor’s expansion to the perturbed loss (in PAC-Bayes bound), we can reduce the third-order expansion term based on the closed-form expression of linear regression.
>
> - In the general case, this $\epsilon$ term arises due to the $\sigma^3$ expansion term from applying Taylor’s expansion (see Equation (22) and the lines below on page 22).
>
> - From a practical perspective, our experience is that this $(1 + \epsilon)$ factor of $\hat L(W)$ is negligible because the $\epsilon$ term has been absorbed in the empirical loss, which does not affect our empirical measurements of the trace of the Hessian.
>
> - From a (PAC-Bayes) theory perspective, one technical question is whether we could improve the $(1 + \epsilon)$ term to $1$ by imposing higher-order Lipschitz-continuity conditions. These higher-order Lipschitz conditions are studied in the optimization literature, where having access to higher-order Lipschitz derivatives can lead to better oracle complexity for finding stationary points. This is consistent with your notes regarding different convergence rates in sample size $n$ from various prior works.
>
>
> You are also correct that our main contribution may be viewed more as an application of the PAC-Bayes generalization theory, rather than developing a new PAC-Bayes generalization bound theory. We hope our work can inspire more interest in developing novel applications of this beautiful theory.
>
>
> We would like to thank you once again for your insightful comments, which have greatly reshaped our thinking and substantially enhanced our paper. Thanks also for taking our response & revision into consideration while evaluating the audience and claims.

---

### Review · Reviewer_X1ow · 2024-07-09

**Summary Of Contributions:**

The authors analyze training under weight noise, a scheme which has largely been abandoned for its lack of efficacy despite its potential as an implicit regularizer of statistics of the Hessian. The authors then propose Noise Stability Optimization, which trains with pairs of noised points, where each pair involves perturbations $\pm U$ respectively. They show that this specific coupling reduces the influence of low order terms from a Taylor expansion of the weight-noise objective, and improves finetuning over SAM and related methods in a variety of settings. They additionally provide theoretical analysis of PAC-Bayes generalization bounds as well as convergence analysis of the new method.

**Audience:**

Yes

**Claims And Evidence:**

Yes

**Requested Changes:**

Please edit the abstract. The first part reads like a mathematical proof and not an informative summary for a reader. The overall presentation could use some work as well; there are already detailed mathematical equations in the first paragraph for example.

There are a few concerns about the experiments:
1. It's mentioned in Appendix C that the experiments were done without momentum and weight decay. What happens when these are included? It seems very odd to present results without these basic features of modern deep learning pipelines.
2. The radii selected for SAM may be too large; in general the ratio of $\rho/\eta$ (SAM radius to learning rate) gives a dimensionless number which gives a sense of the strength of SAM.
3. In that vein, unnormalized SAM is a better comparison for NSO. See for example [2]. In this setting one can search over equivalent regularization strengths. Generally $\sigma^{2}$ from NSO should be compared to $\rho$ from SAM - another point which suggests that perhaps the SAM radius was set too large.

Small changes:

In the experiments for Table 1, what are the results training with normal weight perturbation and not the paired perturbation introduced in the paper?

There are many typos and incomplete sentences, e.g.: "In the case that f is a strongly convex function, the lowest eigenvalue of the Hessian is above from below" - what does this mean?

Key definitions like $\ell_{Q}$ are buried in the middle of proofs, even though they are later used in experiments.

Table 2 should be a plot not a table.

One of the key practical features of SAM seems to be its sensitivity to batch size. What batch sizes were experimented with for the NSO-SAM comparisons? Does NSO have a different sensitivity to batch size compared to SAM?

Minor points:
* The introduction discusses the link between edge of stability and SAM, [1] is a key reference here. It also may be relevant for points made about the difficulty of analyzing SAM at convergence - something which may not be relevant for analysis of practical algorithms.
* For the experiments involving the Hessian trace: what activation function was used in the networks? For ReLU networks the trace mainly captures the Gauss-Newton component of the Hessian, see [2] for example
* Proposition 2.1 does not require the paired noise construction to work, including it in the proof is confusing - unless one can also make the point that it reduces the variance of the linear term in finite samples. I think it should be modified to include such a statement.
* Reference [2] has some discussion and experiments about training with weight noise which may also be relevant for the analysis.

[1] https://proceedings.mlr.press/v202/agarwala23a
[2] https://arxiv.org/abs/2401.10809

**Strengths And Weaknesses:**

The overall idea of the method is simple and interesting. It feels like an in-hindsight-obvious modification of weight noise - which is to the authors' credit. The experimental results in the fine-tuning settings seem promising in general. There are however some major concerns with the experiments; these are listed in "requested changes".

The presentation overall is very confusing. It seems to go back and forth between the theory, experiments, and different versions of the algorithms without guidance for the reader. More specific critiques and suggestions can be found in "requested changes".

I'm overall confused at the point that sampling multiple perturbations per iteration is essential to the method's success, and that simultaneously the method works with only 1 sample.

The experimental section would be strengthened if the authors included some experimental results on how NSO operates in (pre)training rather than fine-tuning tasks - an area where SAM tends to do well. I don't think a negative result here would weaken the paper but would instead help practitioners understand more about NSO. There is also a question about the batch sizes used in "requested changes" that I ask the authors address.

I believe the method (and the paper) are promising, but further work is needed to clarify the claims and the presentation. I believe the theoretical analysis is sound but I haven't checked some of the details carefully.

Edit: Claims and evidence update to yes following rebuttal period.

---

> ### Author Response · Authors · 2024-07-25
> **Response to Reviewer X1ow 1/2**
>
> We thank the reviewer for carefully reading through our paper and providing very insightful suggestions. We have revised our paper to incorporate the requested changes, including running additional experiments to compare SAM with our approach in a pretraining setting. Below, we first address the high-level comments and then respond to the other remaining comments.
>
> **Does our approach apply to pre-training settings?** We first note that the main rationale of our approach is to design algorithms that can effectively regularize the Hessian. The noise injection schemes are good for this purpose since the weight noise can provide an unbiased estimate of the trace of the Hessian. With this in mind, it may not be very surprising that our approach should apply both to pre-training and fine-tuning settings.
>
> In the paper, our experiments focus on fine-tuning as we feel that the regularization effect is most pronounced, as this setting directly connects to our generalization bound, which comprises the trace of the Hessian and the radius of the fine-tuning region as compared to the model initialization.
>
> Following the reviewer's suggestion, we have now applied our approach to pre-training contrastive language-image models (CLIP) on a dataset of image-caption pairs. We use the Conceptual Caption data set from Google, which contains 3.3M image caption pairs. Each caption provides a short description of the corresponding image, averaging 10 tokens in length. We use a 12-layer ViT as the image encoder and a 12-layer GPT-2-style transformer as the text encoder. We train the encoders jointly to maximize the cosine similarity between the embeddings of image caption pairs.
>
> We compared our approach with both SAM and SGD, for pretraining CLIP models within the above setting. Our preliminary results show that our approach can effectively reduce the trace of the Hessian by **17**% compared to both SAM and SGD. In addition, our approach also achieves **1.4**% better (validation) recall scores (in image retrieval).
>
> Besides this experiment, we have also applied our approach to another setting, where we are fine-tuning pretrained language models on chain-of-thought reasoning datasets. The task is to generate the reasoning process and the answer for a given commonsense reasoning question. We fine-tune GPT-2 models on both CommonsenseQA and StrategyQA datasets. The results show that our approach can find solutions whose trace of the Hessian is **25**% lower than SAM and SGD and whose test accuracy is **5.3**% higher.
>
> We hope these additional contexts and experimental results help address the reviewer's concern regarding the applicability of our approach.
>
> **Various robustness checks for our approach.** We thank the reviewer for numerous valuable suggestions! We have tested our approach under various settings following the review’s suggestions:
>
> - We compared our approach with SAM, each combined with weight decay and momentum for fine-tuning ResNets on several image classification datasets. For this setting, we find that our approach can achieve 23% lower trace of the Hessian and 1.4% higher test accuracy than SAM.
>
> - We compared our approach with the unnormalized version of SAM (denoted as USAM for short). We vary the radius of USAM between 0.01, 0.02, and 0.05. On the six image classification datasets that we tested on, we find that our approach can yield 13% lower Hessian traces and 1.5% higher test accuracy than USAM.
>
> - To make sure that the radius of SAM is adjusted properly, we vary the radius of SAM and USAM between 0.001, 0.002, and 0.005. In the experiments, we find that using a smaller radius (e.g., less than 0.01) results in better results.
>
> - We used a batch size of 32 in our experiments. Following the reviewer's suggestion, we have further varied the batch size between 8, 16, 32, and 64. We observe that our approach is generally not very sensitive to the changes in batch size; This is unlike the phenomena that has been observed by prior references regarding SAM's sensitivity to batch sizes. Across various batch sizes, the comparative results are fairly consistent, with our approach yielding 19% lower trace values and 1.5% higher test accuracies.
>
> We hope these additional experiments and tests address the reviewer's comments regarding the robustness of our claims.
>
> We have added these results to the experiments section of our paper, which are highlighted in blue color for ease of reading.

---

> ### Author Response · Authors · 2024-07-25
> **Response to Reviewer X1ow 2/2**
>
> Next, we respond to the requested changes remaining from the reviews:
>
> <<< *Please edit the abstract... The overall presentation could use some work as well.*
>
> Thanks for your thoughtful suggestion! We have significantly revised the abstract and introduction based on the feedback. In particular, we have clarified the overall story as our paper went through several rounds of revision. Besides, we added statements regarding our new results in the abstract and the introduction.
>
> <<< *I'm overall confused at the point that sampling multiple perturbations per iteration is essential to the method, and simultaneously the method works with only 1 sample.*
>
> Just to clarify, our algorithm uses a two-point / paired noise injection scheme. Since this is a stochastic optimization algorithm, it is possible to reduce the variance of the stochastic gradient, if we can sample multiple perturbations to take their average (similar to mean estimation, where having more samples can reduce the variance of the estimator). In our algorithm, this is indicated by the parameter denoted as $k$. So, if $k = 1$, it means that the algorithm will sample one random variable $U$ from the noise distribution. It will perturb the neural network according to $W + U$ and $W - U$, where $W$ is the parameters of the network. This results in two gradient queries, which is exactly the same as SAM.
>
> In our experiments, we first want to make sure that we are making a fair comparison, in the sense that our algorithm uses the amount of computation costs as SAM. For this purpose, in the two-point / paired noise injection scheme, we can just set $k = 1$ in our algorithm. This will use two times the costs of SGD, which would be the same as SAM. This is the setting in which we conduct the comparisons in Table 3.
>
> In order to further justify this design choice, we have also compared it with weight-perturbed SGD, which samples two independent perturbations each time. This ensures that WP-SGD also uses twice the computation cost of SGD, which would be the same as SAM. Our experiments show that our approach can still outperform WP-SGD by **3.6**% for this setting.
>
> <<< *In Table 1, why are the results training with normal weight perturbation and not the paired perturbation introduced in the paper?*
>
> Recall that this experiment is used to motivate our study, where we would like to design a modified noise injection scheme, given that the straightforward noise injection implementation does not work well. Table 1 shows precisely this. Therefore, in this Table, we focus on comparing SGD with WP-SGD.
>
> Following your comment, we have now also added the results for the paired perturbation to this Table. Thank you.
>
> <<< *Table 2 should be a plot, not a table.*
>
> Thanks for the suggestion. We have added a plot corresponding to Table 2 in Section 2.2 and moved the Table to Appendix C.
>
> <<< *What activation function was used in the networks?*
>
> We use the ReLU activation function in ResNet-34. We use the GeLU activation function for transformer-based models.
>
> <<< *Proposition 2.1 does not require the paired noise construction to work.*
>
> Thanks for this suggestion. The reviewer is correct that this proposition does not require the paired noise injection to work. Based on Reviewer Z2QG's suggestion, we have decided to remove this proposition and instead replace it with intuitive descriptions of the ideas, since the proof is fairly standard. This helps reduce the mathematical notations in the paper.
>
> <<< *References of Agarwala and Dauphin (2023), and Dauphin et al. (2024).*
>
> Thanks for suggesting these very interesting references! We have read through both references carefully and have now included a discussion of these references in the related work section of our paper.
>
> Our understanding is that both papers are working on understanding the gradient dynamics of Sharpness-Aware Minimization, which is a very intriguing topic, but beyond the scope of our work. In particular, as the reviewer is aware, the dynamics of SAM have a fairly intricate structure.
>
> Instead, our approach may be viewed as an alternative way to instantiate the regularization of the Hessian. Through the noise injection schemes, we have found that, indeed, by modifying the naive weight noise, we can achieve significant performance gains using noise injection.
>
> As the reviewer is aware, SAM is the dominating approach for improving generalization in practice, moving beyond SGD. This doesn't mean that it should be the only approach worth pursuing or studying. Our work is providing an alternative approach to improving generalization. While this may be the less popular approach right now, our theoretical and empirical evidence suggest that this approach is promising. We believe that this alternative perspective is new and should be of interest to the community.
>
> We are happy to answer any further questions should the reviewer have any concerns about the significance of our experiments or the clarity of our paper.

---

> > ### Comment · Reviewer_X1ow · 2024-07-31
> > **Response to the rebuttal**
> >
> > I thank the authors for their very very detailed response! The paper is much improved, I appreciate their point by point response.
> >
> > A few minor points:
> > * For Figure 2, should there be a different y-axis for the difference in loss and difference in trace? Or do they really end up being comparable?
> > * In the abstract, is it possible to motivate the method in a few words before referring to the PAC-Bayes bound? The overall idea is quite elegant and doesn't require learning theory to understand, I wonder if this can be reflected in the abstract?
> > * Table 4 is IMO much more compelling than table 3. Is it possible to add SGD (with momentum etc) to table 4, and maybe put the full table 3 in an appendix? This might be easier for the reader, while still preserving the rigorous testing.
> > * Can table 6 also be made into a plot? I find the results on batch size very intriguing.
> >
> > I have updated claims and evidence to "yes" to reflect the massive improvements made to the paper by the authors.

---

> > > ### Author Response · Authors · 2024-08-03
> > > **RE: Response to the rebuttal**
> > >
> > > We thank the reviewer for acknowledging our response! We have revised the paper to incorporate the minor points as follows:
> > >
> > > - For Figure 2, the trace values are multiplied by $\sigma^2 / 2$. We have revised the labels and clarified this in the caption as well. Thanks for catching this issue!
> > >
> > > - We added a sentence in the abstract to describe the main idea for canceling out the first-order expansion term in the estimation of the Hessian.
> > > - We have re-run all the baselines in this table and reported the results here. For ease of reading, we only included results for SGD, SAM, unnormalized SAM, and adaptive SAM (all trained with momentum and weight decay) to Table 3. As you suggested, we left the other baseline comparisons to the appendix.
> > >
> > > - We have converted Table 6 into a figure in Section 3.1. Thanks for the constructive feedback!

---

### Review · Reviewer_Z2QG · 2024-07-11

**Summary Of Contributions:**

This paper proposes a new algorithm to inject noise in the weights during training of a Neural Network. The noise injection is known to make the training more stable and improve the generalization of the model, thus making it more robust with respect to being trapped in local minima. A crucial difference of the proposed method wrt extisting algorithms is to use negative perturbations to the model's weights in order to zero out the first-order in Taylor's expansion of the function of the weights, i.e., $F(W)$ in the paper, and in turn lead to a better estimate of the gradient.

The theoretical insights presented in the first part of this paper are validated with an extensive set of numerical experiments showcasing the advantages of the proposed method.

As a side note to the authors and the AE: I've noticed a new version of the manuscript (revised) has already been submitted. Unfortunately, the version I reviewed was the original one. The authors should please keep this in mind when going through my comments. It could be that some of the concerns raised below have already been addressed.

**Audience:**

Yes

**Broader Impact Concerns:**

I could not see any concerns regarding the ethical implications of the submitted work.

**Claims And Evidence:**

Yes

**Requested Changes:**

### Requested changes, recommendations and questions

- I'd recommend running a spellchecker through the manuscript and checking all the cross-refs in LaTeX.
- I'd recommend editing the structure of the manuscript by removing (or at the very least heavily restructuring) page 2. In particular, I'd recommend focusing on the very general idea in very intuitive and easy terms, anticipating the structure of the paper and clearly listing the main contributions.
- I'd recommend making a sketch/visual that represents the core idea of this work to put on page 2 instead of (or along with) the example 1.1. This should help the reader to better follow the flow.
- Make sure to introduce all acronyms. For example, Sharpness-Aware Minimization has never been explicitly defined as SAM, and it took me a while to realize this. The same may apply elsewhere.
- Algorithm 1: I'd suggest to add a comment on line 2: highlighting that the loop is over different perturbations.
- I feel that Figure 1 is out of place on page 3. I'd rather, as mentioned earlier, show a visual describing how the algorithm works and leave this for the result section, after the theory part.
- Currently, the bullet points that summarize the contributions of this work are very verbose and do not go straight to the points. I'd recommend shortening them and stating more clearly what the modified algorithm actually does.
- I cannot see where the variable $\epsilon$ in the middle of page 5 has been defined.
- To which extent are Assumptions 3.2 and 3.3 *standard*? I think the clarity of the paper would benefit if the authors could give a brief explanation of why those assumptions could be considered "easy to realize" in practice.
- Section 3.2 uses a specific example to show that gradient norms bounds are tight. I wonder how general this result is and if it is really necessary for the development of the main story of the paper. While reading the manuscript I perceived this section rather as an add-on instead of a necessary contribution (in light of the results shown in the **Experiments** section). In order to enhance clarity and streamline the reading, I wonder if it would be a suitable option to relegate all these details to the appendix and perhaps spend only a couple of sentences describing this in the main text. It may also be the case that I am overlooking something here. In that case, I'd be grateful if the authors could make me aware of this.
- If necessary, I'd introduce Example 1.1 at the beginning of Section 4.1. People who are not strictly familiar with the field may have forgotten what Example 1.1 was when they reached this point of the paper.
- I find Table 2 a bit misplaced. This is not really a *contribution of this work* but rather an ablation study of SGD and WP SGD. While I think this is still interesting to show, I'd add NSO there as well to compare directly with the method introduced in the manuscript.

### Open Question to Authors

- At the bottom of page 11, the authors note that using $k\geq 3$ perturbations does not bring any obvious improvement. Have the authors made any investigation about the correlation between the number of perturbations and the level of noise injected? In other words, if I think of making parallelism with annealing, I might imagine that having more levels of perturbations, each of which injects less noise at each step, might make the procedure even more robust. Do the authors have an intuition as to whether this speculation is correct and why?

**Strengths And Weaknesses:**

## Strengths

- The paper is well structured, and the idea is clearly presented.
- The results unequivocally show the superior performance of the proposed algorithm.
- The authors have done a very thorough numerical analysis to validate many aspects of the proposed work also when combined with other existing methods. I found particularly interesting the ablation study.
- I find the main idea behind this work to be intuitive, natural, yet very effective.


## Weaknesses
- I find the writing often lacking clarity. For example, some sentences are particularly convoluted, and repetitions sometimes occur (e.g., I assume there are some problems with cross-referencing in latex as it sometimes appears as "equation equation X").
- I find the structure suboptimal. The **Results** section on page 2 feels a bit out of place and breaks the flow of the paper. Moreover, I find the paper very hard to follow from Example 1.1 onwards. Although I assume the goal of the authors was to provide examples to enhance clarity, this had the exact opposite effect on me (for context, I am not an expert on this specific subfield, so most of the techniques discussed in this work were new to me).
- For being part of the **introduction** Example 1.1 already gets to technical and I felt lost already because of the heavy notation.
- I found the **related work** section to be hard to parse. In fact, this is, at times, repetitive with respect to the previous section. Moreover, the authors discuss some evidence from their empirical analysis in the **related work** section, which I found confusing. I'd recommend leaving out any comment on the present work and just focusing on the findings, advantages, and limitations of previous works.
- From page 5 until page 8, I found the paper extremely hard to follow. Specifically, I find the notation particularly heavy and dense. I'd like to know if this can be restructured in a way that highly technical details and proofs are left out and the focus is on the main contribution, i.e., injecting noise in both positive and negative directions.

---

> ### Author Response · Authors · 2024-07-25
> **Response to Reviewer Z2QG**
>
> We are very grateful for the reviewer's careful reading of our paper. Thanks for all of the constructive comments! We have significantly revised the paper and did our best to incorporate the reviewer’s suggestions and requested changes into the paper. Below, we describe the revisions to the requested changes.
>
> **Changes to the paper’s presentation.** We have significantly revised the paper according to the reviewer’s suggestions. In particular, we have significantly restructured Section 2. As a result, we have also rewritten the introduction and the abstract, as well as the related work section. The main change points here are to reduce the mathematical notations like the reviewer suggested, and instead focus on the core ideas of our approach.
>
> In addition, we have now added a visualization of our algorithm in Section 2, as well as a Table to highlight the key comparisons between our approach and Sharpness-Aware Minimization.
>
> Finally, we have carefully went through the reviewers' requested changes and made any necessary revisions as needed, including:
> - Adding a comment in Algorithm 1.
> - Moving Figure 1 to the experiment section.
> - Checking typos and introducing all acronyms appropriately.
>
> **Other requested changes.**
>
> <<< *I cannot see where the variable $\epsilon$ in the middle of page 5 has been defined.*
>
> $\epsilon$ denotes the gap between the perturbed loss $\ell_{\mathcal{Q}}$ and the original loss $\ell$. We have revised this notation to make it clear in the paper.
>
> <<< *To which extent are Assumptions 3.2 and 3.3 standard? Why could those assumptions be considered "easy to realize" in practice?*
>
> Assumption 3.2 requires the stochastic gradient to be unbiased and have bounded variance. For instance, in SGD, the stochastic gradient is unbiased, and then the variance of the stochastic gradient is bounded based on the maximum length of the gradient vector. Assumption 3.3 requires the function to be Lipschitz continuous. Geometrically, this condition says that the gradient should not change very quickly. These assumptions are commonly used in the optimization literature. Checking their validity is tedious but can be done, and they are expected to hold in typical deep-learning scenarios. We are happy to conduct more experiments to provide further evidence along these two assumptions if the reviewer would like to see additional evidence.
>
> <<< *Section 3.2 uses a specific example to show that gradient norms bounds are tight. I wonder how general this result is and if it is really necessary for the development of the main story of the paper.*
>
> Thanks for this comment. Indeed this lower bound example serves as part of the convergence analysis. We agree that the excessive mathematical notations in this part are not necessarily helpful for building up the storytelling. Hence, we have decided to move the original Section 3.2 to a later part of the paper now. Instead, we are using Section 2 to better highlight the idea of our approach.
>
> <<< *I find Table 2 a bit misplaced… I'd add NSO there as well to compare directly with the method introduced in the manuscript.*
>
> We conducted this experiment to better understand how noise injection works. The straightforward way is to sample independent perturbations, which is termed WP-SGD in the Table. Our findings suggest that WP-SGD introduces marginal empirical improvement over SGD.
>
> Following the reviewer's suggestion, we have now added the results for NSO in this Table as well. In particular, we vary the noise distribution and find that NSO achieves 3.6% higher test accuracy than WP-SGD.
>
> <<< *Using $k\geq3$ perturbations does not bring any obvious improvement. Have the authors made any investigation about the correlation between the number of perturbations and the level of noise injected? Having more levels of perturbations, each of which injects less noise at each step, might make the procedure even more robust.*
>
> Thanks for this excellent question. In our experiments, we did not find a correlation between the noise level and the number of perturbations. The optimal noise levels remained consistent across various numbers of perturbations.
>
> Following the suggestion, we have further tested NSO by gradually increasing the noise level during training, which gradually increases the regularization strength. Analogous to learning rate schedules, we explored two schedules for the noise standard deviation, $\sigma$: one that linearly increases $\sigma$ to a specified level, and another that exponentially increases $\sigma$. Our preliminary experiments find that neither schedule significantly improved test performance compared to NSO with a constant noise level.
>
> We have added this discussion to the paper.

---

### Review · Reviewer_v3GM · 2024-07-13

**Summary Of Contributions:**

Recent literature has extensively studied the training of overparameterized neural networks. This paper introduces a Hessian-based regularization approach for training these networks, focusing on noise injection algorithms. These algorithms have a desirable theoretical property that allows for an unbiased estimation of the Hessian. However, the paper finds that naive noise injection does not perform well empirically. To address this issue, the authors propose a new two-point noise injection algorithm that effectively cancels out the first-order gradient term. This algorithm demonstrates strong performance in practice, as evidenced by tests on various image classification datasets. Additionally, the paper derives generalization bounds and convergence rates to theoretically support the proposed algorithm.

**Audience:**

Yes

**Claims And Evidence:**

Yes

**Requested Changes:**

- It would be helpful to polish the writing of the paper.
- To generate broader interest, it would be beneficial to demonstrate the effectiveness of the proposed algorithms in additional application domains.

**Strengths And Weaknesses:**

Strengths:

- This paper offers theoretical guarantees and extensive empirical evidence to support the effectiveness of the proposed algorithm.
- The insights into noise injection algorithms are promising and could potentially stimulate future research on the geometric properties of training large models, such as transformers, as well as the development of new regularization schemes.

Weaknesses:
- I noticed that the authors uploaded a revision. The revised paper seems somewhat unpolished.
- To support the arguments in the paper, the algorithms could be tested on a wider range of applications and datasets.

---

> ### Author Response · Authors · 2024-07-26
> **Response to Reviewer v3GM**
>
> Thanks for the reviewer’s feedback on our paper. Regarding the applicability of our approach, it is worth noting that, just like sharpness-aware minimization, our algorithm could be used as an off-the-shelf optimizer, for improving generalization. We have now tested our approach for fine-tuning pretrained language models on chain-of-thought reasoning datasets. In particular, the task here is to generate the reasoning process (otherwise also known as chain-of-thoughts) and the answers, given some commonsense reasoning questions (such as arithmetic calculations, etc).
>
> In the experiments, we fine-tune GPT-2 models, which are decoder-only transformer neural networks, on CommonsenseQA and StrategyQA datasets. In particular, we build on the implementation of Ho et al., ACL'23:
>
> - Ho, Namgyu, Laura Schmid, and Seyoung Yun. "Large Language Models Are Reasoning Teachers." 61st Annual Meeting of the Association for Computational Linguistics, ACL 2023. Association for Computational Linguistics (ACL), 2023.
>
> To apply our approach, we replace the SGD algorithm of the training process with our approach (ie. NSO, given in Algorithm 1).
>
> The experimental results also show that our approach is rather effective in this setting. In particular, compared to sharpness-aware minimization and SGD, the solutions obtained using our approach achieved 5.3% higher test accuracy.
>
> When we evaluate the trace of the Hessian, based on the network trained at the last epoch, we find that for the network trained by our algorithm, the trace of the Hessian is 25% lower, compared to the above two alternative training methods.
>
> We have added this experiment to Section 3.1, Page 11, of the revised paper.
>
> In addition to running this experiment, we have gone through the main text and revised various places to improve the coherence and the clarity of our paper. Thank you.

---

### Decision · Action_Editor_D1z7 · 2024-08-16

**Recommendation:** Accept with minor revision

**Comment:**

After a detailed and productive discussion, the paper received uniformly positive reviews, with reviewers appreciating the simplicity of the approach and strong experimental results. I second these impressions and recommend acceptance to TMLR. However, I noticed a few points that still need to be addressed in the final revision of the paper.

1. The theoretical guarantees in this work hold equally for WP-SGD and NSO. This is a significant limitation, because a main (empirical) claim of the paper is that NSO generalizes better than WP-SGD. The revision needs to state this limitation clearly, in a paragraph or section titled “Limitations.”

2. The above point makes the empirical superiority of NSO to WP-SGD mysterious. Viewing NSO/WP-SGD simply as implementation of Hessian-regularized ERM, the lower variance of the NSO gradient estimator should only speed convergence, not improve generalization. Please discuss this point and try to provide some intuition for why NSO generalizes better.

3. The experiment in Section 5 needs some clarification:
   * Are both “NSO” and “WP-GD” applied on exact empirical gradients (as the name of the latter implies?)
   * In light of points 1 and 2 above, does the generalization gap between NSO and WP-GD persist even when thoroughly tuning the learning rate and number of steps for each method? Demonstrate this by showing the test MSE as function of learning rate and number of steps.

4. The novelty of the theoretical components in this paper is also not sufficiently clear. In Section 2 and Section 4, the authors need to clearly state what is the difference between their result and the closest results in prior work, and what are the new ideas. In particular, Theorem 4.2 seems to be a simple corollary of the results in Ghadimi and Lan (2013), i.e., there are no new ideas. To make this clearer please, the proof of Theorem 4.2 should cite Theorem 2.1 of Ghadimi and Lan in a black box way, rather than re-prove it from scratch.

5. The paper is missing a reference to Duchi et al.’s work on randomized smoothing [1], which to my knowledge was the first to propose what you call WP-SGD. More recently, randomized smoothing also became popular as a means for obtaining provable adversarial robustness guarantees [2,3]. The revision should cite and discuss these works.

6. A number of formal statements need improvements:
    * Generalization bounds such as Theorem 2.1 are missing an all-important “for all” quantifier. For example, Theorem 2.1 needs to have a “for all $W\in\mathcal{H}$” appear right before equation (2). (I did not verify this indeed holds, but if not then Theorem 2.1 is not a PAC guarantee.)
    * In Theorem 4.2, the statement “$f$ is Lipschitz-continuous” should be “$\nabla f$ is $C$-Lipschitz continuous”; this error seems to partially propagate to the proof. Note that by using the analysis from [1] one can also obtain a guarantee assuming only that $f$ (and not $\nabla f$) is Lipschitz, but this comes as a cost of introducing dimension-dependence to the rate of convergence.


[1] Duchi, John C., Peter L. Bartlett, and Martin J. Wainwright. "Randomized smoothing for stochastic optimization." SIAM Journal on Optimization 22.2 (2012): 674-701.

[2] Cohen, Jeremy, Elan Rosenfeld, and Zico Kolter. "Certified adversarial robustness via randomized smoothing." international conference on machine learning. PMLR, 2019.

[3] Yang, Greg, et al. "Randomized smoothing of all shapes and sizes." International Conference on Machine Learning. PMLR, 2020.

**Audience:**

The paper deals with a core topic of machine learning and would therefore be relevant to many in the TMLR audience.

**Claims And Evidence:**

The paper’s claims about the practical benefits of the proposed NSO method are backed by thorough and compelling experimental evidence.

However, the paper’s theoretical claims require clarifications about their scope and novelty that can be made in a minor revision - see comments below for details.

---

> ### Author Response · Authors · 2024-08-31
> **Response to Comments**
>
> Response to Comments
>
> 1. We agree that Theorem 2.1 in the stated form does not differentiate between NSO and WP-SGD. To use this result, once we measure the Hessian from data, we will see the difference between these two algorithms. To illustrate this, we have added the dynamics of WP-SGD in Figure 3, which compares with NSO during training. We have also discussed this dependence on the data in a paragraph discussion the limitations in Section 7.
>
> 2. Our intuition is that NSO explicitly cancels out the first-order gradient term, thus eliminating the variance of this term, while giving an approximately unbiased estimate of the Hessian penalty. By contrast, WP-SGD still inherits the variance of this first-order term for estimating the Hessian penalty. This variance may be related to some higher-order derivatives of the loss surface. We added a remark in Section 3.1.2 to discuss this aspect.
>
> 3. Yes for Section 5’s simulation, we compute exact gradients instead of stochastic gradients. Regarding the point about convergence speed, note that we have trained for long enough until the loss curve has fully converged. As for the learning rate, we have added more details about how we selected this and included another figure with a different learning rate, which still gives very similar results.
>
> 4. Yes for Theorem 4.2, we follow the proof of Ghadimi and Lan (2013). There are some additional technical details such as dealing with noise injection, handling multiple injections, etc. Hence, it may be better to have complete proof rather than calling their result in a black box. We have downgraded this result to a Proposition now to avoid any further confusion. We have also clarified the sentences with regard to this convergence rate result in the introduction.
>     - We added Remark 2.2 & Remark 4.5 to discuss the novelty of Sections 2 & 4, respectively. In particular, the novelty of Section 2 is that we provide a non-vacuous generalization bound using Hessian trace and operationalize this for algorithm design. The novelty of Section 4 is that we draw a connection between convergence analysis of finding flat minima and stochastic optimization.
>
> 5. Thanks for pointing out these references! We are not aware of this line of work at the time of writing and agree that they are closely related to our study. We have cited these papers now in Section 6 and various other places when needed.
>
> 6. Thanks for catching these issues! We have now fixed them in the revision.
>
>     - Theorem 2.1 applies to all $W$ in the hypothesis space in the sense of a uniform convergence guarantee as we use the trace norm $\alpha$ and the $\ell_2$ radius of the hypothesis space to bound the Hessian and KL terms, respectively, for all $W$ in the hypothesis space.
>     - In the proof of Proposition 4.2, we use the assumption that $\nabla f$ is $C$-Lipschitz continuous. We have revised the statement to fix this issue and went through the proof to double-check this issue.
>
> We have addressed these points in the revision. We have also proofread the paper again to check any lingering grammar/writing issues as several reviewers have pointed out. Thanks for carefully reading through our manuscript and providing constructive feedback!